# Bazedoxifene reverses sexually dimorphic autistic-like abnormalities in biallelic MDGA1-mutant mice

Seungjoon Kim [1,2,20], Hyeonho Kim[1,2,20], Javier Porta Pelayo[3,20], Sara Alvarez[4,20], Gyubin Jang [1,2], Jinhu Kim[1,2], Byeongchan Kim[1], Victoria M Hoelscher [5], Beatriz Calleja-Pérez[6], Hyunsu Jung[7], Yeji Yang[8,9], Hea Ji Lee[8], Jihae Lee[10], Seoyeon Kim[11,12,13], Mar Jiménez de la Peña[14,15], Yelin Lee[1], Sohye Kim[1], Ah-reum Han[16], Dong Sun Lee[16], Sangho Ji[1], Wookyung Yu [1,2], Ho Min Kim [9,16], Joon-Yong An[10,11,12,13], Won Chan Oh[5], Seok-Kyu Kwon[7,17], Jin Young Kim[8,18], Ji Won Um [1,2], Alberto Fernández-Jaén [18,19 ✉] & Jaewon Ko [1,2 ✉]

## Abstract

**MDGA1 reportedly suppresses GABAergic synaptic inhibition and may be associated with schizophrenia. However, it has been unclear whether and how MDGA1 dysfunction causes neurodevelopmental disorders. Here, we describe two patients with autism spectrum disorder (ASD) carrying missense mutations in *MDGA1*: p.Val116Met/p.Ala688Val and p.Tyr635Cys/p.Glu756Gln. Murine in utero overexpression of MDGA1 p.Val116Met/p.Ala688Val alters normal cortical neuron migration and impairs ultrasonic vocalizations (USVs). The p.Tyr635Cys/p.Glu756Gln substitution disrupts the triangular extracellular structure of MDGA1 and renders it unable to impact GABAergic synapses in hippocampal CA1 neurons. Male *Mdga1* knock-in (KI) mouse pups and adults harboring the p.Tyr636Cys/p.Glu751Gln mutation exhibit impaired USVs and sensorimotor gating, similar to male *Mdga1* conditional knockout (cKO) mice. No behavioral deficits were seen in female counterparts. Bazedoxifene (a selective estrogen receptor modulator) treatment of male *Mdga1*[Y636C/E751Q] KI mice rescues the changes in the expression and phosphorylation of a subset of GABAergic synaptic proteins, as well as behavioral performance and GABAergic synaptic strength. Thus, different *MDGA1* mutations manifest as distinct MDGA1 dysfunctions and are likely to cause ASD via sexually dimorphic loss-of-function and/or gain-of-function mechanisms.**

**Keywords** MDGA1; Autism Spectrum Disorder; Inhibitory Synapse; Communication Deficits; Social Deficits

Subject Category Neuroscience

## Introduction

Mounting evidence from human genetic studies implicates synaptic proteins in the pathogenesis of neuropsychiatric disorders, such as autism spectrum disorders (ASDs), intellectual disability, and schizophrenia (Lee et al, 2017; Lima Caldeira et al, 2019; Torres et al, 2017). Thus, these disorders are considered synaptopathies, which researchers have attempted to model in various animal species (Brose et al, 2010; Lepeta et al, 2016). However, limitations and challenges in producing appropriate animal models of neuropsychiatric disorders still hinder our understanding of the precise mechanisms of the pathogenesis and behavior (Nestler and Hyman, 2010; Südhof, 2018). Moreover, although a series of converging mechanisms have accounted for ASD-relevant dysfunctions at glutamatergic synapses (Fernandes and Carvalho, 2016; Jang et al, 2017; Kettenmann et al, 2013; Lima Caldeira et al, 2019), the molecular and cellular mechanisms underlying GABAergic synaptic dysfunctions remain largely enigmatic.

Meprin, A-5 protein, and receptor protein tyrosine phosphatase mu domain-containing glycosylphosphatidylinositol anchor protein 1 (MDGA1) is a cell surface glycoprotein attached to the cell membrane by a glycosylphosphatidylinositol motif (Litwack et al, 2004; Ko, 2025). It functions throughout the central nervous system

[1]Department of Brain Sciences, Daegu Gyeongbuk Institute of Science and Technology (DGIST), Daegu, Korea. [2]Center for Synapse Diversity and Specificity, DGIST, Daegu, Korea. [3]Genomics, Genologica, Málaga, Spain. [4]Genomics, NimGenetics, Madrid, Spain. [5]Department of Pharmacology, University of Colorado School of Medicine, Anschutz Medical Campus, Aurora, CO, USA. [6]Pediatric Primary Care, C.S. Doctor Cirajas, Madrid, Spain. [7]Brain Science Institute, Korea Institute of Science and Technology (KIST), Seoul, Korea. [8]Digital Omics Research Center, Korea Basic Science Institute (KBSI), Ochang, Korea. [9]Department of Biological Sciences, Korea Advanced Institute of Science and Technology (KAIST), Daejeon, Korea. [10]School of Biosystem and Biomedical Science, College of Health Science, Korea University, Seoul, Korea. [11]Department of Integrated Biomedical and Life Science, Korea University, Seoul, Korea. [12]BK21 Four R&E Center for Learning Health Systems, Korea University, Seoul, Korea. [13]L-HOPE Program for Community-Based Total Learning Health Systems, Korea University, Seoul, Korea. [14]Neuroimaging, Hospital Universitario Quirónsalud, Madrid, Spain. [15]Universidad Europea de Madrid, Madrid, Spain. [16]Center for Biomolecular & Cellular Structure, Institute for Basic Science (IBS), Daejeon, Korea. [17]Division of Bio-Medical Science & Technology, KIST School, Korea University of Science and Technology (UST), Daejeon, Korea. [18]College of Pharmacy, Chung-Ang University, Seoul, Korea. [19]Department of Pediatric Neurology, Hospital Universitario Quirónsalud, Madrid, Spain. [20]These authors contributed equally: Seungjoon Kim, Hyeonho Kim, Javier Porta Pelayo, Sara Alvarez. ✉E-mail: aferjaen@telefonica.net; jaewonko@dgist.ac.kr

(CNS) development (Connor et al, 2019; Um and Ko, 2017); particularly, its role in regulating cortical neuron migration in the superficial layers has been extensively investigated (Ishikawa et al, 2011; Perez-Garcia and O'Leary, 2016; Takeuchi et al, 2007; Takeuchi and O'Leary, 2006). However, MDGA1 continues to be expressed at the adult stage of CNS development (Lee et al, 2013), suggesting its putative role in the later phase of CNS development. In keeping with this observation, MDGA1 negatively regulates GABAergic synapse maintenance by binding to a subset of synapse organizers, such as neuroligin-2 (Nlgn2) or amyloid precursor protein (APP) (Kim et al, 2022a; Lee et al, 2013; Pettem et al, 2013). Intriguingly, it appears that MDGA1 targets APP to tune specific GABAergic neural circuit properties in the hippocampus (Kim et al, 2022a). Although the contribution of the Nlgn2-binding activity of MDGA1 to hippocampal synapse organization cannot be ruled out, it is plausible that MDGA1 might deploy distinct synaptic pathways to modulate GABAergic synapse development. MDGA1 has been associated with schizophrenia and bipolar disorder (Kahler et al, 2008; Li et al, 2011). Moreover, MDGA1 knockout (KO) mice exhibit impaired prepulse inhibition of the startle response with altered dopamine and serotonin metabolism (Hossain et al, 2020).

In the present study, we identified two missense mutations in MDGA1 in two patients with autistic features. Extensive gain-of-function (GoF) and loss-of-function (LoF) analyses in cultured neurons and hippocampal CA1 neurons revealed that the two MDGA1 variants nullify the ability of MDGA1 wild-type (WT) to regulate neuronal migration and synaptic function. Moreover, the genetic introduction of one of these mutations (p.Tyr636Cys/p.Glu751Gln; Y636C/E751Q; human MDGA1 Glu756 corresponds to mouse MDGA1 Glu751) into mice recapitulated Mdga1 KO phenotypes, including abnormalities in USV. Our study provides novel insights into a potential synaptopathy mechanism contributing to ASD-related phenotypes, highlighting the pathophysiological significance of a previously underappreciated aspect of the structural features of MDGA1.

# Results

## Clinical manifestations of the two patients with *MDGA1* variants and genetic analyses

Two patients with autistic features were referred to our clinical institute. The first patient (P1) is a 5-year-old girl diagnosed with moderate psychomotor impairment with autistic features, according to clinical and neuropsychological evaluations (IQ of 63; ADOS score of 13). On examination, she showed some mild dysmorphic features: frontal bossing, prominent eyes, and a thin tented upper lip. The second patient (P2) is a 5-year-old boy with low intellectual functioning and autistic features, also based on clinical and cognitive assessments (IQ of 71; ADOS score of 19). He displayed a prominent forehead, blepharophimosis, and a thin vermilion of the upper lip. The family histories of both patients were not relevant; their initial neurodevelopmental milestones were significantly delayed. Both patients showed severe delays in verbal and non-verbal communication, as confirmed in their respective neuropsychological evaluations. Routine laboratory screening, neurophysiological tests (electroencephalography and auditory evoked potentials), and conventional genetic studies (karyotype and aCGH arrays) revealed no significant abnormalities.

Whole-exome trio analysis revealed compound heterozygous *MDGA1* variants in both cases (Table EV1). A paternally inherited missense variant (c.2266 G > C; p. Glu756Gln) and a maternally inherited missense variant (c.1904 A>G; p. Tyr635Cys) of *MDGA1* were identified in patient P1 (Fig. 1A–D). Both variants were classified as variants of uncertain significance (VUS). An extremely low frequency was observed for both variants in the Genome Aggregation Database (gnomAD), with a maximal frequency of 0.07% and 0.001%, respectively, and no healthy homozygotes. The gnomAD missense Z-Score of 2.57 (greater than 0.647) and the CADD (Combined Annotation-Dependent Depletion) scores of 32 and 27.2 predicted a deleterious effect. In the proband of patient P2, a paternally inherited missense variant (c.346 G > A; p. Val116Met) and a maternally inherited missense variant (c.2063 C > T; p. Ala688Val) of *MDGA1* were observed (Fig. 1A–D). These variants were classified as variants of uncertain significance and likely benign, respectively. An extremely low frequency was observed for both variants in the gnomAD population database (with a maximal frequency of < 0.001% and 0.07%, respectively, and no healthy homozygotes). Their CADD scores of 23.6 and 21.3 suggested a deleterious effect. These mutations were also observed in a compound heterozygous state in two of his three brothers (one with mild psychomotor impairment and one who is a very young infant). While direct measurement of MDGA1 protein levels in patient-derived biospecimens would be valuable, practical limitations, including challenges in obtaining patient consent, precluded their collection for the current study.

Segregation analysis confirmed a compound heterozygous state, as both were identified in a heterozygous state in both the mother and father, consistent with an autosomal recessive inheritance; these mutations were confirmed by Sanger sequencing in both cases. In the WES-trio analysis, no other missense variants with a CADD score over 15 or LoF variants with a clear phenotypic association or a compatible segregation pattern were identified. In both cases, brain 3-T MRI did not reveal any significant structural malformations (Appendix Fig. S1). However, in patient P1, diffusion tensor imaging with 3D-tractography reconstruction showed a marked asymmetry of the superior longitudinal fasciculus (Appendix Fig. S1).

## Cellular and temporal expression of *MDGA1* in human brain development and autism spectrum disorder

To investigate *MDGA1* expression throughout brain development, we utilized a single-cell atlas of the neurotypical human post-mortem brain (Data ref: Kim et al, 2024b). The atlas collated eight single-nucleus and single-cell transcriptomic datasets from prior studies, covering the developmental stages across the human lifespan (from gestational week 7–90 years) (Fig. EV1A–C). Among the 41 clusters comprising the atlas (Fig. EV1C), *MDGA1* showed the highest expression in cluster 12 (C12). This cluster comprises excitatory neurons of early cortical development, mainly fetal second-trimester cortical excitatory neurons (Fig. EV1D). *MDGA1* appears to be highly expressed in immature cortical excitatory neurons, particularly from the mid-fetal period to infancy (gestational week 15 to postnatal week 70) (Fig. EV1E). We confirmed the excitatory neuron-specific expression of MDGA1

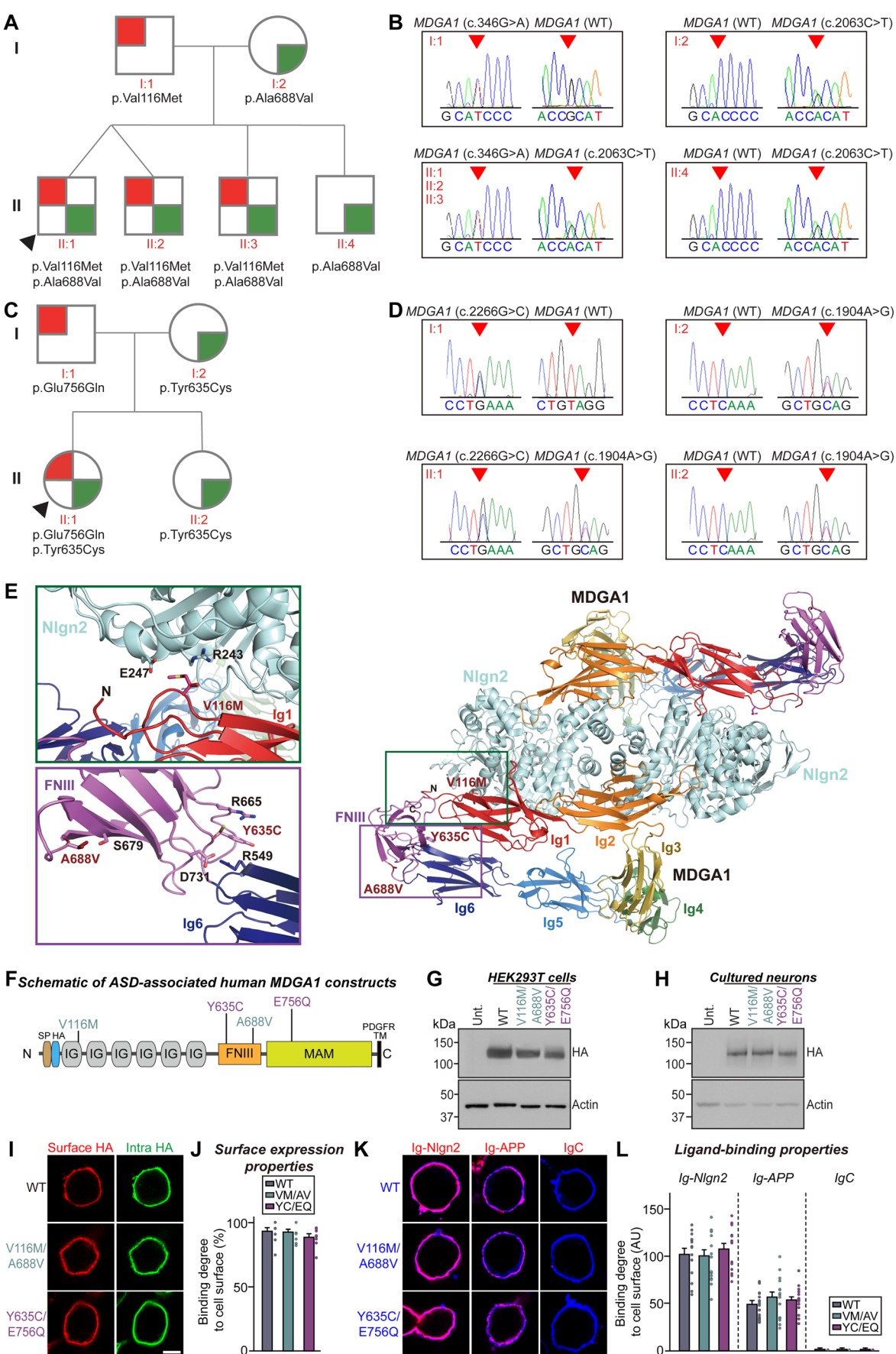

◄  **Figure 1.  Genetic analysis of human MDGA1 missense mutations found in individuals with autism spectrum disorders and the biochemical and cellular properties of these mutations.**

(A) Pedigree of a family with MDGA1 (c.346 G > A and c.2063 C > T) mutations. Squares represent males, circles represent females, and arrows indicate affected individuals. Individuals with the c.346 G > A (p.Val116Met) mutation are denoted by red symbols, while those with the c.2063 C > T (p.Ala688Val) mutation are denoted by green symbols. (B) Sequencing chromatograms showing the MDGA1 (c.346 G > A and c.2063 C > T) mutations. The chromatograms display the nucleotide sequences surrounding the mutation sites. Genomic DNA was isolated from peripheral blood samples of probands and parents, and mutations were verified by Sanger sequencing. (C) Pedigree of a family with MDGA1 (c.1904 A>G and c.2266 C > C) mutations. Symbols and color codes are as described in (A). Individuals with the c.1904 A>G (p.Glu756Gln) mutation are denoted by red symbols, and those with the c.2266 G > C (p.Tyr635Cys) mutation are denoted by green symbols. (D) Sequencing chromatograms showing the MDGA1 (c.1904 A>G and c.2266 G > C) mutations. The chromatograms display nucleotide sequences surrounding the mutation sites. (E) Structural model of the hetero-tetrameric human MDGA1/neuroligin complex (PDB: 5OJ6; PDB: 5XEQ) (Elegheert et al, 2017; Kim et al, 2017) and patient mutations in human MDGA1. For clarity, only the crystal structure of Nlgn2 and the aligned structural model of human MDGA1 are presented. Domains of human MDGA1 (six Ig domains and one FnIII domain) are colored with a rainbow gradient from Ig1 (red) to FnIII (violet). Dimeric Nlgn2 is colored with light cyan. Close-up views of mutated and neighboring residues in human MDGA1 (left). Residues corresponding to patients' mutations described in the current study are presented as sticks and labeled. (F) Diagram illustrating the ASD-associated MDGA1 variants used in the current study. (G, H) Immunoblot analyses probing the expression levels of MDGA1 WT and the indicated variants in HEK293T cells (G) or cultured hippocampal neurons (H). (I, J) Analyses of the surface transport activity of HA-tagged MDGA1 WT and its variants in HEK293T cells. (I) Representative images of transfected HEK293T cells, illustrating the expression and surface transport activity of MDGA1 variants. Transfected cells were fixed (not permeabilized) and incubated with anti-HA antibody to detect the extracellular region of MDGA1 (red). Intracellular levels of MDGA1 constructs were visualized by incubating cells with an anti-HA antibody after permeabilization (green). Scale bar, 10 μm (applies to all images). (J) Quantification of the surface transport activity of MDGA1 constructs. All MDGA1 variants exhibited normal surface expression comparable to WT, indicating that the ASD-associated mutations did not impair surface transport. Data are means ± SEMs (nonparametric Kruskal–Wallis test with Dunn's *post hoc* test; n = 10 images/group). (K, L) Cell-surface-binding assays. (K) Representative images showing HEK293T cells expressing N-terminally HA-tagged MDGA1 WT and its variants incubated with purified Ig-fused neuroligin-2 (Ig-Nlgn2), Ig-fused APP (Ig-APP), or IgC alone (control), as analyzed by immunofluorescence imaging for Ig-fusion proteins (red) and HA (blue). Scale bar, 10 μm (applies to all images). (L) Quantification of cell-surface binding. All MDGA1 variants displayed comparable binding to Nlgn2 and APP. Data are means ± SEMs (n = 12–18 cells/group). Source data are available online for this figure.

using a previously published single-nucleus atlas (Appendix Fig. S2A,B) (Data ref: Wang et al, 2025), which demonstrated a consistently high level of MDGA1 in excitatory neurons (Appendix Fig. S2C–F). Specifically, MDGA1 exhibited high-level expression in immature intratelencephalic excitatory neurons (EN-IT-Immature) and layer II/III intratelencephalic excitatory neurons (EN-L2 3-IT) (Appendix Fig. S2G). Thus, *MDGA1* expression exhibits cellular and temporal specificity in excitatory neurons during the early development of the human cortex.

Since risk genes for ASD and neurodevelopmental disorders (NDDs) are enriched during early cortical neurodevelopment (Choi and An, 2021; Data ref: Satterstrom et al, 2020), we performed gene set enrichment tests for neurological genes among the cluster-specific genes. We found that the C12-specific genes (n = 1632) were significantly enriched for ASD and developmental delay risk genes (False Discovery Rate [FDR] < 0.05; Fisher's exact test). Additionally, C12 was robustly associated with bipolar risk genes and epilepsy risk genes ($Log_2(OR) > 1$, $P < 0.05$). These observations collectively suggest that MDGA1 may play a role across a spectrum of neuropsychiatric conditions with a shared genetic underpinning (Fig. EV1F; Dataset EV1). Notably, many of the C12-specific genes were previously reported by Satterstrom et al (Data ref: Satterstrom et al, 2020) as being neuronal communication genes involved in ASD risk (11/24), suggesting that *MDGA1* expressed in immature neurons is relevant to early synaptogenesis.

We compared the cell-type-specific *MDGA1* expression between ASD patients and control individuals using a single-nucleus atlas composed of 15 ASD patients and 16 controls (Data ref: Velmeshev et al, 2019) (Appendix Fig. S3A–C). We found that layer II/III excitatory neurons (L2/3) and protoplasmic astrocytes (AST-PP) from ASD individuals showed significantly decreased MDGA1 expression levels (FDR < 0.01) (Appendix Fig. S3D; Dataset EV2) compared to those from control subjects. These findings suggest that MDGA1 should be examined in a cell-type-specific manner

and that dysregulated MDGA1 expression in a specific cell population may contribute to the pathophysiology of ASD.

## Normal expression level, surface transport, ligand-binding, and folding of MDGA1 in the *MDGA1* mutations of P1 and P2

To investigate the structural effect of the patient mutations on human MDGA1, we predicted the structure of hMDGA1 using AlphaFold2 (Jumper et al, 2021; Tunyasuvunakool et al, 2021). The p.Val116Met mutation is in the Ig1 domain and two mutations (p.Tyr635Cys and p.Ala688Val) are in the fibronectin type III (FNIII) domain of MDGA1 (Fig. 1E). Human MDGA1 and neuroligin-2 (Nlgn2) form a hetero-tetrameric complex with 2:2 stoichiometry, and long-range electrostatic interactions and hydrogen bonds enable the strong binding of MDGA1 to the Nlgn2 dimer (Kim et al, 2017). The substitution of the small, nonpolar valine with the large methionine (p.Val116Met) in MDGA1, located at the binding interface, is likely to disrupt the interactions between MDGA1 immunoglobulin domain 1 (Ig1) and Nlgn2 by causing steric hindrance with neighboring polar residues (R243 and E247); notably, this mutation was predicted to have little effect on the 3D structure of MDGA1 (Fig. 1E). We were unable to predict the structural alterations in the p.Glu756Gln mutation located in the MDGA1 MAM domain, the structure of which was not determined by X-ray crystallography (Elegheert et al, 2017). The crystal structure of chicken MDGA1 ($Ig_{1-6}$-FNIII) showed that MDGA1 has a triangular shape and the sharp-angled $Ig_2$-$Ig_3$, $Ig_3$-$Ig_5$ and $Ig_6$-FNIII linkers are stabilized by multiple inter-domain contacts (Elegheert et al, 2017). hMDGA1 p.Tyr635 corresponding to chicken MDGA1 p.Tyr629 is located at the interface between Ig6 and FNIII (Appendix Fig. S4). Therefore, we hypothesized that the hMDGA1 p.Tyr635Cys mutation could structurally weaken these inter-domain interactions, which is also supported by free energy calculations (Table EV2), leading to the disruption of the triangular

structure of hMDGA1. We performed negative-stain electron microscopy and confirmed the previous structural data (Elegheert et al, 2017) that hMDGA1 WT has a closed triangular shape (boxed in blue color), while the majority of particles for thehMDGA1 Y635C/E756Q mutant had a linear or irregular structure (boxed in red color) (Appendix Fig. S5A). It is likely that the hMDGA1 p.Ala688Val mutation in the FNIII domain might mildly affect the protein stability of the FNIII domain. We next examined whether the ASD-linked MDGA1 mutants exhibit altered secondary structures, and a circular dichroism (CD) spectrum was measured. A strong negative peak near 210 nm of the CD spectra for MDGA1 wild-type (WT) suggests that the main secondary structure of MDGA1 is composed of β-sheets. The CD spectra of the MDGA1 V116M/A688V and MDGA1 Y635C/E756Q variants are similar to that of MDGA1 WT (Appendix Fig. S5B), indicating that the ASD-linked MDGA1 mutations do not alter the secondary structure of MDGA1.

To further probe the impact of amino acid substitutions on the structure and/or synaptic function of MDGA1, we generated mammalian expression vectors encoding the indicated MDGA1 variants (V116M/A688V and Y635C/E756Q), both of which are considered potentially damaging by various *silico* prediction tools (Table EV1). Notably, Y635 and E756 residues in human MDGA1 are conserved in human MDGA2 at equivalent positions. The aforementioned MDGA1 residues (except for A688) are mostly conserved among various vertebrate species, implying their possible functional significance (Appendix Fig. S4). To compare the expression levels and intracellular transport properties of the MDGA1 variants versus MDGA1 WT, we expressed HA-tagged variants and WT MDGA1 in human embryonic kidney 293T (HEK293T) cells (Fig. 1F). Immunoblot analyses of the HEK293T cell lysates (Fig. 1G) or the cultured hippocampal neuron lysates (Fig. 1H) expressing HA-tagged MDGA1 WT or the respective variants showed that total protein expression levels of MDGA1 V116M/A688V or MDGA1 Y635C/E756Q were comparable to those of MDGA1 WT (Fig. 1F–H). We next examined the surface and intracellular protein levels of MDGA1 WT and the mutants in HEK293T cells. MDGA1 V116M/A688V or MDGA1 Y635C/E756Q exhibited similar surface levels as MDGA1 WT (Fig. 1I,J). To determine whether the indicated MDGA1 mutations affected the interactions with known extracellular ligands (Nlgn2 or APP), we assayed cell-surface binding of recombinant Ig-fusion proteins of MDGA1 (IgC-MDGA1) or IgC alone (negative control) with HEK293T cells expressing HA-tagged Nlgn2 or APP (Fig. 1K,L). IgC-MDGA1 proteins robustly bound to HEK293T cells expressing HA-Nlgn2 or HA-APP (Fig. 1K,L). IgC did not exhibit any noticeable binding activity to HEK293T cells (Fig. 1K,L). We were unable to analyze whether the MDGA1 mutations could alter the synaptic localization of MDGA1 because overexpression of MDGA1 WT and variants at mature-stage cultured hippocampal neurons showed a diffuse distribution along the dendrites of the transfected neurons, in line with prior observations (Kim et al, 2024a; Toledo et al, 2022).

### Overexpression of MDGA1 Y635C/E756Q abolished the MDGA1 activity in negatively regulating GABAergic synapse maintenance in cultured hippocampal neurons

Given previous reports that MDGA1 overexpression specifically reduces the number of GABAergic synapses in cultured hippocampal neurons and hippocampal CA1 pyramidal neurons (Kim

et al, 2022a; Lee et al, 2013; Pettem et al, 2013), we explored whether the ASD-associated MDGA1 mutations could impair the ability of MDGA1 to negatively regulate the GABAergic synapse maintenance. To this end, we transfected the indicated MDGA1 variants at 7 days in vitro (DIV7) in cultured hippocampal neurons and measured the density of puncta positive for both gephyrin (a marker for GABAergic postsynaptic specialization) and vesicular GABA transporter (VGAT; a marker for presynaptic GABAergic nerve terminals) in the transfected neurons at DIV14 (Fig. 2A,B). In line with previous reports (Lee et al, 2013; Pettem et al, 2013), overexpression of MDGA1 WT significantly decreased the density and area of gephyrin$^+$VGAT$^+$ puncta. Interestingly, overexpressed MDGA1 V116M/A688V had similar effects, whereas overexpressed MDGA1 Y635C/E756Q failed to alter GABAergic synapse puncta density or area (Fig. 2A,B). However, both ASD-associated MDGA1 variants effectively suppressed Nlgn2-induced presynaptic assembly in heterologous synapse formation assays, in a manner similar to MDGA1 WT (Appendix Fig. S6). Further experiments showed that the E756Q substitution, but not the Y635C substitution, was responsible for abrogating the activity of MDGA1 WT in negatively regulating GABAergic synapse number in cultured neurons (Appendix Fig. S7). Since MDGA2 overexpression also increases excitatory synapses (Kim et al, 2024a), we questioned whether overexpression of MDGA2 Y700C/E820Q would also fail to suppress excitatory synapse density. However, overexpressed MDGA2 Y700C/E820Q displayed similar abilities to suppress excitatory synapses (Appendix Fig. S8A,B).

To test whether the ASD-associated MDGA1 mutations could also affect the MDGA1-mediated negative regulation of GABAergic synaptic transmission, we measured miniature inhibitory postsynaptic currents (mIPSCs) using whole-cell voltage-clamp recordings. Overexpression of MDGA1 WT significantly decreased the mIPSC frequency without altering the mIPSC amplitude, rise time or decay time of mIPSCs, compared to those from neurons expressing the control plasmid (Fig. 2C–G). Overexpressed MDGA1 Y635C/E756Q had a similar effect, whereas overexpressed MDGA1 V116M/A688V did not alter GABAergic synaptic transmission (Fig. 2C–G). However, overexpression of MDGA2 Y700C/E820Q significantly decreased the frequency of miniature excitatory postsynaptic currents (mEPSCs), in keeping with the two-photon imaging results (Appendix Fig. S8C–G). Taken together, these data suggest that the MDGA1 Y635C/E756Q substitutions manifested as MDGA1 LoF mutations in the regulation of GABAergic synaptic properties.

### Overexpressed MDGA1 Y635C/E756Q fails to negatively regulate GABAergic synaptic properties in hippocampal CA1 pyramidal neurons

We recently showed that overexpression of MDGA1 in the adult mouse hippocampal CA1 suppresses GABAergic synaptic transmission and stabilization (Kim et al, 2022a). To determine whether our ASD-associated MDGA1 mutations could affect the ability of MDGA1 to negatively regulate the GABAergic synaptic properties, we injected adeno-associated viruses (AAVs) expressing MDGA1 WT or the indicated MDGA1 variants into the hippocampal CA1 of adult mice and performed whole-cell patch-clamp recordings to measure the GABAergic synaptic transmission and strength. Consistent with our previous report (Kim et al, 2022a),

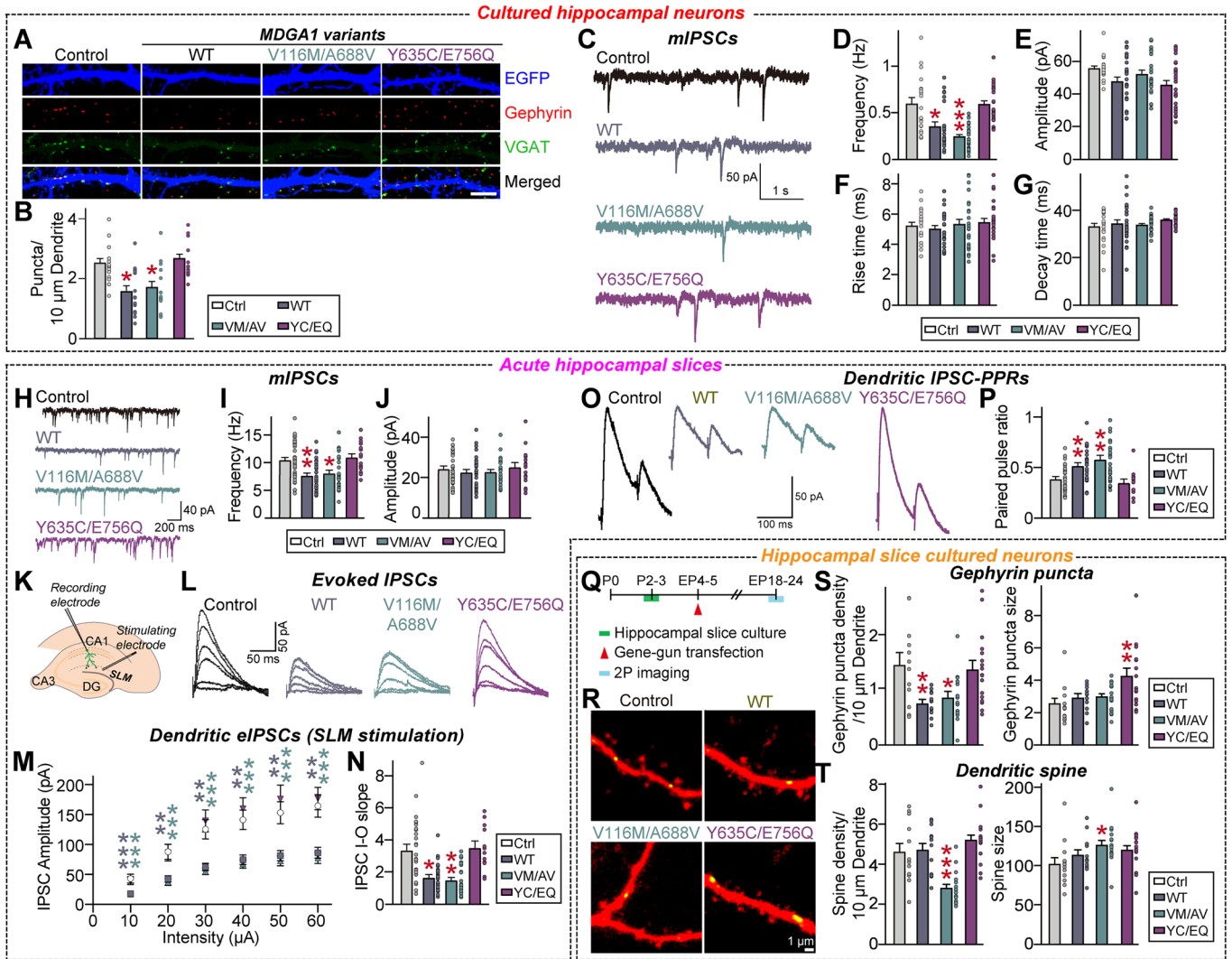

**Figure 2.  Analysis of GABAergic synaptic properties in hippocampal CA1 pyramidal neurons expressing ASD-associated MDGA1 variants.**

(A, B) Representative images (A) and summary graphs (B) showing the density of synaptic puncta in cultured hippocampal neurons transfected at DIV7 with the indicated full-length MDGA1 expression constructs and immunostained at DIV14 with antibodies to VGAT, gephyrin and EGFP. Data are presented as means ± SEMs ($n = 14$–15 neurons/group; *$P = 0.0135$ (vs. WT), *$p = 0.0451$ (vs. V116M/A688V); Kruskal–Wallis test with Dunn's *post hoc* test). Scale bar, 10 μm (applies to all images). (C–G) Representative mIPSC traces (C) and quantification of frequency (D), amplitude (E), rise time (F), and decay time (G) of mIPSCs from cultured neurons transfected with MDGA1 WT or its variants. Data are presented as means ± SEMs (control, $n = 19$; WT, $n = 22$; V116M/A688V, $n = 25$; Y635C/E756Q, $n = 24$; *$P = 0.0226$ (vs. WT), ***$P = 0.0001$ (vs. V116M/A688V); Kruskal–Wallis test with Dunn's *post hoc* test). (H–J) Representative traces (H) of electrophysiological recordings in CA1 pyramidal neurons expressing MDGA1 WT and its variants recorded in 7- to 8-week-old mice, 3 weeks after AAV-mediated expression of the indicated MDGA1 variants in CA1 neurons. Quantification of mIPSC frequency (I) and amplitude (J) recorded in the hippocampal CA1 pyramidal neurons. Data are presented as means ± SEMs ($n = 16$–37 cells/group; *$P = 0.0033$ (vs. WT), *$P = 0.0280$ (vs. V116M/A688V); Kruskal–Wallis test with Dunn's *post hoc* test). (K–N) Experimental setup for recording eIPSCs in hippocampal CA1 pyramidal neurons by placing stimulating electrodes in the *stratum lacunosum-moleculare* (SLM) layer. Representative traces (L) of eIPSCs obtained from dendritic recordings in CA1 pyramidal neurons. Average eIPSC I-O curve (M), and average eIPSC I-O slope (N) from hippocampal CA1 pyramidal neurons expressing the indicated MDGA1 variants. Data are presented as means ± SEMs ($n = 12$–27 cells/group; (M) ***$P = 0.0004$ (10 μA), **$P = 0.0134$ (20 μA), **$P = 0.0091$ (30 μA), **$P = 0.0065$ (40 μA), **$P = 0.0101$ (50 μA), **$P = 0.0086$ (60 μA) vs. WT; ***$P = 0.0002$ (10 μA), ***$P = 0.0003$ (20 μA), ***$P = 0.0003$ (30 μA), ***$P = 0.0010$ (40 μA), ***$P = 0.0009$ (50 μA), ***$P = 0.0007$ (60 μA) vs. VM/AV; (N) *$P = 0.0445$ (vs. WT), **$P = 0.0052$ (vs. VM/AV); Kruskal–Wallis test with Dunn's *post hoc* test). (O, P) Representative somatic eIPSC paired-pulse ratio (PPR) traces (O) and average PPR (P) from hippocampal CA1 pyramidal neurons expressing the indicated MDGA1 variants. Data are presented as means ± SEMs ($n = 12$–27/group; (P) **$P = 0.0016$ (vs. WT), **$P = 0.0001$ (vs. VM/AV); Kruskal–Wallis test with Dunn's *post hoc* test). (Q) Schematic showing biolistic transfection of organotypic hippocampal CA1 slices with MDGA1 mutants at EP4–5 and imaging at EP18–24. (R) Two-photon images of dendritic segments from CA1 pyramidal neurons co-transfected with tdTomato, gephyrin intrabody-GFP, and MDGA1 control, MDGA1 WT, MDGA1 V116M/A688V, or MDGA1 Y635C/E756Q vector. (S, T) Quantitative analysis of gephyrin puncta density and size (S) (puncta/10-μm dendrite) and dendritic spine density and size (T) (spines/10-μm dendrite) in CA1 pyramidal neurons expressing MDGA1 WT or its variants. Data are presented as means ± SEMs ($n = 12$–17 cells/group; (S) **$P = 0.0088$ (puncta density; vs. WT), **$P = 0.0127$ (puncta density; vs. VM/AV), **$P = 0.0019$ (puncta size; vs. YC/EQ); (T) ***$P = 0.0005$ (spine density; vs. VM/AV), *$P = 0.0289$ (spine size; vs. VM/AV); Kruskal–Wallis test with Dunn's *post hoc* test). Source data are available online for this figure.

overexpression of MDGA1 WT significantly decreased the frequency (but not amplitude) of mIPSCs in the hippocampal CA1 pyramidal neurons (Fig. 2H–J). Overexpression of MDGA1 V116M/A688V had a similar effect, whereas that of MDGA1 Y635C/E756Q failed to decrease the mIPSC frequency (Fig. 2H–J). To examine whether MDGA1 has a similar impact on GABAergic synaptic transmission in the medial prefrontal cortex (mPFC), we measured mIPSCs in layer II/III of mPFC pyramidal neurons. Surprisingly, overexpression of MDGA1 WT and its ASD-associated variants did not alter the frequency or amplitude of mIPSCs nor the densities of gephyrin puncta or dendritic spines (Appendix Fig. S9).

We next examined whether the ASD-associated MDGA1 mutations impacted the ability of MDGA1 to suppress evoked inhibitory postsynaptic currents (eIPSCs) in the distal dendritic compartment of hippocampal CA1 pyramidal neurons (Kim et al, 2022a). We found that overexpression of MDGA1 WT or MDGA1 V116M/A688V significantly decreased the amplitude of distal dendritic eIPSCs and increased the paired-pulse ratio (PPR), whereas MDGA1 Y635C/E756Q failed to do so (Fig. 2K–P). In contrast, overexpression of MDGA1 WT, V116M/A688V, or Y635C/E756Q had no effect on the frequency or amplitude of asynchronous IPSCs (aIPSCs) (Appendix Fig. S10).

To corroborate our electrophysiological observations, we used two-photon laser-scanning microscopy to examine gephyrin puncta in hippocampal CA1 slice cultures (Fig. 2Q). Our results revealed that overexpression of MDGA1 WT or MDGA1 V116M/A688V significantly decreased the gephyrin puncta density, whereas overexpression of MDGA1 Y635C/E756Q did not; in fact, overexpression of the latter variant tended to increase the gephyrin puncta size (Fig. 2R,S). Unexpectedly, overexpression of MDGA1 V116M/A688V, but not MDGA1 WT or MDGA1 Y635C/E756Q, significantly reduced the density of dendritic spines while concomitantly increasing their size (Fig. 2R,T). These results were further reinforced by data obtained from cultured hippocampal neurons, wherein overexpression of MDGA1 V116M/A688V, but not MDGA1 WT or MDGA1 Y635C/E756Q, specifically decreased the excitatory synapse density and mEPSC frequency (Appendix Fig. S11). Our findings collectively indicate that the Y635C/E756Q substitution renders MDGA1 unable to control GABAergic synapse organization in the hippocampal CA1, whereas the MDGA1 V116M/A688V substitution affects excitatory synaptic structures.

## In vivo overexpression of MDGA1 V116M/A688V, but not MDGA1 Y635C/E756Q, alters cortical neuron migration and ultrasonic vocalizations in mice

In patients with ASD, the marked neuropathological characteristics include defective neuronal migration (Pan et al, 2019). Intriguingly, Mdga1 knockdown in migrating layer II/III cortical neurons can disrupt their normal migration (Takeuchi and O'Leary, 2006). Moreover, transient glutamatergic synapse formation between subplate neurons and immature migratory neurons instructs neocortical neuronal migration (Ohtaka-Maruyama et al, 2018), reminiscent of the excitatory synaptic phenotype of MDGA1 V116M/A688V (see Fig. 2; Appendix Fig. S11). To investigate whether our ASD-associated MDGA1 variants could alter cortical neuron migration, we performed in utero electroporation

experiments. Expression plasmids encoding HA-tagged MDGA1 (WT, V116M/A688V or Y635C/E756Q) or a control plasmid (pCIG2 harboring an internal ribosome entry site-driven enhanced green fluorescent protein [EGFP] sequence) were electroporated into mice at E15.5 to mainly target progenitors of layer II/III cortical pyramidal neurons, but not GABAergic interneurons (Fig. EV2A), as previously described (Meyer-Dilhet and Courchet, 2020). At P21, neuronal positions in the electroporated mice were examined by EGFP fluorescence (Fig. EV2A). Neurons expressing MDGA1 WT or MDGA1 Y635C/E756Q exhibited patterns of migration across cortical layers similar to those of neurons expressing the control plasmid (> 95% of neurons reached layers II/III) (Fig. EV2B,C). Strikingly, neurons expressing MDGA1 V116M/A688V displayed abnormally altered migration: only ~55% of the transfected neurons were seen in layer II/III, and a large portion was detected in layer V/VI/white matter (~44.5%) (Fig. EV2B,C). These results suggest that the V116M/A688V substitution might have a dominant-negative inhibitory effect on MDGA1 function, leading to abnormal cortical neuron migration.

In view of the observation that P1 and P2 patients harboring the MDGA1 missense mutations commonly exhibit communication abnormalities, we evaluated maternal isolation-induced USVs, which are distress calls emitted by pups when separated from their mother (representing infant-mother vocal communicative behavior relevant to ASD) (Kazdoba et al, 2016; Premoli et al, 2021). In utero electroporation was performed for control, MDGA1 WT, MDGA1 V116M/A688V, or MDGA1 Y635C/E756Q vectors, and maternal isolation-induced USVs of pups were recorded on postnatal day 3 (P3), P6, P9, and P12 (Fig. EV2D,E). Call numbers in all pups increased from P3 to P6, decreased slightly by P9, and reached zero by P12 (Fig. EV2E). Intriguingly, MDGA1 V116M/A688V-expressing mice emitted significantly fewer USVs compared to age-matched pups expressing control, MDGA1 WT, or MDGA1 Y635C/E756Q at all ages (up to P12) (Fig. EV2E).

## Mdga1-cKO mice exhibit various autistic-like abnormal behaviors

Given that the overexpression of ASD-associated MDGA1 variants revealed functionally divergent outcomes depending on the mutation, we next investigated the effects of the complete loss of MDGA1 in forebrain excitatory neurons. To this end, we utilized Emx1-Cre::Mdga1^(f/f) mice (also denoted as Mdga1-cKO mice) as a conditional knockout model. Since MDGA1 is highly enriched in excitatory neurons of the cortex and hippocampus in both mice and humans (Lee et al, 2013; see Appendix Fig. S2), this excitatory neuron-specific deletion model was selected for downstream behavioral and molecular phenotyping, as employed in our previous study (Kim et al, 2022b; Han et al, 2024). Notably, previous studies reported that constitutive Mdga1-KO mice exhibit impaired hippocampus-dependent spatial learning and memory, along with a reduced startle response—phenotypes often regarded as schizophrenia-related endophenotypes (Connor et al, 2017; Hossain et al, 2020; Turetsky et al, 2007).

Regarding other autistic-like behavioral phenotypes, adult male (P60) Mdga1-cKO mice exhibited levels of anxiety/exploration-related behavior, locomotor activity, working memory, and repetitive/compulsive-like behavior within the expected ranges

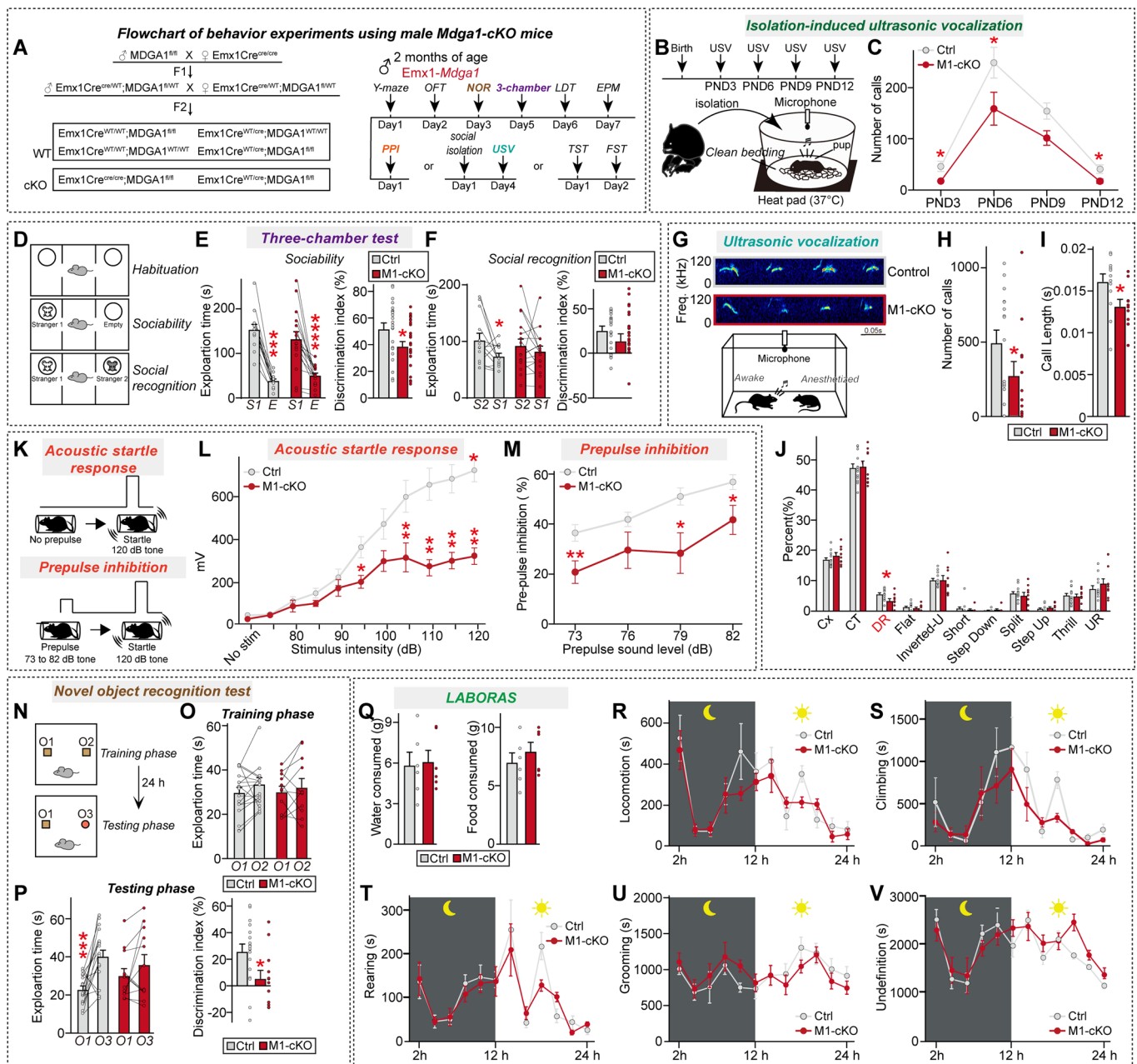

(Fig. 3A; Appendix Fig. S12). However, adult male *Mdga1*-cKO mice exhibited deficits in social behavior, as evident from their reduced time spent exploring animated mice over an object (sociability) and reduced time spent exploring unfamiliar mice (social novelty recognition) in the three-chamber test (Fig. 3D–F). These mice also emitted reduced USVs upon encountering a female mouse, as well as showing a shift in the distribution of syllable types in the USVs (Fig. 3G–J). Notably, even at the early postnatal stage, *Mdga1*-cKO pups exhibited decreased maternal isolation-induced USVs, indicating impairments in early communicative behavior (Fig. 3B,C). Impaired sensorimotor gating was observed in adult male *Mdga1*-cKO mice, as characterized by decreases in the acoustic startle response and prepulse inhibition (Fig. 3K–M), as

well as decreased novel object-recognition memory (Fig. 3N–P). The home-cage activities of these mice were unaltered (Fig. 3Q–V).

Given that there is a strong male bias in human ASDs (Bolte et al, 2023) and many ASD animal models bear sexually dimorphic behavioral abnormalities (Bruce et al, 2021; Werling, 2016), we further tested whether adult female *Mdga1*-cKO mice displayed similar autistic-like behavioral phenotypes (Fig. EV3). Indeed, adult female *Mdga1*-cKO mice exhibited normal sociability, USV numbers, USV syllable compositions, sensorimotor gating, and novel object-recognition memory (Fig. EV3). Also, unlike their male counterparts, adult female *Mdga1*-cKO mice showed reduced anxiety-like behavior (Fig. EV3D,E). These results suggest that MDGA1 could be a potential sex-specific biomarker for ASDs.

**Figure 3. Analysis of mouse behavior in male adult *Mdga1*-cKO mice.**

(A) Breeding scheme followed to generate *Mdga1*-cKO mice. *Emx1*-Cre driver mice were crossed with *Mdga1*-floxed mice. The timeline of behavioral experiments performed on male *Mdga1*-cKO mice, including Y-maze, open field test (OFT), novel object recognition (NOR), three-chamber, light-dark transition (LDT), elevated plus maze (EPM), tail suspension test (TST), forced swim test (FST), social isolation-induced ultrasonic vocalization (USV), and prepulse inhibition (PPI) tests. (B) Experimental setup for recording USVs from isolated pups. Pups were separated from their dams and recorded for 5 min on clean bedding with a heat pad (37 °C) using a condenser microphone. (C) Quantification of USV call numbers from control and *Mdga1*-cKO pups at PND3, PND6, PND9 and PND12. Data are presented as means ± SEMs ($n = 7$–9 pups/group from ≥ 3 litters; *$P = 0.0110$ (PND3), *$P = 0.0278$ (PND6), *$P = 0.0187$ (PND12); Mann–Whitney $U$ test). (D–F) Three-chamber social interaction test design (D), and quantification of sociability (E) and social recognition (F). In the sociability phase (E), *Mdga1*-cKO mice exhibited a significantly reduced discrimination index compared to controls, indicating impairment of social preference. In the recognition phase (F), *Mdga1*-cKO mice failed to prefer a novel stranger over a familiar one, suggesting deficits in social recognition. Data are means ± SEMs ($n = 13$–16 mice/group; (E) ***$P = 0.0002$ (Ctrl/sociability), ****$P < 0.0001$ (*Mdga1*-cKO/ sociability), *$P = 0.0151$ (Discrimination index); (F) *$P = 0.0398$ (Ctrl/social recognition); Wilcoxon matched-pairs signed rank test). (G–J) Representative USV traces (G), quantification of USV call number (H) and duration (I), and distribution of call types (J; complex trill, downward ramp, flat, inverted-U, short, split, step down, step up, trill, upward ramp) in adult male mice recorded under social isolation conditions. *Mdga1*-cKO mice emitted fewer and shorter calls, with the altered proportions of call types including reductions in downward ramp calls. Data are means ± SEMs ($n = 12$–15 mice/group; (H) *$P = 0.0414$; (I) *$P = 0.0155$; (J) *$P = 0.0355$; Mann–Whitney $U$ test). (K–M) Acoustic startle response and prepulse inhibition (PPI) test. In the PPI paradigm (K), mice were presented with a prepulse (73–82 dB) followed by a 120-dB startle pulse. *Mdga1*-cKO mice displayed reduced startle response (L) and reduced PPI at multiple prepulse intensities (M), suggesting the presence of sensorimotor gating deficits. Data are means ± SEMs ($n = 12$–18 mice/group; (L) *$P = 0.0275$ (95 dB), **$P = 0.0091$ (105 dB), **$P = 0.0013$ (110 dB), ***$P = 0.0007$ (115 dB), ***$P = 0.0009$ (120 dB); (M) **$P = 0.0070$ (79 dB), *$P = 0.0442$ (82 dB); Mann–Whitney $U$ test). (N–P) NOR test schematic (N) showing the object configurations used during the training and test phases. *Mdga1*-cKO mice showed reductions in the discrimination index and exploration of the novel object (O, P), consistent with an impairment of recognition memory. Data are means ± SEMs ($n = 12$–18 mice/group; (P) ***$P = 0.0003$ (Exploration time), *$P = 0.0246$ (Discrimination index); Wilcoxon matched-pairs signed rank test). (Q–V) Quantification of spontaneous behaviors in home cages using the LABORAS system. Measurements of food and water intake (Q), locomotion (R), climbing (S), rearing (T), grooming (U), and undefined behaviors (V) were recorded over 24 h in 2-h bins. The light and dark cycles of the day are represented by sun and moon icons, respectively. Data are means ± SEMs ($n = 6$ mice/group). Source data are available online for this figure.

## Adult male *Mdga1*^Y636C/E751Q^ knockin mice exhibit autistic-like behavioral abnormalities that are distinct from those of adult male *Mdga1*-cKO mice

Our previous functional analyses revealed that the two ASD-associated MDGA1 substitutions induce abnormalities in distinct facets of neuronal and synaptic developmental processes. However, in utero overexpression of MDGA1 Y635C/E756Q did not significantly impair pup isolation-induced USVs or affect neuronal migration (Fig. EV2). We further investigated whether this substitution could influence behaviors in adult mice (Fig. 4A). To this end, we generated *Mdga1* knock-in (KI) mice carrying Y636C and E751Q (Tyr636 and Glu751 in mouse *Mdga1* correspond to Tyr635 and Glu756 in human MDGA1) using the clustered regularly interspaced short palindromic repeats (CRISPR)/ Cas9 system. Our results showed that *Mdga1*^Y636C/E751Q^ pups exhibited reduced numbers of USV calls throughout development (Fig. 4B,C), similar to *Mdga1*-cKO pups. In addition, homozygous *Mdga1*^Y636C/E751Q^ pups showed more severe impairment than their heterozygous counterparts in USV (Fig. 4B,C). This discrepancy may reflect a dosage-dependent effect of the mutant protein, as behavioral impairment was more pronounced in homozygous KI mice compared to heterozygotes, whereas in utero electroporation likely resulted in partial overexpression against a background of endogenous wild-type MDGA1, potentially masking the phenotype (Fig. EV2).

Next, we subjected adult male *Mdga1*^Y636C/E751Q^ KI mice to a battery of behavioral tests similar to those used for *Mdga1*-cKO mice. Both adult male *Mdga1*^Y636C/E751Q^ KI and *Mdga1*-cKO mice were viable and fertile, exhibited no obvious abnormalities, morbidity, or premature mortality, and were comparable to control mice in brain weight and size (Appendix Fig. S13). Male *Mdga1*^Y636C/E751Q^ KI mice showed normal anxiety/exploration-related behavior, locomotor activity, working memory, and repetitive/compulsive-like behavior (Appendix Fig. S14). In addition, male *Mdga1*^Y636C/E751Q^ KI mice exhibited normal sociability and

object recognition memory (Fig. 4D–F,N–P); however, similar to their male cKO counterparts, male *Mdga1*^Y636C/E751Q^ KI mice encountering female mice exhibited an altered syllable composition in their USV calls with an increased USV call number (Fig. 4G–J). Intriguingly, male *Mdga1*^Y636C/E751Q^ KI mice exhibited a decreased acoustic startle response (ASR) but increased prepulse inhibition (Fig. 4K–M) and thereby differed from the male *Mdga1*-cKO mice (Fig. 3). Overall, male *Mdga1*^Y636C/E751Q^ KI mice exhibited a narrower range of behavioral impairments than male *Mdga1*-cKO mice, particularly in sensorimotor gating and sociability, suggesting that *Mdga1*^Y636C/E751Q^ KI might not compromise all the MDGA1-mediated functions. Again, the female *Mdga1*^Y636C/E751Q^ KI mice did not exhibit any of the behavioral deficits seen in their male KI counterparts, further highlighting the sexually dimorphic manifestation of MDGA1 dysfunctions (Fig. EV4).

## Adult male *Mdga1*^Y636C/E751Q^ KI mice phenocopy the electrophysiological abnormalities of *Mdga1*-cKO mice

We additionally analyzed the levels of MDGA1 and other synaptic proteins in *Mdga1*^Y636C/E751Q^ KI and cKO mice. Quantitative RT-PCR analyses revealed that forebrain regions (hippocampus and cortex), but not the cerebellum, exhibited marked decreases in the mRNA levels of *Mdga1* with no change in those of *Mdga2* (Appendix Fig. S15A). Semi-quantitative immunoblotting experiments showed that *Mdga1*-cKO caused a significant loss of MDGA1 whereas the Y636C/E751Q substitution did not affect the protein level of MDGA1 (Appendix Fig. S15B,C). We also observed that the expression of GABA_ARγ2 and the vesicular GABA transporter (VGAT) was increased in cKO and KI mice, gephyrin levels were substantially increased in cKO mice, and Nlgn2 levels were increased in KI mice (Appendix Fig. S15B,C). We did not observe any significant change in the levels of the other examined proteins, including those characteristic of excitatory synapses. These data suggest that *Mdga1*^Y636C/E751Q^ KI and cKO do not cause global changes in the molecular composition of the brain, except for the

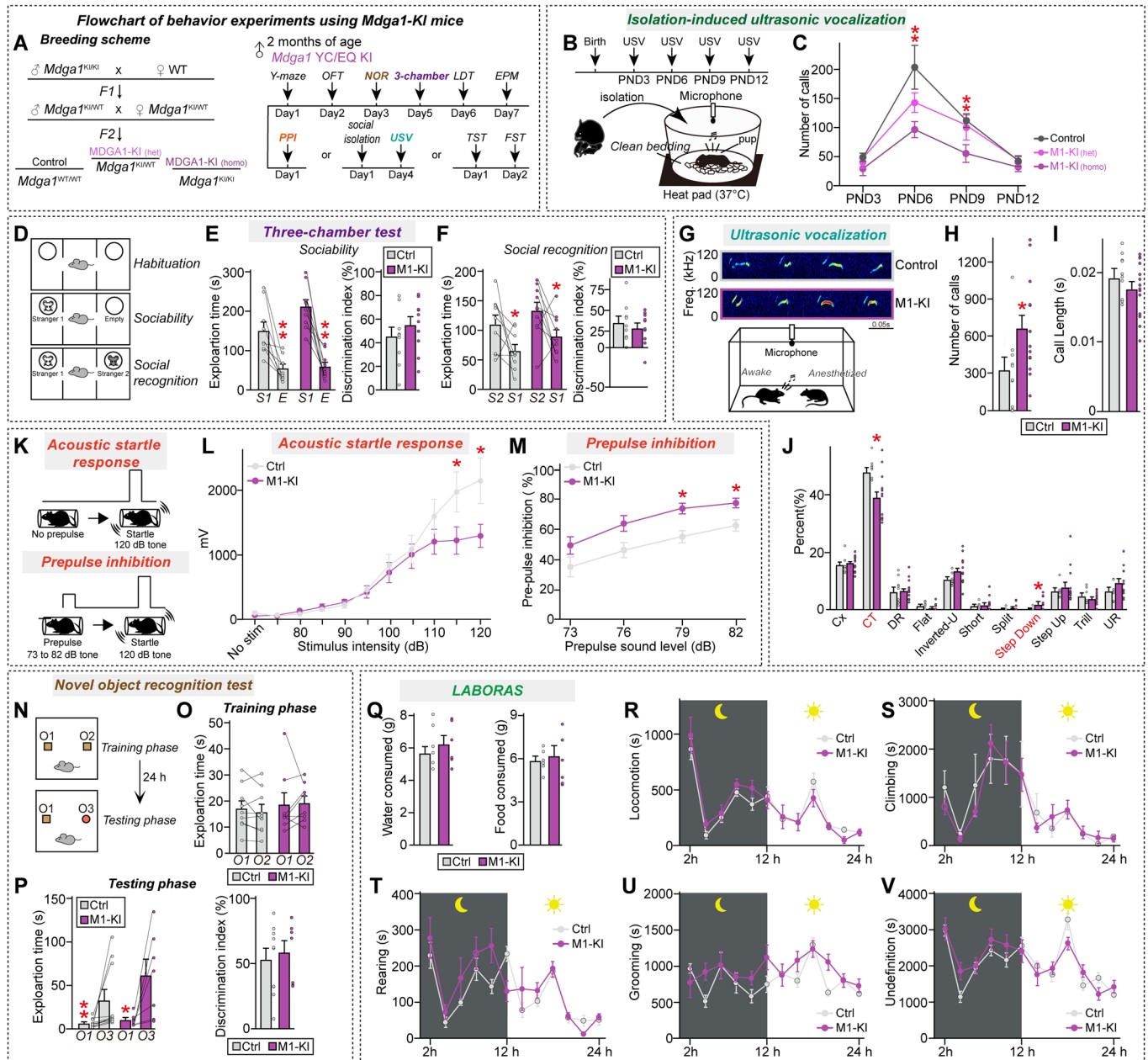

small increases seen in inhibitory synaptic markers in both KI and cKO mice.

We further performed immunohistochemical analyses using antibodies against VGAT and gephyrin (to label inhibitory synapses) or VGLUT1 and PSD-95 (to label excitatory synapses) in the hippocampus and mPFC of adult cKO and KI mice. An increased density of VGAT+ puncta was specifically observed in the *stratum lacunosum moleculare* layer of the hippocampus of male cKO and KI mice but not their female counterparts (Appendix Figs. S16–S23). No change was observed in either inhibitory or excitatory synapses in mPFC layer II/III of adult cKO and KI mice.

Next, we examined whether the *Mdga1*[Y636C/E751Q] variant affects the ability of endogenous MDGA1 to regulate synaptic transmission in hippocampal CA1 pyramidal neurons (Fig. 5A–C). Male KI

mice were subjected to electrophysiological recordings at various stages for measurement of spontaneous synaptic transmission, evoked synaptic strength, and neurotransmitter release probability. Since discrepancies in electrophysiological phenotypes were previously reported for *Mdga1*-KO mice at different stages of growth (Connor et al, 2017; Kim et al, 2022a), we included both juvenile (further subdivided into P10–12 or P18–20) and adult (P60) male *Mdga1*[Y636C/E751Q] mice for the study. Whole-cell patch-clamp recordings revealed no marked alterations in the frequency/amplitude of mIPSCs or mEPSCs in hippocampal CA1 neurons from both juvenile and adult male *Mdga1*[Y636C/E751Q] mice, except for P10–12 *Mdga1*[Y636C/E751Q] neurons, which exhibited increased frequencies of both mIPSCs and mEPSCs (Fig. 5D–O). Parallel recordings using male *Mdga1*-cKO mice showed similar

**Figure 4.   Analysis of mouse behavior in adult male $Mdga1^{Y636C/E751Q}$ KI mice.**

(A) Breeding scheme used to generate $Mdga1^{Y636C/E751Q}$ KI mice and timeline of behavioral testing. Behavioral assays, including the Y-maze, open field test (OFT), novel object recognition (NOR), three-chamber social test, light-dark transition (LDT), elevated plus maze (EPM), tail suspension test (TST), forced swim test (FST), social isolation-induced USV, and prepulse inhibition (PPI), were performed in adult male mice starting at 8 weeks of age. (B) Schematic of the setup used to record USVs in pups. The pups were separated from the dam and placed on clean bedding with a heating pad (37 °C). USVs were recorded for 5 min using a condenser microphone. (C) Number of USV calls recorded from $Mdga1^{Y636C/E751Q}$ KI mice and control pups at PND3, PND6, PND9, and PND12. KI pups emitted significantly fewer USVs at PND6 and PND9, suggesting the presence of early-life communication deficits. Data are means ± SEMs ($n = 9$–13 pups/group; *$P = 0.0278$ (PND6), **$P = 0.0023$ (PND9); Mann–Whitney $U$ test). (D–F) Schematic of the three-chamber social interaction assay (D), and quantification of sociability (E) and social recognition (F). Data are means ± SEMs ($n = 9$–10 mice/group; (E) **$P = 0.0039$ (Ctrl/sociability), **$P = 0.0020$ (M1-KI/sociability); (F) *$P = 0.0391$ (Ctrl/social recognition), *$P = 0.0371$ (M1-KI/social recognition); Wilcoxon matched-pairs signed rank test). (G–J) Representative USV traces (G), quantification of USV call number (H) and duration (I), and distribution of call types (J; complex trill, downward ramp, flat, inverted-U, short, split, step down, step up, trill, upward ramp) in adult male mice recorded under social isolation conditions. $Mdga1^{Y636C/E751Q}$ KI mice emitted a significantly greater number of USV calls compared to controls, but with altered call-type composition—specifically, a decrease in complex trill calls and an increase in step-down calls—suggesting that there is a shift in the qualitative nature of vocal communication. Data are means ± SEMs ($n = 12$–15 mice/group; (H) *$P = 0.0220$; (J) *$P = 0.0327$ (CT), *$P = 0.0377$ (Step Down); Mann–Whitney $U$ test). (K–M) Schematic of the PPI paradigm (K). Mice were presented with a prepulse (73–82 dB) followed by a 120-dB startle pulse. $Mdga1^{Y636C/E751Q}$ KI mice exhibited reduced startle responses (L) and significantly enhanced PPI at higher prepulse intensities (M), indicating the presence of sensorimotor gating deficits. Data are means ± SEMs ($n = 12$–13 mice/group; (L) *$P = 0.0156$ (115 dB), *$P = 0.0156$ (120 dB); (M) *$P = 0.0206$ (79 dB), *$P = 0.0300$ (82 dB); Mann–Whitney $U$ test). (N–P) Schematic of the NOR test (N), showing object placements during the training and testing phases. Testing occurred 24 h after the training session. $Mdga1^{Y636C/E751Q}$ KI mice exhibited exploration times and discrimination indices comparable to those of control mice (O, P), indicating that they had intact recognition memory. Data are means ± SEMs ($n = 7$–10 mice/group; (P) **$P = 0.0020$ (Ctrl), *$P = 0.0156$ (M1-KI); Wilcoxon matched-pairs signed rank test). (Q–V) Quantification of spontaneous behaviors in home cages using the LABORAS system. Food and water intake (Q), locomotion (R), climbing (S), rearing (T), grooming (U), and undefined behaviors (V) were recorded over 24 h in 2-h bins. The light and dark cycles of the day are represented by sun and moon icons, respectively. Data are means ± SEMs ($n = 6$ mice/group). Source data are available online for this figure.

phenotypes aligned with those from the same-staged male $Mdga1^{Y636C/E751Q}$ mice; specifically, an increased frequency of mIPSCs and mEPSCs in P10–12 CA1 pyramidal neurons with no changes in spontaneous synaptic transmission in P18–20 or P60 CA1 pyramidal neurons (Fig. EV5).

Hippocampal CA1-specific $Mdga1$-cKO mice exhibit increased dendritic IPSC amplitudes with increased paired-pulse ratios (PPR) specifically when the distal dendrites are stimulated (Kim et al, 2022a). Accordingly, we examined whether these electrophysiological features could be replicated in male $Mdga1^{Y636C/E751Q}$ CA1 pyramidal neurons. Both P18–20 and P60 CA1 pyramidal neurons from male $Mdga1^{Y636C/E751Q}$ KI mice exhibited a marked increase in eIPSC amplitudes and IPSC-PPRs (Fig. 5P–Y). Parallel measurements from male $Mdga1$-cKO mice showed identical phenotypes (Fig. EV5), reinforcing the LoF effects of the Y636C/E751Q substitution. Unexpectedly, we observed a significant increase in dendritic aIPSC amplitudes in CA1 pyramidal neurons from male $Mdga1^{Y636C/E751Q}$ KI and $Mdga1$-cKO mice (Appendix Fig. S24). However, measurement of somatic aIPSCs revealed a specific increase in male $Mdga1^{Y636C/E751Q}$ KI mice, but not in male $Mdga1$-cKO mice (Appendix Fig. S24E–H). Strikingly, CA1 pyramidal neurons from both adult female $Mdga1^{Y636C/E751Q}$ KI and $Mdga1$-cKO mice displayed no alterations in eIPSC amplitudes or IPSC-PPRs compared to control neurons (Fig. EV6A–J). In contrast, CA1 neurons from juvenile female $Mdga1^{Y636C/E751Q}$ KI and $Mdga1$-cKO mice showed increased eIPSC amplitudes and IPSC-PPRs (Fig. EV6K–T), suggesting that sex-specific changes in GABAergic synaptic strength reflect developmental effects. These results are in keeping with the lack of detectable impairment of certain social behaviors in adult female $Mdga1^{Y636C/E751Q}$ KI and $Mdga1$-cKO mice (see Figs. EV3 and EV4).

Numerous neuropathological studies have revealed lower interneuron density in postmortem tissues from individuals with ASDs and hypoactive parvalbumin (PV)$^+$ interneurons in various ASD animal models (Contractor et al, 2021; Filice et al, 2020). Based on these findings, we examined the density of PV$^+$ interneurons and neuronal activity across hippocampal subfields

in our animal models (Appendix Fig. S25). No changes in interneuron density or neuronal activity (assessed via c-Fos immunostaining) were evident in either $Mdga1^{Y636C/E751Q}$ KI or $Mdga1$-cKO mice. Moreover, no changes in the density of neurons, astrocytes or microglia were observed in male adult $Mdga1^{Y636C/E751Q}$ KI mice (Appendix Fig. S26). Given the previous report that MDGA1 is robustly expressed in the upper cortical layer neurons (Takeuchi and O'Leary, 2006), we examined whether $Mdga1^{Y636C/E751Q}$ KI mice exhibited altered targeting of cortical layer II/III neurons, as seen in $Mdga1$-cKO mice (see Fig. EV2).

Immunohistochemical analyses revealed that the density of CB$^+$ and Brn2$^+$ neurons in somatosensory cortex layer II/III was reduced in $Mdga1$-cKO mice (Appendix Fig. S27), whereas no such changes were observed in adult male $Mdga1^{Y636C/E751Q}$ KI mice when assessed using the cortical layer II/III-specific markers, Wfs1, calbindin (CB), and Brn2 (Appendix Fig. S28). Our data collectively demonstrate that the Y636C/E751Q substitution induces a LoF change in MDGA1 in both electrophysiological and behavioral phenotypes with sexual dimorphism, similar to the results obtained for $Mdga1$-cKO neurons, but without affecting the maturation of the cortical laminar structure.

## Proteomic analyses identify hypophosphorylation of synapsin II in adult male $Mdga1$-cKO and $Mdga1^{Y636C/E751Q}$ KI mice

To understand the mechanisms underlying MDGA1-LoF-induced ASD-like phenotypes, we attempted proteomic and phosphoproteomic analyses of hippocampal tissues from embryonic mice (E18), adult (P60) male $Mdga1$-cKO mice, male $Mdga1^{Y636C/E751Q}$ KI mice, and littermate control embryos or mice. The proteomic analysis showed that the MDGA1 protein level was significantly reduced in $Mdga1$-cKO mice but not in $Mdga1^{Y636C/E751Q}$ KI mice (Datasets EV3 and EV4; see also Appendix Fig. S15). Phosphopeptides were enriched in tryptic peptides using immobilized metal ion affinity chromatography (IMAC) and subjected to LC-MS/MS (Fig. 6A; Datasets EV5 and EV6). Compared to control mice, our

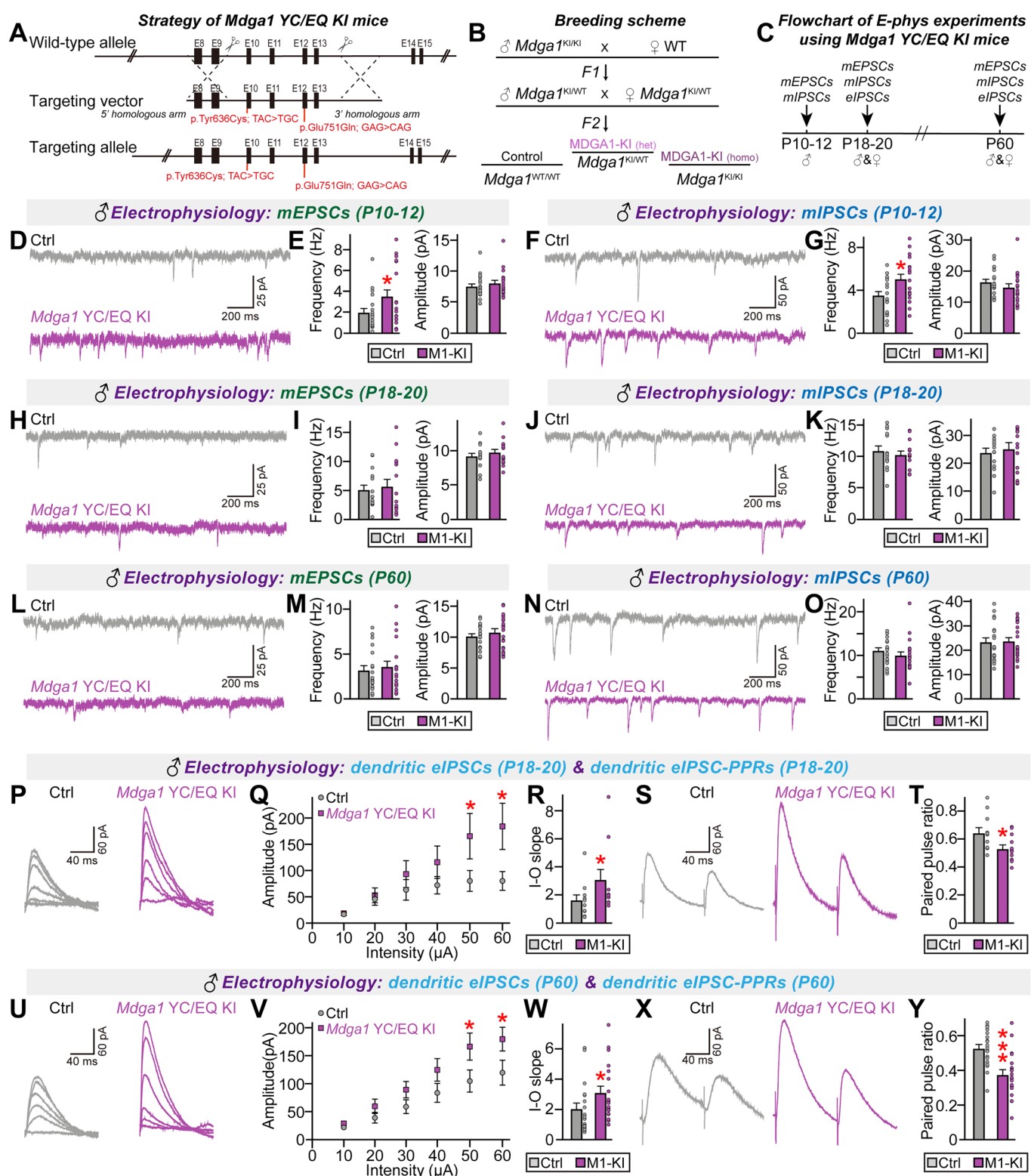

analysis of adult male *Mdga1*-cKO mice identified 28 differentially expressed phosphopeptides (DEPPs, *P* < 0.05; Dataset EV6) with significant changes (Log₂FC > 0.4 or Log₂FC < -0.4) belonging to 27 proteins (DEPs). In contrast, adult male *Mdga1*^Y636C/E751Q KI mice showed 83 DEPPs belonging to 75 DEPs (Fig. 6B,C; Dataset EV7).

Gene Ontology (GO) analyses were performed on the DEPs using Metascape, which enriched for GO terms related to synapse-related functions in both the *Mdga1*-cKO and *Mdga1*^Y636C/E751Q KI mouse analysis (Fig. 6D). A substantial fraction of the DEPs from adult male *Mdga1*-cKO and adult male *Mdga1*^Y636C/E751Q KI, mice (33.3%

◀

**Figure 5. Analysis of GABAergic synaptic properties in the hippocampal CA1 pyramidal neurons of Mdga1$^{Y636C/E751Q}$ KI mice.**

(A) Schematic illustrating the genetic targeting strategy used to generate Mdga1$^{Y636C/E751Q}$ KI mice. The point mutations p.Tyr636Cys (TAC > TGC) and p.Glu751Gln (GAG > CAG) are shown on the targeting allele and WT allele. (B) Breeding scheme followed to generate Mdga1$^{Y636C/E751Q}$ KI mice. (C) Flowchart of the electrophysiology experiments conducted on Mdga1$^{Y636C/E751Q}$ KI mice at different developmental stages (P10–12, P18–20 and P60). (D, E) Representative traces (D) and quantification (E) of mEPSCs in CA1 pyramidal neurons from control and Mdga1$^{Y636C/E751Q}$ KI mice at P10–12. Mdga1$^{Y636C/E751Q}$ KI mice exhibited a significant increase in mEPSC frequency. Data are presented as means ± SEMs ($n = 19$ cells/group; *$P = 0.0497$; Mann–Whitney $U$ test). (F, G) Representative traces (F) and quantification (G) of mIPSCs in CA1 pyramidal neurons from control and Mdga1$^{Y636C/E751Q}$ KI mice at P10–12. Mdga1$^{Y636C/E751Q}$ KI mice showed a significant increase in mIPSC frequency. Data are presented as means ± SEMs ($n = 18$–19 cells/group; *$P = 0.0335$; Mann–Whitney $U$ test). (H, I) Representative traces (H) and quantification (I) of mEPSCs in CA1 pyramidal neurons from control and Mdga1$^{Y636C/E751Q}$ KI mice at P18–20. Data are presented as means ± SEMs ($n = 15$ cells/group). (J, K) Representative traces (J) and quantification (K) of mIPSCs in CA1 pyramidal neurons from control and Mdga1$^{Y636C/E751Q}$ KI mice at P18–20. Data are presented as means ± SEMs ($n = 14$ cells/group). (L, M) Representative traces (L) and quantification (M) of mEPSCs in CA1 pyramidal neurons from control and Mdga1$^{Y636C/E751Q}$ KI mice at P60. Data are presented as means ± SEMs ($n = 19$ cells/group). (N, O) Representative traces (N) and quantification (O) of mIPSCs in CA1 pyramidal neurons from control and Mdga1$^{Y636C/E751Q}$ KI mice at P60. Data are presented as means ± SEMs ($n = 19$ cells/group). (P–T) Representative traces (P, S) and quantification of dendritic eIPSCs in CA1 pyramidal neurons from control and Mdga1$^{Y636C/E751Q}$ KI mice at P18–20. Mdga1$^{Y636C/E751Q}$ KI mice exhibited increased eIPSC amplitude (Q, R) and decreased PPR (T). Data are presented as means ± SEMs ($n = 11$–12 cells/group; (Q) *$P = 0.0358$ (50 μA), *$P = 0.0066$ (60 μA); (R) *$P = 0.0492$; (T) *$P = 0.0225$; Mann–Whitney $U$ test). (U–Y) Representative traces (U, X) and quantification of dendritic eIPSCs in CA1 pyramidal neurons from control and Mdga1$^{Y636C/E751Q}$ KI mice at P60. Mdga1$^{Y636C/E751Q}$ KI mice showed increased eIPSC amplitude (V, W) and decreased PPR (Y). Data are presented as means ± SEMs ($n = 19$ cells/group; (V) *$P = 0.0247$ (50 μA), *$P = 0.0178$ (60 μA); (W) *$P = 0.0219$; (Y) *$P = 0.0004$; Mann–Whitney $U$ test). Source data are available online for this figure.

and 3%, respectively) belonged to the SynGO protein category (Fig. 6E–G). Among the DEPs, synapsin II (SynII) and OTU domain-containing protein 5 (Otud5) were commonly down-regulated in adult male Mdga1-cKO and Mdga1$^{Y636C/E751Q}$ KI mice (Fig. 6H,I).

A STRING analysis of protein-protein interactions (PPIs) was performed using downregulated DEPs (Fig. 6J,K). Intriguingly, clear downregulation of phosphorylation levels was seen for SynII in adult male Mdga1-cKO and adult male Mdga1$^{Y636C/E751Q}$ KI mice, respectively, compared to the corresponding control mice. Synaptic vesicle clustering and maintenance of synapse structure were identified as the most relevant biological processes; the relevant DEPs included SynI, SynII, Piccolo, Bassoon, and RIMS1 (Appendix Fig. S30). Notably, the phosphorylation levels of these proteins were not significantly altered in our phosphoproteomic analysis of E18 embryos from Mdga1$^{Y636C/E751Q}$ KI mice (Appendix Fig. S31; Dataset EV8). These extensive proteomic analyses revealed that there are marked developmental-stage-related differences in the profiles of DEPPs, DEPs, and phosphorylation level changes in a subset of presynaptic proteins, including Syn proteins, that might manifest as sexually dimorphic abnormalities in MDGA1 functions (see below).

## Bazedoxifene administration rescues reduced SynII phosphorylation, impaired communicative behavior and GABAergic synaptic transmission in adult male Mdga1$^{Y636C/E751Q}$ KI mice

Given that both male and female Mdga1-mutant mice exhibited comparable increases in inhibitory synaptic transmission during the juvenile period, whereas only males (and not females) retained these abnormalities into adulthood (see Figs. 5 and EV6), we considered the possibility that puberty-associated, sex-dependent factors contribute to this divergence (Ferri et al, 2018). We therefore examined whether estrogen-sensitive pathways—which are known to influence inhibitory synapse maturation and circuit refinement—modulate the persistent adult male phenotype. Proteomic analyses revealed no significant changes in the levels of estrogen receptor-associated proteins (Ddx5 and Phb2), but marginal (yet significant) changes in the levels of the NDD risk factors, Cnot1 and Parp1 (Dong et al, 2023; Humphries et al, 2023;

Vissers et al, 2020) (Appendix Fig. S32), in male Mdga1-cKO and Mdga1$^{Y636C/E751Q}$ KI mice, hinting that there is a link between the MDGA1 and estrogen receptor signaling pathways. We also performed enzyme-linked immunosorbent assay (ELISA)-based measurement of estradiol levels in male juvenile and adult Mdga1$^{Y636C/E751Q}$ KI mice and found no significant differences in estradiol levels between WT and KI mice (Appendix Fig. S33). These findings indicate that baseline estrogen signaling is not substantially altered in Mdga1-mutants, but raise the possibility that enhancing estrogen-responsive pathways might still modulate the persistent adult male phenotypes.

Thus, we utilized bazedoxifene (BZD), which is a third-generation selective estrogen receptor modulator that is blood-brain barrier (BBB)-penetrant and Food and Drug Administration (FDA)-approved for treating menopausal symptoms (Hill et al, 2020; Peng et al, 2017; Zafar et al, 2022). Our decision to employ acute BZD treatment was based on a previous report that BZD shows rapid brain penetrance and swift neuromodulatory potential (Hill et al, 2020). We acutely injected adult male Mdga1$^{Y636C/E751Q}$ KI mice with a single dose of BZD (4.1 mg/kg) and, 6 h later, performed semi-quantitative immunoprecipitation analyses using phosphoserine antibodies (recognizing phosphorylated Syn residues) to measure changes in Syn protein phosphorylation, and applied a subset of behavioral tests to measure ASRs, PPI and USVs (Fig. 7A). As hypothesized, administration of BZD rescued the reduced protein phosphorylation of SynII isoforms (but not SynI), the downregulation of GABAergic synaptic proteins (Nlgn2, GABA$_A$Rγ2 and VGAT; Fig. 7B–E), and several behavioral abnormalities, all of which returned to the levels seen in littermate control or female counterpart mice (Fig. 7F–K). Notably, a subset of altered USV syllable types (including complex trill) and ASRs, but not PPI, was rescued by BZD treatment of adult male Mdga1$^{Y636C/E751Q}$ KI mice (Fig. 7H–K). In addition, the enhancement of GABAergic synaptic strength in the distal dendrites of the CA1 pyramidal neurons from adult male Mdga1$^{Y636C/E751Q}$ KI mice was restored to control levels upon BZD administration (Fig. 7L–P). The normalized eIPSC amplitudes and PPRs of eIPSCs in the CA1 pyramidal neurons from adult male Mdga1$^{Y636C/E751Q}$ KI mice were partially reversed by acute treatment with the estrogen receptor antagonist, Fulvestrant (5 μM) (Robertson and Harrison, 2004) (Appendix Fig. S34). Although this partial reversal does not establish a direct molecular mechanism, it is in line with the idea that acute modulation of estrogen-sensitive signaling can

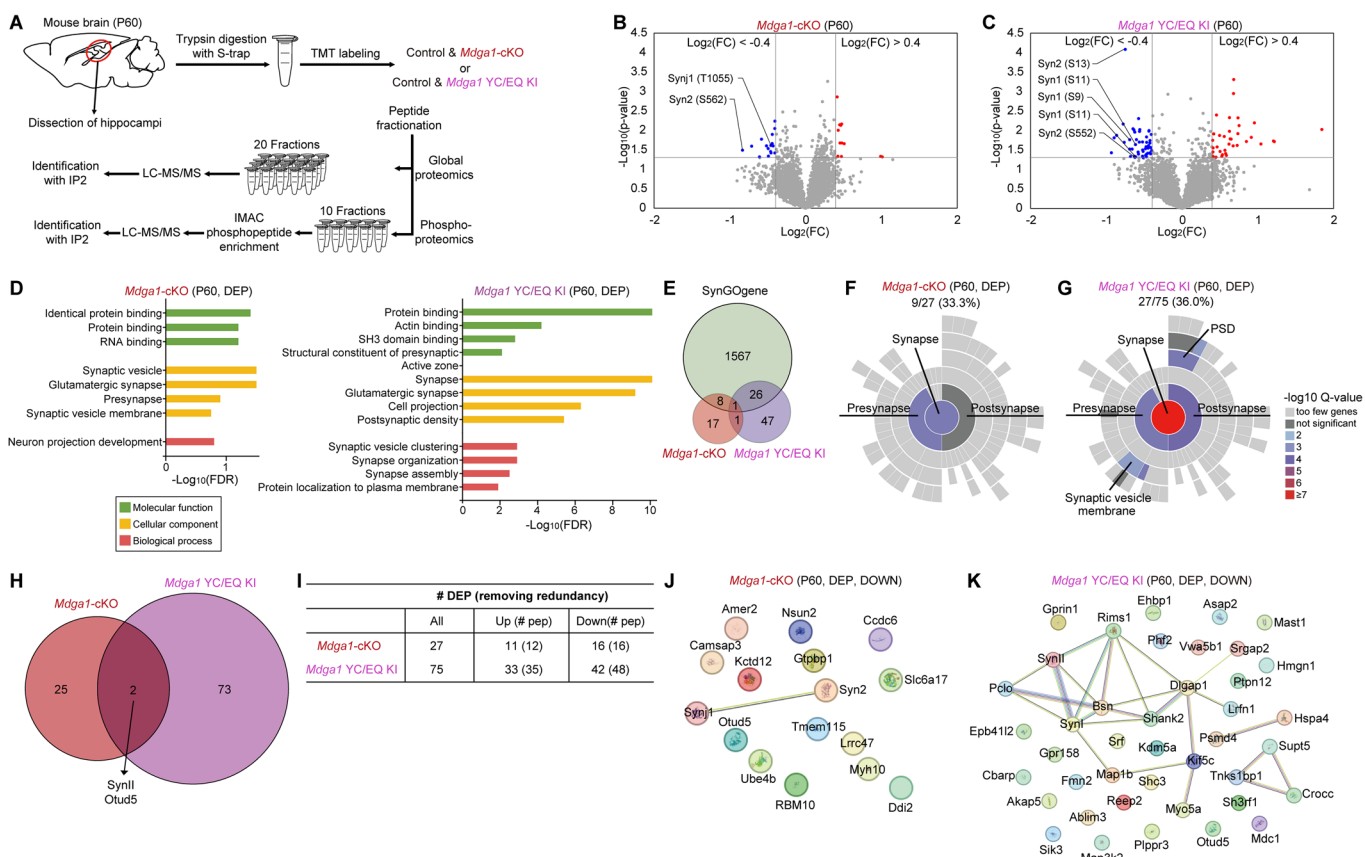

**Figure 6. Phosphoproteomic analysis of hippocampi from adult *Mdga1*-cKO and *Mdga1*^Y636C/E751Q^ KI mice.**

(A) Schematic diagram of phosphoproteomic workflow using hippocampal lysates from adult (8-week-old) male *Mdga1*-cKO, *Mdga1*^Y636C/E751Q^ KI and littermate control mice ($n = 4$ per group). (B) Volcano plot of 4338 phosphopeptides identified from hippocampal lysates of adult male *Mdga1*-cKO mice. Differentially expressed phosphopeptides (DEPPs) with significant increases or decreases (Log2FC < -0.4 or > 0.4; $P < 0.05$; Welch's $T$ test) are shown in red (12 phosphopeptides) and blue (16 phosphopeptides), respectively. (C) Volcano plot of 5124 phosphopeptides identified from hippocampal lysates of adult male *Mdga1*^Y636C/E751Q^ KI mice. DEPPs with significant increases or decreases (Log2FC < -0.4 or > 0.4; $P < 0.05$; Welch's T-test) are shown in red (35 phosphopeptides) and blue (48 phosphopeptides), respectively. (D) Gene ontology (GO) analysis was performed on differentially expressed phosphoproteins (DEPs) identified from the phosphoproteomic analyses of adult male *Mdga1*-cKO (left) and *Mdga1*^Y636C/E751Q^ KI (right) mice. The bar charts depict the significantly enriched GO terms of the Molecular Function (green), Cellular Component (yellow), and Biological Process (red) categories, represented as -log₁₀(FDR) values. (E) Venn diagram visualization of SynGO category proteins. (F) Localization annotations of SynGO category proteins from adult male *Mdga1*-cKO mice. (G) Localization annotations of SynGO category proteins from adult male *Mdga1*^Y636C/E751Q^ KI mice. (H) Venn diagram showing the overlap of DEPs between *Mdga1*-cKO and *Mdga1*^Y636C/E751Q^ KI mice. Two proteins, SynII and Otud5, are shared between condition; 25 DEPs are unique to *Mdga1*-cKO mice and 73 are unique to *Mdga1*^Y636C/E751Q^ KI mice. (I) Summary table of DEPs after removal of redundancy. The table shows the total number DEPs with a breakdown of upregulated and downregulated proteins in *Mdga1*-cKO and *Mdga1*^Y636C/E751Q^ KI mice, together with the number of unique peptides (# pep). (J) STRING analysis of downregulated DEPs from *Mdga1*-cKO mice ($P < 0.05$ and Log2FC < -0.4; Welch's $T$ test). (K) STRING analysis of downregulated DEPs from *Mdga1*^Y636C/E751Q^ KI mice ($P < 0.05$ and Log2FC < -0.4; Welch's $T$ test).

influence the persistent inhibitory abnormalities observed in adult males. Viewed together, our results highlight different combinations of synaptic, behavioral and hormonal mechanisms that can produce ASD-like manifestations and suggest that MDGA1 could be a reliable biomarker for studying a putative biological basis for the male preponderance of ASDs.

# Discussion

Here we propose that MDGA1, previously characterized as a GABAergic synapse-specific suppressor, may serve as a potential translational biomarker for *MDGA1*-associated neurodevelopmental and/or neuropsychiatric disorders, including ASDs. Given that

various components of MDGA1-containing synaptic complexes have consistently been linked to ASDs (Kim et al, 2021), we examined the possibility that MDGA1 might also be implicated in ASDs. The current study provides new insights into the pathways by which two ASD-associated *MDGA1* variants induce previously recognized pathogenesis, including abnormal neuronal migration and imbalanced excitation/inhibition (E/I) ratios (Contractor et al, 2021; de la Torre-Ubieta et al, 2016; Delorme et al, 2013; Nisar et al, 2022; Pan et al, 2019; Sohal and Rubenstein, 2019; Willsey et al, 2022). Intriguingly, MDGA1 variants manifested distinct LoF and GoF phenotypes during various stages of postnatal development, highlighting MDGA1 as a multi-functional protein that impinges on a spectrum of neurodevelopmental processes in a spatiotemporally specific manner. Specifically, we found that the V116M/

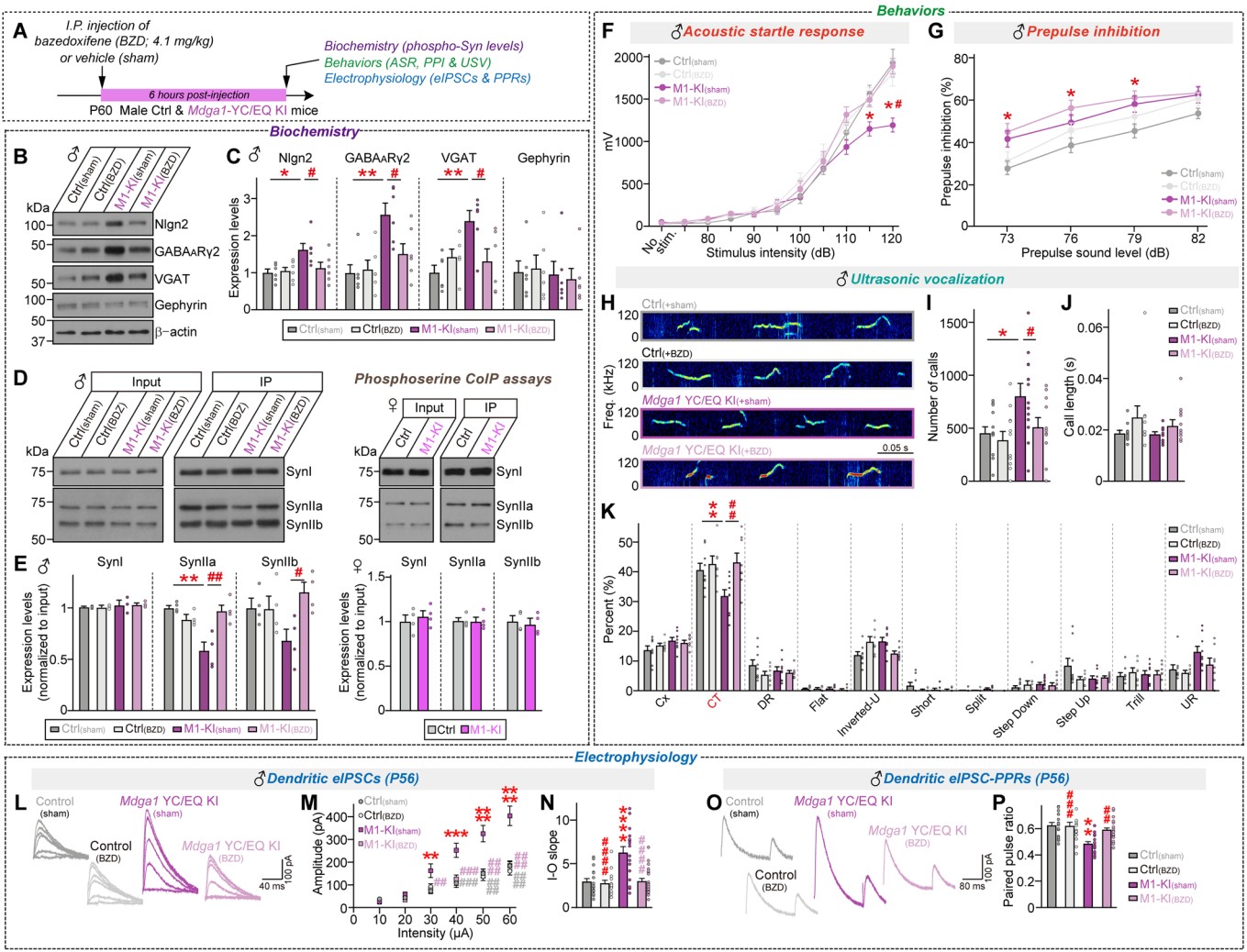

**Figure 7. Bazedoxifene rescues sexually dimorphic alterations in the hippocampal CA1 pyramidal neurons of *Mdga1*[Y636C/E751Q] KI mice.**

(A) Schematic illustrating the experimental paradigm. Adult male control and *Mdga1*[Y636C/E751Q] KI mice were administered bazedoxifene (BZD; 4.1 mg/kg, intraperitoneally) or vehicle (sham). Six hours post-treatment, the mice were subjected to biochemical, electrophysiological, and behavioral analyses. (B, C) Semi-quantitative immunoblotting of hippocampal synaptic proteins, including Nlgn2, $GABA_A$ receptor subunit, VGAT, and gephyrin from BZD- or sham-treated mice. β-actin was used as a loading control for normalization. Quantification of normalized protein levels (C) revealed that BZD rescued the enhanced expression of synaptic proteins observed in *Mdga1*[Y636C/E751Q] KI mice. Data are means ± SEMs ($n = 6$ per group; (Nlgn2) *$P = 0.0220$, #$P = 0.0313$; ($GABA_A$Rγ2) **$P = 0.0021$, #$P = 0.0495$; (VGAT) **$P = 0.0059$, #$P = 0.0343$; '#' indicates statistical comparisons with their counterparts; two-way ANOVA with Tukey's *post hoc* test). (D, E) Representative immunoblots (D) and quantification (E) of phosphorylated SynI, SynIIa, and SynIIb proteins obtained via immunoprecipitation with anti-phospho-serine antibody and normalized to total Syn levels. BZD significantly restored phospho-SynII levels in KI hippocampi. Data are means ± SEMs ($n = 4$ independent experiments; (SynIIa) **$P = 0.0010$, ##$P = 0.0019$; (SynIIb) #$P = 0.0441$; '#' indicates statistical comparisons with their counterparts; two-way ANOVA with Tukey's *post hoc* test). (F, G) ASRs (F) and PPI of ASR (G) in adult male control and *Mdga1*[Y636C/E751Q] mice given BZD or sham treatment. BZD treatment reversed the reduced ASR in *Mdga1*[Y636C/E751Q] KI mice, but failed to rescue the deficits in PPI. Data are presented as means ± SEMs ($n = 20–21$ mice/group; (F) *$P = 0.0444$ (115 dB), *$P = 0.0281$ (120 dB), #$P = 0.0344$ (120 dB); (G) *$P = 0.0195$ (73 dB), *$P = 0.0205$ (76 dB), *$P = 0.0429$ (79 dB); '#' indicates statistical comparisons with their counterparts; two-way ANOVA followed by Tukey's *post hoc* test). (H–K) USVs recorded from socially isolated adult male mice. Representative traces are shown in (H). BZD treatment normalized the increased number of USV calls (I) observed in *Mdga1*[Y636C/E751Q] KI mice. Call length (J) was not significantly altered across groups. The altered distribution of call types in *Mdga1*[Y636C/E751Q] KI mice was also partially rescued by BZD, including restoration of the reduced proportion of complex trill calls (K). Data are means ± SEMs ($n = 9–10$ mice/group; (I) *$P = 0.0323$, #$P = 0.0456$; (K) **$P = 0.0089$, ##$P = 0.0045$; # indicates statistical comparisons with their counterparts; two-way ANOVA followed by Tukey's *post hoc* test). (L–P) Electrophysiological recordings of dendritic eIPSCs in CA1 pyramidal neurons at P56. Representative traces (L, O) and quantification of I-O responses (M, N) and paired-pulse ratio (PPR; P) revealed that BZD rescued the increased inhibitory input strength and the presynaptic release probability abnormalities observed in *Mdga1*[Y636C/E751Q] KI mice. Data are means ± SEMs ($n = 14–21$ cells/group; (M) (30 μA) **$P = 0.0065$ (vs. MI-KI_Sham), ##$P = 0.0064$ (vs. MI-KI_BZD); (40 μA) ###$P = 0.0003$ (vs. Ctrl_BZD), ***$P = 0.0001$ (vs. MI-KI_Sham), ###$P = 0.0004$ (vs. MI-KI_BZD); (50 μA) ####$P < 0.0001$ (vs. Ctrl_BZD), ****$P < 0.0001$ (vs. MI-KI_Sham), ####$P < 0.0001$ (vs. MI-KI_BZD); (60 μA) ####$P < 0.0001$ (vs. Ctrl_BZD), ****$P < 0.0001$ (vs. MI-KI_Sham), ####$P < 0.0001$ (vs. MI-KI_BZD); (N) ####$P < 0.0001$ (vs. Ctrl_BZD), ****$P < 0.0001$ (vs. M1-KI_sham); (P) ###$P = 0.0004$ (vs. Ctrl_BZD), **$P = 0.0010$ (vs. M1-KI_sham), ##$P = 0.0072$ (vs. MI-KI_BZD); '#' indicates statistical comparisons with their counterparts; two-way ANOVA followed by Tukey's *post hoc* test). Source data are available online for this figure.

A688V mutant impacts neuronal migration in the cortex and glutamatergic transmission in the hippocampus, whereas the Y635C/E756Q mutant selectively modulates hippocampal GABAergic synapses. These results differentiate MDGA1 from other proteins encoded by known ASD risk genes (Heavner and Smith, 2020) and provide a framework for understanding the divergent behavioral outcomes between genotypes. In particular, the GABAergic dysfunction observed in the Y635C/E756Q mutant contrasts with classic pathophysiological mechanisms. While ASD-related alterations at glutamatergic synapses have been comprehensively investigated (Galineau et al, 2023; Moretto et al, 2018; Nisar et al, 2022; Ramaswami and Geschwind, 2018), GABAergic pathologies are typically attributed to global deficits, such as decreased GABA levels and loss of PV$^+$ interneurons that increase the E/I ratio and network hyperactivity (Coghlan et al, 2012; Zhao et al, 2021). Furthermore, a number of synaptic genes affect both glutamatergic and GABAergic systems (Zhao et al, 2021), complicating the evaluation of specific contributions from GABAergic synapse dysfunctions in ASD pathophysiology. However, the *MDGA1* variants analyzed here did not directly alter the numbers of PV$^+$ interneurons, or intrinsic neuronal excitability, nor did they affect the binding of MDGA1 to its ligands, Nlgn2 and APP, both of which are also encoded by ASD risk genes (Ali et al, 2020; Sokol et al, 2006). Instead, the Y635C/E756Q mutant induced a circuit-specific alteration of inhibitory synapses, providing a novel mechanistic basis for the malfunction of GABAergic synapse components in ASDs distinct from global network hyperactivity.

Despite providing significant mechanistic insights into the role of MDGA1 in neurodevelopmental processes, our study has several limitations. While the *Mdga1*$^{Y636C/E751Q}$ knock-in (KI) mice meticulously recapitulate the specific patient-identified genetic variants, we acknowledge that no single animal model can fully mirror the complex and multifactorial nature of human neurodevelopmental and neuropsychiatric disorders. The full spectrum of complex human clinical features, such as those seen in ASD, is indeed only partially recapitulated in any rodent model. Our model primarily serves as a valuable tool for dissecting the causal molecular and cellular mechanisms underlying MDGA1 dysfunction and specific, quantifiable behavioral deficits, rather than offering high face validity for the entire human condition. Further research is necessary to bridge these gaps between preclinical models and human pathologies.

Both MDGA1 variants led to altered USVs in pups, a phenotype reminiscent of communicative deficits in autism. In utero overexpression of MDGA1 V116M/A688V impaired both USVs and neuronal migration, whereas MDGA1 Y635C/E756Q did not elicit such effects in the same context. Intriguingly, pups from *Mdga1*$^{Y636C/E751Q}$ KI and *Mdga1*-cKO models exhibited reduced isolation-induced USVs, suggesting that endogenous MDGA1 may buffer mutant-induced defects in transient expression systems. These phenotypic discrepancies likely reflect differences in genetic context and developmental timing across models. Notably, USV impairments in *Mdga1*$^{Y636C/E751Q}$ KI mice occurred despite intact cortical lamination, unlike the disrupted positioning seen with MDGA1 V116M/A688V overexpression. This divergence suggests that MDGA1 may regulate early communicative behaviors via region- and temporally specific mechanisms beyond its ability to modulate cortical lamination. Supporting this, MDGA1 expression is enriched in cortical excitatory neurons during mid-gestation, but

its overexpression in the adult cortex has no electrophysiological effect. Together, these findings underscore the spatiotemporal specificity of MDGA1 function and highlight the need for refined models to dissect its discrete roles in ASD pathogenesis.

The Y636C/E751Q substitution did not alter the hippocampal level of MDGA1, despite our in silico prediction that these changes would exert a subtle destabilizing effect. Remarkably, adult male *Mdga1*$^{Y636C/E751Q}$ KI mice exhibited increased USV calls with abnormal USV syllable composition; this behavior contrasts with the decreased USV calls observed in adult male *Mdga1*-cKO mice, which is typical of other ASD model mice. Rather than a direct GoF effect, the increased USV calls in adult male *Mdga1*$^{Y636C/E751Q}$ KI mice likely represent a paradoxical behavioral consequence of network dysregulation driven by a LoF mechanism. While this interpretation aligns with the observed synaptic deficits, we do not exclude the potential for subtle, context-dependent GoF effects contributed by the mutation. Further studies are warranted to unravel the mechanisms by which the Y635C/E756Q substitution leads to these functional deficits despite preserved protein stability, and to determine whether the V116M/A688V substitution causes a similar change in MDGA1 protein levels.

The two *MDGA1* variants exerted distinct effects on synaptic properties. MDGA1 WT and MDGA1 V116M/A688V suppressed GABAergic synapses, whereas MDGA1 Y635C/E756Q did not exhibit this activity. Strikingly, MDGA1 V116M/A688V impaired the dendritic spine structure and excitatory synaptic transmission. Whereas forebrain-specific *Mdga1*-cKO mice unambiguously demonstrated effects only in GABAergic synapses (Kim et al, 2024a), MDGA1 V116M/A688V had unexpected effects in excitatory synapses. Previous studies established that glutamate plays an important role in neuronal migration, and neuronal migration defects are linked to disruptions in glutamatergic neurotransmission (Gressens, 2000; Luhmann et al, 2015; Moretto et al, 2018). MDGA1 forms complexes with the gap junction subunit, connexin 43 (Perez-Garcia and O'Leary, 2016), to provide a neurogenic niche within subventricular zones during cortical neuron migration, and contributes to sustaining glutamatergic synaptic activity (Chever et al, 2014). Thus, it is plausible that during the early phase of CNS development, MDGA1-mediated biological processes might be driven by glutamate. Alternatively, a small population of endogenous MDGA1 expressed at glutamatergic synapses could possibly contribute to this process. Further investigation is warranted to unravel the underlying mechanisms. The MDGA1 Y635C/E756Q mutant appears to exhibit a destabilized triangular extracellular structure of MDGA1 (as suggested by in silico prediction and confirmed by TEM analyses). Our findings thus support the recent claim that maintenance of the compact three-dimensional structure of MDGA1 is physiologically significant and required for its function at the synaptic cleft in mature neurons (Lee et al, 2023). Indeed, we tested the biological effects of MDGA1 mutations that could modulate compactness of its extracellular triangular structure and found that optimal integrity of MDGA1 structures is essential for its GABAergic synapse-suppressing activities (Appendix Fig. S35).

Our investigation into the role of MDGA1 in influencing synaptic protein localization and function, particularly the observed suppression of GABAergic synapses, involved overexpression experiments. While the obtained results provide valuable mechanistic insights into the intrinsic ability of MDGA1 to modulate

synaptic architecture and its interaction with key synaptic partners, it is crucial to interpret these findings with caution regarding their direct translation to physiological endogenous conditions. Over-expression systems can sometimes influence native protein trafficking, induce non-physiological interactions, or mask subtle effects that occur at endogenous expression levels. Thus, results from overexpression approaches should primarily serve as a complementary tool for exploring underlying molecular mechanisms, as partially supported by our parallel LoF experiments in $Mdga1^{Y636C/E751Q}$ KI mice. Furthermore, our approach of incorporating two MDGA1 missense variants into a single expression vector to study their combined effect does not precisely replicate the compound heterozygous genetic context observed in patients, where each mutation occurs on a separate allele. While this design allowed us to efficiently investigate the synergistic or additive functional consequences of these co-occurring mutations on the properties of MDGA1, it hinders us from fully dissecting the individual contributions or potential epistatic interactions of each variant in vivo. Future studies employing separate allelic models or more sophisticated genetic engineering approaches would be invaluable for elucidating the precise role of each individual mutation within a true compound heterozygous setting.

Our proteomic analyses indicated that $Mdga1$-cKO and $Mdga1^{Y636C/E751Q}$ KI neurons display reduced phosphorylation of presynaptic SynII, which likely desynchronizes neurotransmitter release at GABAergic synapses. This is consistent with the observation that the action of MDGA1 is transsynaptically linked to presynaptic neurons (Kim et al, 2022a). We speculate that MDGA1-mediated negative transsynaptic regulation occurs via precisely tethered APP protein, which is further coupled with the activity of SynII proteins to orchestrate GABA release at the synaptic cleft that is designed to maximize synaptic strength via the precise alignment of pre- and postsynaptic compartments (Biederer et al, 2017; Savtchenko and Rusakov, 2007). It is strategically advantageous for MDGA1 to adopt a compact triangular conformation that facilitates mechanical stabilization and efficient transsynaptic signaling (Savtchenko and Rusakov, 2007; Zuber et al, 2005). It would be interesting to explore whether the circuit-specific introduction of extracellularly stabilized MDGA1 protein or drugs that selectively boost specific SynII phosphorylation site(s) could have beneficial effects on ASDs in the context of various neurodevelopmental trajectories.

Adult male $Mdga1$-cKO mice exhibited a range of autistic-like behavioral abnormalities. These changes included mild deficits in sociability, reduced USV call numbers and USV duration, and abnormal sensorimotor gating, supporting a genetic association of $MDGA1$ with ASDs. The presence of deficits in both sociability and USVs indicates that these mice display abnormalities in multiple ASD-relevant domains (i.e., two of the three core diagnostic domains of ASD). Importantly, USV deficits have been consistently reported in well-established ASD models (Giua et al, 2024; Premoli et al, 2021; Scattoni et al, 2008), aligning well with our findings and reinforcing USVs as a crucial ASD-related endophenotype. Furthermore, $Mdga1$-cKO mice showed abnormal sensorimotor gating, as evidenced by deficits in PPI. While PPI has been traditionally used in schizophrenia research, it should be noted that it is also widely employed in ASD studies. This assay is particularly valuable in capturing deficits in sensory processing and sensorimotor gating, which are increasingly recognized as part of the ASD

phenotype. PPI deficits have been consistently reported in multiple genetic ASD models (Doornaert et al, 2025; Ellison et al, 1989; Kokash et al, 2019). Human studies have similarly shown reduced PPI in individuals with autism and Asperger's syndrome compared to neurotypical controls (McAlonan et al, 2002; Perry et al, 2007), indicating its robust translational value. The behavioral profiles of adult male $Mdga1$-cKO mice differed from those recently reported for $Mdga2$-KO mice (Wang et al, 2024), which is consistent with the report that MDGA1 and MDGA2 play distinct roles in negatively regulating GABAergic and glutamatergic synapses, respectively (Kim et al, 2024a). However, given that MDGA2 forms complexes with Nlgn1 (or Nlgn3) and Nrxns, and Nlgn3 is closely linked to ASDs, it is highly likely that MDGA2 is also associated with ASDs. Intriguingly, the total level of MDGA paralogs appears to be critical for maintaining synaptic homeostasis in neurons (Kim et al, 2024a). Thus, the mechanisms by which the unique action of each MDGA paralog contributes to synaptic homeostasis in neurons and whether its impairment triggers ASDs require further elucidation. Although the behavioral phenotype in the $Mdga1$-cKO mice is broad compared to the selective deficits in $Mdga1^{Y636C/E751Q}$ KI mice, we emphasize that their molecular and electrophysiological profiles closely mirror those observed in the $Mdga1^{Y636C/E751Q}$ KI mice. This supports the notion that there is a shared synaptopathic mechanism across genotypes, reinforcing the behavioral findings.

The behavioral impairments we herein documented in forebrain excitatory neuron-specific $Mdga1$-cKO mice support the pathogenicity of MDGA1 deficiency. However, it is challenging to directly translate these animal model-derived results to human patients, as the same genetic mutation can often result in different clinical presentations. The functional consequences of MDGA1 mutations may depend on variable factors, such as protein stability, interacting protein network and/or neuronal functions. Given the critical role of MDGA1 in suppressing GABAergic synaptic properties, it is possible that even slight differences in MDGA1 dysfunction could contribute to variations in disease severity. While our findings established a strong connection between biallelic MDGA1 mutations and neurodevelopmental processes, further study is required to fully understand how different variant combinations influence clinical outcomes. In this regard, patient-derived neuronal models will be crucial for unraveling the genotype-phenotype relationship in MDGA1-associated NDDs. In view of this, the immediate goals of future studies should be to introduce idiopathic ASD-associated MDGA1 mutation(s) into induced neuronal cells and determine whether this can replicate major cellular disease phenotypes in human neurons. Although further explorations are warranted, our study provides important clues into how proper control of GABAergic synaptic inhibition is mechanistically linked to ASD pathogenesis. Intriguingly, adult female $Mdga1$-cKO mice did not show any of the prominent behavioral deficits seen in their male counterparts. Together with our extensive analyses of male and female $Mdga1^{Y636C/E751Q}$ knock-in mice, these results suggest that our MDGA1 mouse models could be used in the future to elucidate the sexually dimorphic nature of ASDs.

Our study raises several questions/issues that remain to be addressed. Whereas adult female $Mdga1$-cKO and $Mdga1^{Y636C/E751Q}$ KI mice displayed no abnormal behavioral or electrophysiological phenotype, juvenile female $Mdga1$-cKO and $Mdga1^{Y636C/E751Q}$ KI

mice exhibited electrophysiological phenotypes similar to those of their male counterparts. Although dimorphic phenotypes have been reported in numerous ASD models (Jeon et al, 2018), definitive explanations are lacking. The female protective effect theory of ASDs (Dougherty et al, 2022; Tsai et al, 1981) is compelling, since a sudden peak in gonadal hormone levels during the prepubertal period of female mouse development appears to contribute to an anti-inflammatory state that further regulates hippocampal CA1 GABAergic synapse organization (Mukherjee et al, 2017; Tabatadze et al, 2015). We herein demonstrate that BZD may be useful for treating certain aspects of ASDs, and our findings further imply that estradiol signaling may act on synaptic disinhibition to normalize SynII activities (Ohtani-Kaneko et al, 2010). Although studies on the acute neuromodulatory effects of BZD treatment are still limited, other SERMs, such as tamoxifen and raloxifene, have demonstrated similar rapid effects on neuronal plasticity behavior (Finney et al, 2021; Velazquez-Zamora et al, 2012), supporting the broader concept that estrogen receptor modulation can exert biologically significant acute effects on the brain within hours of administration. Consistent with our model, acute application of the estrogen receptor degrader, Fulvestrant, partially suppressed the BZD-induced normalization of inhibitory synaptic function in Mdga1-KI slices. Because Fulvestrant requires ≥6–12 h to induce ubiquitin–proteasome–mediated degradation of nuclear ERα (Hammes and Levin, 2007; Long and Nephew, 2006), the brief 20-min exposure used here is expected to selectively interfere with rapid membrane-initiated estrogen signaling (Callige and Richard-Foy, 2006; Vasudevan and Pfaff, 2007). This selective and mechanism-specific partial blockade provides direct functional evidence that the BZD-induced rescue requires intact estrogen receptor signaling, supporting the involvement of an estrogen-sensitive pathway in the MDGA1-dependent synaptic phenotype. Although the details of the relevant molecular interactions remain to be fully elucidated, we note that estrogen signaling is known to rapidly reduce inhibitory synaptic strength in the hippocampus (Huang and Woolley, 2012; Mukherjee et al, 2017). These fast-acting suppressive effects on GABAergic function may act synergistically with the previously established role for MDGA1 as an inhibitory synapse suppressor. Thus, we speculate that BZD may transiently compensate for the loss of MDGA1 function in male Mdga1-KI mice through partially overlapping pathways that remain to be identified in future studies. Moreover, GABAergic synaptic inhibition is essential for controlling the window of the critical period of plasticity (Andrade-Talavera et al, 2023) and its precocious closure is linked to ASDs (Berger et al, 2013; LeBlanc and Fagiolini, 2011), giving rise to the speculation that MDGA1 is a key regulatory factor in the development of critical-period plasticity. The effects of BZD administration should be systematically examined in other ASD model animals that exhibit sexually dimorphic abnormalities. We also need to better understand the synaptic and circuit-level actions of BZD and how long its rescue effects last during development. Such studies would help inform the optimization of BZD dosing and administration frequency, with the goal of avoiding potentially adverse side effects (Raina et al, 2024). Given the significant expression of MDGA1 in astrocytes, it would also be interesting to establish whether both neuronal and astrocytic MDGA1 are engaged in shaping the critical periods that determine sexual dimorphism during development across diverse brain areas (Ackerman et al, 2021; McCarthy and Wright, 2017;

Ribot et al, 2021). Another future challenge will be to determine the cell types, circuits, and/or regions that are primarily involved in eliciting various behavioral manifestations of Mdga1-cKO and Mdga1[Y636C/E751Q] KI mice. Notably, overexpression or deletion of MDGA1 in mPFC neurons failed to exert significant effects on synaptic properties and mouse behavior, partly in line with the low-level expression of MDGA1 in the mPFC. Considering the contributions of the hippocampus to the social and cognitive deficits in ASDs (Banker et al, 2021), our results obtained from the hippocampal CA1 GABAergic circuits are compellingly relevant to ASD pathophysiology. To gain a better understanding of ASD biology, the investigation of MDGA1 functions should be extended beyond the hippocampus and mPFC to other regions related to regulating ASD-like behaviors, such as the thalamus (Hwang et al, 2017; Roy et al, 2022), with the goals of addressing the canonical/non-canonical roles of MDGA1 and establishing its utility as a genuine biomarker of ASD. Importantly, in addition to homozygous Mdga1[Y636C/E751Q] KI mice, heterozygous Mdga1[Y636C/E751Q] KI mice should be studied to elucidate the detailed abnormalities relevant to ASD patients. Finally, while our study provides compelling preclinical evidence for the pathogenic role of MDGA1 in GABAergic synaptic dysfunction and neurodevelopmental phenotypes, the direct medical relevance and translational biomarker potential of MDGA1 require further rigorous validation. This potential has been suggested, but large-scale clinical and functional studies are needed for further validation (Table EV3). The number of reported human cases with MDGA1 mutations currently remains limited, and MDGA1 is not yet widely recognized as a major ASD risk gene in comprehensive databases such as SFARI. Our findings establish MDGA1 as a novel candidate gene and provide critical mechanistic insights, but extensive future validation in larger human cohorts, including functional studies of patient-derived cells and comprehensive genetic screenings, will be essential for firmly establishing its utility as a reliable diagnostic, prognostic, and/or therapeutic biomarker. These ongoing efforts will be crucial for bridging the gap between basic mechanistic discoveries and clinical applications.

## Methods

**Reagents and tools table**

| Reagent/resource | Reference or source | Identifier or catalog number |
|---|---|---|
| **Experimental models** | | |
| Mouse: ICR (E15.5 embryos) | Daehan Biolink | N/A |
| Mouse: Mdga1[f/f] | PMID: 35074912 | N/A |
| Mouse: Mdga1[Y636C/E751Q] KI | Biocytogen | N/A |
| **Recombinant DNA** | | |
| pAAV-MDGA1 WT | PMID: 35074912 | N/A |
| pAAV-MDGA1 V116M/A688V | This paper | N/A |
| pAAV-MDGA1 Y635C/E756Q | This paper | N/A |
| pCAGG-His-HA-MDGA1 WT | This paper | N/A |

| Reagent/resource | Reference or source | Identifier or catalog number |
|---|---|---|
| pCAGG-His-HA-MDGA1 V116M/A688V | This paper | N/A |
| pCAGG-His-HA-MDGA1 Y635C/E756Q | This paper | N/A |
| pCIG2-HA-MDGA1 WT | This paper | N/A |
| pCIG2-HA-MDGA1 V116M/A688V | This paper | N/A |
| pCIG2-HA-MDGA1 Y635C/E756Q | This paper | N/A |
| pCMV-IgC APP | PMID: 35074912 | N/A |
| pCMV-IgC Nlgn2 | PMID: 23248271 | N/A |
| pDisplay-MDGA1 WT | PMID: 23248271 | N/A |
| pDisplay-MDGA1 V116M/A688V | This paper | N/A |
| pDisplay-MDGA1 Y635C/E756Q | This paper | N/A |
| pDisplay-MDGA1 T135C/N313C | This paper | N/A |
| pDisplay-MDGA1 V314D | This paper | N/A |
| pDisplay-MDGA1 T333C/Y519C | This paper | N/A |
| pDisplay-MDGA1 I336D | This paper | N/A |
| L313-MDGA1 WT | PMID: 23248271 | N/A |
| L313-MDGA1 V116M/A688V | This paper | N/A |
| L313-MDGA1 Y635C/E756Q | This paper | N/A |
| pDisplay-MDGA2 WT | PMID: 23248271 | N/A |
| pDisplay-MDGA2 Y631C/E751Q | This paper | N/A |
| **Antibodies** | | |
| Alexa Fluor 488-conjugated antibodies | Jackson ImmunoResearch | Cat# 715-545-150; RRID: AB_2341099 |
| Alexa Fluor 647-conjugated antibodies | Jackson ImmunoResearch | Cat# 715-605-150; RRID: AB_2340905 |
| Alexa Fluor 647-conjugated secondary antibodies | Jackson ImmunoResearch | Cat# 715-605-150; RRID: AB_2340905 |
| Chicken polyclonal anti-EGFP | Aves Labs | Cat# GFP-1020; RRID: AB_10000240 |
| Chicken polyclonal anti-Tbr1 | Millipore | Cat# AB2261; RRID: AB_10615497 |
| Cy3-conjugated anti-mouse IgG | Jackson ImmunoResearch | Cat# 715-165-150; RRID: AB_2340813 |
| Cy3-donkey anti-human IgG | Jackson ImmunoResearch | Cat# 709-165-149; RRID: AB_2340535 |
| FITC-conjugated anti-rabbit IgG | Jackson ImmunoResearch | Cat# 711-095-152; RRID: AB_2340612 |
| Goat polyclonal anti-EGFP | Rockland | Cat# 600-101-215; RRID: AB_218182 |
| Guinea pig polyclonal anti-VGLUT1 | Millipore | Cat# ab5905; RRID: AB_2301751 |
| Mouse anti-Synapsin II | Sigma-Aldrich | Cat# S2312; RRID: AB_477499 |

| Reagent/resource | Reference or source | Identifier or catalog number |
|---|---|---|
| Mouse monoclonal anti-APP | Millipore | Cat# MAB348; RRID: AB_94882 |
| Mouse monoclonal anti-Calbindin | Swant | Cat# 300; RRID: AB_10000347 |
| Mouse monoclonal anti-GAD67 | Millipore | Cat# MAB5406; RRID: AB_2278725 |
| Mouse monoclonal anti-HA | BioLegend | Cat# 901501; RRID: AB_2565006 |
| Mouse monoclonal anti-NMDAR1 | Millipore | Cat# MAB363; RRID: AB_94946 |
| Mouse monoclonal anti-Nlgn1 | Synaptic Systems | Cat# 129 111; RRID: AB_887747 |
| Mouse monoclonal anti-PV | Swant | Cat# 235; RRID: AB_10000343 |
| Mouse monoclonal anti-Synapsin I | Synaptic Systems | Cat# 106 011; RRID: AB_2619722 |
| Mouse monoclonal anti-Synapsin II | Synaptic Systems | Cat# 106 211; RRID: AB_2619774 |
| Mouse monoclonal anti-gephyrin | Synaptic Systems | Cat# 147 011; RRID: AB_887719 |
| Mouse monoclonal anti-β-actin | Sigma-Aldrich | Cat# sc-47778; RRID: AB_476743 |
| Rabbit anti-Gephyrin | Synaptic Systems | Cat# 147002; RRID: AB_887702 |
| Rabbit anti-VGAT | Synaptic Systems | Cat# 131003; RRID: AB_887871 |
| Rabbit monoclonal anti-HA | Cell Signaling Technology | Cat# 3724; RRID: AB_1549585 |
| Rabbit monoclonal anti-c-Fos | Cell Signaling Technology | Cat# 14609; RRID: AB_2798537 |
| Rabbit polyclonal anti-Brn2 | Abcam | Cat# ab94977; RRID: AB_10615497 |
| Rabbit polyclonal anti-Calbindin | Swant | Cat# 300; RRID: AB_10000347 |
| Rabbit polyclonal anti-GABA$_A$ receptor gamma2 | Synaptic Systems | Cat# 224 003; RRID: AB_2263066 |
| Rabbit polyclonal anti-Nlgn2 | Synaptic Systems | Cat# 129 202; RRID: AB_993011 |
| Rabbit polyclonal anti-PSD-95 | Jaewon Ko's Laboratory (JK016) | RRID: AB_2722693 |
| Rabbit polyclonal anti-VGAT | Synaptic Systems | Cat# 131 003; RRID: AB_887869 |
| Rabbit polyclonal anti-VGLUT1 | Millipore | Cat# ab5905; RRID: AB_2301751 |
| Rabbit polyclonal anti-Wfs1 | Proteintech | Cat# 11558-1-AP; RRID: AB_2216046 |
| Rabbit polyclonal anti-pan-SHANK | Eunjoon Kim's Laboratory (1172) | RRID: AB_2810261 |
| Rabbit polyclonal anti-phospho-Synapsin II (Ser425) | Thermo Fisher Scientific | Cat# PA5-64855; RRID: AB_2663626 |
| Rabbit polyclonal anti-phosphoserine | Millipore | Cat# AB1603; RRID: AB_390205 |
| Rat monoclonal anti-SST | Millipore | Cat# MAB354; RRID: AB_2255365 |

| Reagent/resource | Reference or source | Identifier or catalog number |
| --- | --- | --- |
| Rabbit polyclonal anti-Iba-1 | Fujifilm Wako | Cat# 016–20001; RRID: AB_839506 |
| Mouse monoclonal anti-GFAP | Cell Signaling Technology | Cat# 3670; RRID: AB_561049 |
| Mouse monoclonal anti-NeuN | Millipore | Cat# MAB377; RRID: AB_2298772 |
| **Chemicals, enzymes and other reagents** | | |
| 18-plex TMT reagents | Thermo Fisher Scientific | Cat# A52048 |
| Bovine serum albumin (BSA) | Sigma-Aldrich | Cat# A9418 |
| Dulbecco's Modified Eagle Medium (DMEM) | Welgene | |
| Fetal bovine serum (FBS) | Tissue Culture Biologicals | |
| HEPES | Sigma-Aldrich | Cat# H3375 |
| Neurobasal Medium | Thermo Fisher Scientific | Cat# 21103049 |
| Protein A-Sepharose beads | GE Healthcare | Cat# 17-0780-01 |
| 2,2,2-tribromoethyl alcohol | Sigma-Aldrich | Cat# T48402 |
| B-27 Supplement | Thermo Fisher Scientific | Cat# 17504044 |
| Bazedoxifene acetate | MedchemExpress | Cat# HY-A0036 |
| Fulvestrant | MedchemExpress | Cat# HY-13636 |
| CNQX | Tocris | Cat# 0190 |
| CalPhos Transfection Kit | Takara | Cat# 631312 |
| D-AP5 | Tocris | Cat# 0106 |
| Triton X-100 | Sigma-Aldrich | Cat# X100 |
| Trypsin | Thermo Fisher Scientific | |
| Fast Green FCF | Sigma-Aldrich | Cat# F7252-5G |
| GlutaMax | Thermo Fisher Scientific | Cat# 35050061 |
| Glycine | Sigma-Aldrich | Cat# G8898 |
| 1.0 µm Gold Microcarriers | Bio-Rad | Cat# 1652263 |
| Ni-NTA magnetic agarose beads | Qiagen | Cat# 36113 |
| Ethylene-Diamine-Tetraacetic Acid | Invitrogen | Cat# AM9261 |
| Iron(III) chloride | Sigma-Aldrich | Cat# 701122 |
| Paraformaldehyde | Sigma-Aldrich | Cat# 158127 |
| Penicillin-streptomycin | Thermo Fisher Scientific | Cat# 15140122 |
| Poly-L-lysine | Sigma-Aldrich | Cat# P4832 |
| Polyethylene glycol | Sigma-Aldrich | Cat# 81260 |
| Protease K | Sigma-Aldrich | Cat# P2308 |
| Sodium pyruvate | Thermo Fisher Scientific | Cat# 11360070 |
| Picrotoxin | Tocris | Cat# 1128 |
| Tetrodotoxin (TTX) | Cayman Chemical | Cat# 14964 |

| Reagent/resource | Reference or source | Identifier or catalog number |
| --- | --- | --- |
| Mouse Estradiol (E2) ELISA Kit | AssayGenie | Cat# MOEB2488 |
| **Oligonucleotides and other sequence-based reagents** | | |
| ASD gene sets | PMID : 35982160; 31981491 | N/A |
| Alzheimer's gene sets | PMID: 24162737 | N/A |
| Epilepsy gene sets | PMID: 29942082 | N/A |
| Schizophrenia gene sets | PMID: 35396579 | N/A |
| **Software** | | |
| Clampfit 10.8 | Molecular Devices | https://www.moleculardevices.com |
| IGV (Integrative Genomics Viewer) | Broad Institute, Cambridge | https://igv.org/ |
| Functool 3D Fiber Tracking | GE Medical Systems | |
| GraphPad Prism | GraphPad Software | RRID: SCR_002798 |
| ImageJ | National Institutes of Health (NIH) | |
| FATHMM | BioCompute | https://fathmm.biocompute.org.uk |
| DeepSqueak | GitHub | DeepSqueak |
| MetaMorph | Molecular Devices | https://www.moleculardevices.com |
| Perseus | Max Planck Institute of Biochemistry | Version 1.6.15 |
| 3D Tractography Software | GE Medical Systems | |
| EthoVision XT 15.5 | Noldus Information Technology | |
| GenoSystem | Genologica | |
| scvi-tools | scvi-tools.org | https://scvi-tools.org |
| pCLAMP11 | Molecular Devices | https://www.moleculardevices.com |
| SIFT | A*STAR Bioinformatics Institute | https://sift.bii.a-star.edu.sg |
| Scrublet | Github | https://github.com/swolock/scrublet |
| **Instruments** | | |
| Electroporator | BTX | Model ECM 830 |
| Vibratome | Leica Biosystems | Model VT1200S |
| Confocal microscope | Carl Zeiss | Model LSM800 |
| Transmission electron microscope | FEI (Tecnai) | Model T12 Bio-TWIN |
| Liquid chromatography system | Thermo Fisher Scientific | UltiMate 3000 RSLCnano |
| MRI System (3T) | GE Medical Systems | Milwaukee, Wisconsin |
| Proteomics system | Thermo Fisher Scientific | Orbitrap Fusion Lumos Mass Spectrometer |
| Two-photon microscope | Bruker Nano, Inc. | |
| Patch pipette puller | Sutter Instruments | Model P-97 |
| Circular dichroism spectrophotometer | JASCO | Model J-1500 |

| Reagent/resource | Reference or source | Identifier or catalog number |
|---|---|---|
| Sound-attenuating chamber | SD Instruments | SR-LAB-Startle Response System |
| Ti:Sapphire laser | Spectra Physics | Model MaiTai HP |
| **Deposited Data** | | |
| Single-cell brain atlas data | PMID: 39363111 | N/A |

## Identification of cases with *MDGA1* variants

In total, 505 cases of trio exome sequencing performed in our department since 2014 were reviewed in search of *MDGA1* mutations in homozygous or compound heterozygous states, according to the haploinsufficiency of this gene. All studies were performed on patients with neurodevelopmental disorders of probable genetic origin. Two unrelated cases with compound heterozygous *MDGA1* mutations were identified.

## Genetic analysis

Whole exome sequencing was performed using genomic DNA isolated from whole blood from the proband and parents. Genomic DNA extraction was carried out from blood using the MagNa Pure 24 equipment (Roche Diagnostics). The quantity of extracted gDNA was measured with a fluorimeter (Quibit 3.0). The absorbance ratios at 260/280 and 260/230 were also studied to determine the quality of the DNA obtained, using a NanoDrop ND-2000 equipment. In addition, the integrity of the genomic DNA was analyzed by electrophoresis in 0.8% agarose gels. Libraries were prepared using the KAPA Hyper plus Kit (Roche Diagnostics) following the manufacturer's specifications and a capture enrichment protocol with specific probes (KAPA Hyper-Exome; Roche Diagnostics). Then, we performed subsequent massive parallel sequencing in a NextSeq 550 equipment (Illumina). Signal processing, base calling, alignment, and variant calling were performed with Genologica variant analysis software (GenoSystem). This software developed by Genologica contains an optimized algorithm that includes (among other steps), the following: (a) Initial quality control of the sequences; (b) Filtering the sequences by eliminating indeterminacies, adapters and low-quality areas; (c) Second quality control of the sequences; (d) Mapping on the hg19 reference genome; (e) Obtaining variants and CNVs; (f) Mapping coverage study; and (g) Annotating variants. Finally, variant prioritization was based on stringent assessments at both the gene and variant levels and taking into consideration the patient's phenotype and the associated inheritance pattern. Candidate variants were visualized using IGV (Integrative Genomics Viewer). Candidate variants were evaluated based on stringent assessments at both the gene and variant levels, taking into consideration both the patient's phenotype and the inheritance pattern. Variants were classified following the guidelines of the American College of Medical Genetics and Genomics (ACMG). A board of molecular clinical geneticists evaluated each variant classified as pathogenic, likely pathogenic, or a variant of uncertain significance, and decided which, if any, had to be reported. In every case, causal variants were discussed with the referring physician

and/or clinical geneticist. Effects of *MDGA1* missense variants were predicted using various in silico tools, including SIFT (Sorting Intolerant from Tolerant; https://sift.bii.a-star.edu.sg); FATHMM (Functional Analysis through Hidden Markov Models; https://fathmm.biocompute.org.uk); Eigen (https://www.columbia.edu/eigen); LRT (Likelihood Ratio Test) (Chun and Fay, 2009); MutationTaster (https://www.mutationtaster.org); and MVP (Missense Variant Pathogenicity prediction) (Qi et al, 2021).

## Neuroimaging

The brain MRIs entailed different sequences, including DTI and 3D-SPGRT1, regardless of the reason for the study. DTI images were obtained with a 3 T system (GE Medical System, Milwaukee, Wisconsin) by using a SS-SE echoplanar diffusion-weighted imaging (DWI) sequence (TR:12,000; FOV: 240 mm; section thickness: 3 mm, 0 spacing; matrix $128 \times 128$; bandwidth: 250; 1 NEX; diffusion encoding in 45 directions) with maximum b = 1000 sec/mm$^2$. 3D-tractography was performed in an off-line workstation by using commercially available processing software as provided by the manufacturer (Functool 3D Fiber Tracking, GE, France) based on the fiber assignment by contiguous tracking (FACT) method, achieved by connecting voxel to voxel. The threshold values were 0.3 for FA and 45° for the trajectory angles, between the regions of interest (ROIs). DTI tracts were also co-registered to the 3D-T1 weighted dataset. MRI evaluation and tractography analysis were conducted by a radiologist specializing in these techniques who was unaware of the patient's genetic diagnosis.

## Ethical procedure

The study was carried out in accordance with the Declaration of Helsinki of the World Medical Association and approved by the local ethics committees. Informed consent was obtained from each family after a full explanation of the procedures.

## Identification of *MDGA1* in a single-cell human brain atlas

We utilized the single-cell brain atlas from our previous study (Data ref: Kim et al, 2024b) to examine *MDGA1* expression. The atlas collated eight single-cell and single-nucleus datasets representing neurotypical samples from 11 developmental stages (from gestational week 7–90 years), with the data classified into 41 clusters. Among the collected data, cells and cell clusters annotated as being of poor quality in the original studies and those lacking sample information were excluded. Metadata, including sample ID, batch, platform, demographics, brain region, and postmortem interval were standardized with age information categorized into developmental stages. Gene names were standardized with respect to the NCBI and HGNC databases, and doublets detected by Scrublet version 1.8.2 (https://github.com/swolock/scrublet) were excluded. Normalization, log transformation, and selection of highly variable genes were executed using Scanpy version 1.8.2 (https://scanpy.readthedocs.io). Integration with batch effect correction was conducted using scvi-tools version 1.0.3 (https://scvi-tools.org), and clustering was conducted with the Leiden algorithm (resolution = 0.6). Primary cell types were annotated based on the expression of canonical cell type marker genes defined by the Allen Brain Institute (Hodge et al, 2019). Genes that were differentially

expressed in each cluster were identified using the Wilcoxon rank-sum test and cluster-specific genes were defined as those with an FDR < 0.05 and a $\log_2$-fold change >0.2.

## Analysis of MDGA1 expression in single-nucleus transcriptomic data from developing human neocortex

The single-nucleus RNA-sequencing data from the developing human neocortex single-nucleus multiome atlas (Data ref: Wang et al, 2025) were utilized to examine the cell-type specificity of MDGA1 expression. This dataset comprises 38 human neocortical samples collected from the prefrontal cortex and primary visual cortex across 5 developmental stages: the first trimester, second trimester, third trimester, infancy, and adolescence. The processed and annotated h5ad data were obtained from the CELLxGENE database (https://cellxgene.cziscience.com/collections/ad2149fc-19c5-41de-8cfe-44710fbada73). All data processing and visualization were performed using Scanpy version 1.8.2 (Wolf et al, 2018).

## Gene set enrichment test

Gene set enrichment tests were performed for risk genes associated with various neurological disorders. ASD risk genes selected from multiple comparisons were retrieved from Fu et al ($n = 184$) (Data ref: Fu et al, 2022) and Satterstrom et al ($n = 96$) (Data ref: Satterstrom et al, 2020) and categorized as being involved with gene expression regulation ($n = 54$) and neuronal communication ($n = 24$) as previously defined (Data ref: Satterstrom et al, 2020). Alzheimer's disease risk genes were obtained from Lambert et al ($n = 32$) (Data ref: Lambert et al, 2013). Epilepsy risk genes were identified from Heyne et al ($n = 50$) (Data ref: Heyne et al, 2018). The schizophrenia gene set was retrieved from Singh et al ($n = 32$) (Data ref: Singh et al, 2022). Genes significantly enriched for PTVs or damaging missense variants ($P$ value < 0.01) in cases and listed as bipolar disorder risk genes in the BipEx dataset (https://bipex.broadinstitute.org; $n = 58$) were selected. The developmental disorder gene set ($n = 372$) was sourced from Fu et al (Data ref: Fu et al, 2022). Intellectual disability risk genes ($n = 28$) were collected from Willsey et al (Data ref: Willsey et al, 2018). ADHD risk genes ($n = 74$) were gathered from Demontis et al (Data ref: Demontis et al, 2023). Gene identifiers for these gene sets were matched between gene sets by converting each gene alias into the corresponding HGNC gene symbol. For the enrichment test, a Fisher's exact test (one-sided) with multiple comparisons by the Bonferroni procedure was applied.

## Comparison of MDGA1 expression between ASD and control cells

The single-nucleus atlas representing cells retrieved from 15 ASD patients and 16 controls (Data ref: Velmeshev et al, 2019) was utilized. Data processing and UMAP visualization were performed using Scanpy version 1.8.2 (Wolf et al, 2018). Statistical comparisons of MDGA1 expression were performed using the two-sided Wilcoxon rank-sum test for each cell-type cluster. To account for multiple testing across cell types, FDR correction was applied using the Benjamini-Hochberg method. An overall comparison combining all cells across clusters was conducted separately for descriptive purposes and was not included in the

multiple testing correction. Summary statistics, including means, medians, standard errors of the mean (SEM), standard deviations, fold-changes, and $\log_2$-fold changes were calculated for each comparison. Effect sizes were computed using Cohen's d formula with pooled standard deviation. Sample sizes for each group and cell type were recorded, and statistical significance was determined using FDR-corrected $p$-values (FDR < 0.05) for individual cell-type comparisons (Dataset EV2). All calculations and visualizations were performed using Python SciPy and statsmodels.

## Expression vectors

pDisplay-MDGA1 V116M/A688V, pDisplay-MDGA1 Y635C/E756Q, pDisplay-MDGA1 V116M, pDisplay-MDGA1 A688V, pDisplay-MDGA1 Y635C, pDisplay-MDGA1 E756Q, pDisplay-MDGA1 T135C/N313C, pDisplay-MDGA1 V314D, and pDisplay-MDGA1 T333C/Y519C, and pDisplay-MDGA1 I336D were generated via site-directed mutagenesis (Stratagene) using the pDisplay-MDGA1 WT plasmid (Lee et al, 2013) as a template. L313-MDGA1 V116M/A688V and L313-MDGA1 Y635C/E756Q were similarly generated using the L313-MDGA1 WT construct (Lee et al, 2013) as a template. pDisplay-MDGA2 Y70C/E820Q was generated via site-directed mutagenesis using pDisplay-MDGA2 WT plasmid (Lee et al, 2013) as a template. pAAV-MDGA1 V116M/A688V and pAAV-MDGA1 Y635C/E756Q were similarly generated using pAAV-MDGA1 WT plasmid (Kim et al, 2022a) as a template. pCAGGS-His-HA-MDGA1 variants (WT, V116M/A688V, and Y635C/E756Q) were constructed by amplification of the indicated full-length sequences via polymerase chain reaction (PCR), followed by digestion with EcoRI and NotI and subcloning into the pCAGGS-His-HA vector (Addgene). pCIG2-HA-MDGA1 variants (WT, V116M/A688V and Y635C/E756Q) were constructed by amplification of the indicated full-length sequences via PCR, followed by digestion with EcoRI and XmaI, and subcloning into the pCIG2 vector (Addgene). The pCMV-IgC APP (Kim et al, 2022a) and pCMV-IgC Nlgn2 (Lee et al, 2013) constructs have been described previously.

## Antibodies

The following commercially available antibodies were used: mouse monoclonal anti-APP (clone 22C11; Millipore; Cat# MAB348; RRID: AB_94882); mouse monoclonal anti-β-actin (clone AC-74; Sigma-Aldrich; Cat# A2228; RRID: AB_476743); mouse monoclonal anti-Nlgn1 (Synaptic Systems; Cat# 129 111; RRID: AB_887747); rabbit polyclonal anti-Nlgn2 (Synaptic Systems; Cat# 129 202; RRID: AB_993011); guinea pig polyclonal anti-VGLUT1 (Millipore; Cat# AB5905; RRID: AB_2301751); mouse monoclonal anti-gephyrin (Synaptic Systems; Cat# 147 011; RRID: AB_887719); rabbit polyclonal anti-VGAT (Synaptic Systems; Cat# 131 003; RRID: AB_887869); rabbit polyclonal anti-GABA$_A$ receptor γ2 (Synaptic Systems; Cat# 224 003; RRID: AB_2263066); mouse monoclonal anti-GAD67 (clone 1G10.2; Millipore; Cat# MAB5406; RRID: AB_2278725); mouse monoclonal anti-NMDAR1 (clone 54.1; Millipore; Cat# MAB363; RRID: AB_94946); goat polyclonal anti-EGFP (Rockland; Cat# 600-101-215; RRID: AB_218182); chicken polyclonal anti-EGFP (Aves Labs; Cat# GFP-1020; RRID: AB_10000240); mouse monoclonal anti-HA (clone 16B12; BioLegend; Cat# 901501; RRID: AB_2565006); rabbit monoclonal anti-HA (clone C29F4; Cell Signaling Technology; Cat# 3724; RRID: AB_1549585); rat monoclonal anti-SST (clone YC7;

Millipore; Cat# MAB354; RRID: AB_2255365); mouse monoclonal anti-PV (clone PV235; Swant; Cat# 235; RRID: AB_10000343); rabbit polyclonal anti-Iba-1 (Fujifilm Wako; Cat# 016–20001; RRID: AB_839506); mouse monoclonal anti-GFAP (Cell Signaling Technology; Cat# 3670; RRID: AB_561049); mouse monoclonal anti-NeuN (Millipore; Cat# MAB377; RRID: AB_2298772); rabbit monoclonal anti-c-Fos (clone 9F6; Cell Signaling Technology; Cat# 14609; RRID: AB_2798537); rabbit polyclonal anti-phosphoserine (Millipore; Cat# AB1603; RRID: AB_390205); mouse monoclonal anti-Synapsin I (Synaptic Systems; Cat# 106 011; RRID: AB_2619722); mouse monoclonal anti-Synapsin II (Synaptic Systems; Cat# 106 211; RRID: AB_2619774); rabbit polyclonal anti-phospho-Synapsin II (Ser425) (Thermo Fisher Scientific; Cat# PA5-64855; RRID: AB_2663626); mouse monoclonal anti-Calbindin (Swant; Cat# 300; RRID: AB_10000347); rabbit polyclonal anti-Wfs1 (Proteintech; Cat# 11558-1-AP; RRID: AB_2216046); chicken polyclonal anti-Tbr1 (Millipore; Cat# AB2261; RRID: AB_10615497); rabbit polyclonal anti-Brn2 (Abcam; Cat# ab94977; RRID: AB_10615497); and Cy3-donkey anti-human IgG antibodies (Jackson ImmunoResearch; Cat# 709-165-149; RRID: AB_2340535). The following antibodies were previously described: rabbit polyclonal anti-pan-SHANK (1172; RRID: AB_2810261) and rabbit polyclonal anti-PSD-95 (JK016; RRID: AB_2722693) (Han et al, 2020).

### Drugs

The following drugs were commercially purchased: Bazedoxifene acetate (Cat# HY-A0036) and Fulvestrant (Cat# HY-13636) from MedChemExpress (MCE); CNQX (Cat# 0190), D-AP5 (Cat# 0106) and picrotoxin (Cat# 1128) from Tocris Bioscience; Tetrodotoxin (TTX; Cat# 14964) from Cayman Chemical; and 2,2,2-tribromoethyl alcohol (Cat# T48402) from Sigma-Aldrich.

### Cell culture

HEK293T cells were cultured in Dulbecco's Modified Eagle's Medium (DMEM; Welgene) supplemented with 10% fetal bovine serum (FBS; Tissue Culture Biologicals) and 1% penicillin-streptomycin (Thermo Fisher) at 37 °C in a humidified 5% $CO_2$ atmosphere. All procedures were performed according to the guidelines and protocols for rodent experimentation approved by the Institutional Animal Care and Use Committee of the Daegu Gyeongbuk Institute of Science and Technology (DGIST).

### Cell-surface binding assays

Ig-fusion proteins of Nlgn2, APP, and IgC alone (control) were produced in HEK293T cells. Soluble Ig-fused proteins were purified using protein A-Sepharose beads (GE Healthcare) as previously described. Bound proteins were eluted with 0.1 M glycine (pH 2.5) and immediately neutralized with 1 M Tris-HCl (pH 8.0). Transfected HEK293T cells expressing the indicated plasmids were incubated with 10 µg/ml Ig-fused proteins for 2 h at 37 °C with gentle agitation. Images were acquired using a confocal microscope (LSM800; Carl Zeiss).

### Staining for surface/intracellular protein levels

HEK293T cells were transfected with expression vectors for HA-MDGA1 WT or the indicated variants. After 48 h, cells were washed twice with PBS, fixed with 3.7% formaldehyde for 10 min at 4 °C, and blocked with 3% horse serum/0.1% bovine serum albumin (BSA; crystalline grade) in PBS for 15 min at room temperature. Surface-expressed protein was detected by staining with mouse anti-HA antibody at room temperature. After 90 min, cells were washed twice with PBS and incubated with Cy3-conjugated anti-mouse antibodies for 1 h at room temperature. Next, cells were permeabilized with PBS containing 0.2% Triton X-100 for 10 min at 4 °C and incubated with rabbit anti-HA antibody for 90 min at room temperature to label intracellularly expressed proteins, followed by FITC-conjugated anti-rabbit secondary antibodies. Images were acquired using a confocal microscope (LSM800; Carl Zeiss).

### Production of recombinant lentiviruses

Lentiviruses were produced by transfecting HEK293T cells with three plasmids–lentivirus vectors, psPAX2, and pMD2.G–at a 2:2:1 ratio. After 72 h, lentiviruses were harvested by collecting the media from the HEK293T cells and briefly centrifuging at $1000 \times g$ to remove cellular debris. Filtered media containing 5% sucrose were centrifuged at $117,969 \times g$ for 2 h; supernatants were then removed, and the virus pellet was washed with ice-cold PBS and resuspended in 80 µl PBS, as previously described (Lee et al, 2013).

### Heterologous synapse formation analyses

HEK293T cells were transfected with EGFP (control) or co-transfected with pDisplay-HA-Nlgn2, alone or together with the indicated MDGA1 variants using PEI. After 48 h, the transfected HEK293T cells were trypsinized, seeded onto day in vitro 12 (DIV12) hippocampal neurons, co-cultured for an additional 24 h, and double-immunostained on DIV13 with antibodies against EGFP, HA, and synapsin. All images were acquired using a confocal microscope (LSM800; Carl Zeiss). For quantification, the contours of transfected HEK293T cells were chosen as the region of interest (ROI). Fluorescence intensities of synaptic markers in each ROI were quantified for both red and green channels using MetaMorph software (Molecular Devices). Normalized synapse density on transfected HEK293T cells was expressed as the ratio of red to green fluorescence.

### Transfection, immunocytochemistry, confocal microscopy imaging, and analyses

Hippocampal cultured neurons were prepared from E18 rat brains cultured on coverslips coated with poly-L-lysine, and grown in Neurobasal medium supplemented with B-27 (Thermo Fisher Scientific), 0.5% FBS, 0.5 mM GlutaMax (Thermo Fisher Scientific), and sodium pyruvate (Thermo Fisher Scientific), as previously described (Um et al, 2020). Cultured neurons were co-transfected with EGFP alone or with the indicated MDGA1 variant expressing plasmids using a CalPhos Transfection Kit (Takara) at DIV5 and immunostained at DIV14. For immunocytochemistry, cultured neurons were fixed with 4% paraformaldehyde/4% sucrose, permeabilized with 0.2% Triton X-100 in phosphate-buffered saline (PBS), immunostained with the indicated primary antibodies, and detected with the indicated Cy3-, fluorescein isothiocyanate (FITC)-, or Alexa Fluor 647-conjugated secondary

antibodies (Jackson ImmunoResearch). Transfected neurons were chosen randomly, and images were acquired using a confocal microscope (LSM800; Carl Zeiss) with a ×63 objective lens; all image settings were kept constant. Z-stack images were converted to maximum intensity projection and analyzed to obtain the size, intensity, and density of puncta immunoreactivities derived from marker proteins. Quantification was performed in a blinded manner using MetaMorph software.

## Animals

All mice were maintained and handled in accordance with protocols (DGIST-IACUC-23112809-0003) approved by the Institutional Animal Care and Use Committee of DGIST under standard, temperature-controlled laboratory conditions. Mice were kept on a 12:12-h light/dark cycle (lights on at 7:00 am) and received water and food ad libitum. All experimental procedures were performed on male mice, using littermate controls without Cre expression. Conditional *Mdga1*-KO mice were previously described (Kim et al, 2022a; Perez-Garcia and O'Leary, 2016). Pregnant rats (Daehan Biolink) were used to prepare in vitro cultures of dissociated hippocampal neurons. *Mdga1*$^{Y636C/E751Q}$-KI mice were generated by Biocytogen Pharmaceuticals (Beijing, China) based on CRISPR/Cas9 approach. To generate *Mdga1*$^{Mut/+}$ mice, the candidate sgRNAs, located in the intron 9 and intron 13 of *Mdga1*, were searched by the CRISPR design tool (http://www.sanger.ac.uk/htgt/wge/) and then were screened for on-target activity using a Universal CRISPR Activity Assay (UCA™, Biocytogen Pharmaceuticals (Beijing Co., Ltd). A gene-targeting vector containing a 5' homology arm, donor fragment (containing the mutations of p.Tyr636Cys; TAC > TGC and p.Glu751Gln; GAG > CAG), and 3' homology arm was used as a template to repair the double-stranded breaks (DSBs) generated by Cas9/sgRNA. The T7 promoter sequence was added to the Cas9 or sgRNA template by PCR amplification in vitro. The Cas9 mRNA, targeting vector, and sgRNAs were co-injected into the cytoplasm of one-cell stage fertilized C57BL/6 N eggs. The injected zygotes were transferred into the oviducts of Kunming pseudopregnant females to generate F0 mice. F0 mice with the expected genotype confirmed by tail genomic DNA PCR and sequencing were mated with C57BL/6 N mice to establish germline-transmitted F1 heterozygous mice. F1 heterozygous mice were genotyped by tail genomic PCR, Southern blot, and DNA sequencing.

## Immunohistochemistry, confocal microscopy imaging, and analyses

P60 mice were anesthetized and immediately perfused, first with PBS for 3 min, and then with 4% paraformaldehyde for 5 min. Brains were dissected out, fixed in 4% paraformaldehyde overnight, and sliced into 30-μm-thick coronal sections using a vibratome (Model VT1200S; Leica Biosystems). Sections were permeabilized by incubating with 2% Triton X-100 in PBS containing 5% bovine serum albumen and 5% horse serum albumin for 30 min. For immunostaining, sections were incubated for 8–12 h at 4 °C with primary antibodies diluted in 0.1% Triton X-100 in PBS containing bovine and horse serum. The following primary antibodies were used: anti-VGLUT1 (1:300), anti-PSD-95 (1:300), anti-VGAT (1:300), anti-gephyrin (1:500), anti-PV (1:300), anti-SST (1:50), anti-c-Fos (1:400), anti-Brn2 (1:100), anti-Wfs1 (1:100), anti-Calb (1:100), anti-Iba1 (1:1000), anti-GFAP (1:300) and anti-NeuN

(1:500). Sections were washed three times in PBS and incubated with appropriate Cy3- or FITC-conjugated secondary antibodies (Jackson ImmunoResearch) for 2 h at room temperature. After three washes with PBS, sections were mounted onto glass slides (Superfrost Plus; Fisher Scientific) with VECTASHIELD mounting medium (H-1200; Vector Laboratories). Images were acquired by confocal microscopy (LSM800; Zeiss). Synaptic puncta were quantified using MetaMorph software, and their density and average area were measured. For immunohistochemical analysis of electroporated mice (P21), the mice were heart-perfused with 4% paraformaldehyde. Brains were collected, embedded in 3% low melting-temperature agarose, and sectioned at 100 μm thickness with a vibratome (VT1200S; Leica). The sections were incubated with primary antibodies (anti-EGFP, 1:2000; anti-HA, 1:1000) in PBS containing 0.1% Triton X-100, 0.1% BSA, and 2.5% goat serum, and then with fluorescent secondary antibodies (Alexa Fluor 488 and Alexa Fluor 647). Slices were mounted in Antifade mounting medium with DAPI (VECTASHIELD) and images were acquired by confocal microscopy (A1R; Nikon).

## Preparation of adeno-associated viruses and titration

HEK293T cells were co-transfected with the indicated AAV vectors and pHelper and pAAV1.0 (serotype 2/9) vectors. 72 h later, transfected HEK293T cells were collected, lysed, and mixed with 40% polyethylene glycol and 2.5 M NaCl, and centrifuged at 2000 × g for 30 min. The cell pellets were resuspended in HEPES buffer (20 mM HEPES; 115 mM NaCl, 1.2 mM CaCl$_2$, 1.2 mM MgCl$_2$, 2.4 mM KH$_2$PO$_4$) and an equal volume of chloroform was added. The mixture was centrifuged at 400 × g for 5 min, and concentrated three times with a Centriprep centrifugal filter (Millipore) at 1220 × g for 5 min each and with an Amicon Ultra centrifugal filter (Millipore) at 16,000 × g for 10 min. Before the titration of AAVs, contaminating plasmid DNA was eliminated by treating 1 μl of concentrated, sterile-filtered AAVs with 1 μl of DNase I (Sigma-Aldrich) for 30 min at 37 °C. After treatment with 1 μl of stop solution (50 mM ethylenediaminetetraacetic acid) for 10 min at 65 °C, 10 μg of proteinase K (Sigma-Aldrich) was added and AAVs were incubated for 1 h at 50 °C. Reactions were inactivated by incubating samples for 20 min at 95 °C. The final virus titer was quantified by qRT-PCR detection of EGFP sequences and subsequent reference to a standard curve generated using the pAAV-T2A-EGFP plasmid. All plasmids were purified using a Plasmid Maxi Kit (Qiagen GmbH).

## Preparation and transfection of organotypic hippocampal and prefrontal cortical slice cultures

Organotypic slice cultures from mouse hippocampus or prefrontal cortex were prepared from P2–3 mice, as described previously (Kim et al, 2022a; Ogelman et al, 2024), in accordance with the guidelines of the Institutional Animal Care and Use Committee at the University of Colorado Anschutz Medical Campus (protocol number: 721; IBC protocol number: 1305). Mice were acquired from the Jackson Laboratory (C57BL/6 N wild-type). Slices were transfected with pDisplay MDGA1 WT, pDisplay MDGA1 V116M/A688V or pDisplay MDGA1 Y635C/E756Q 13–20 days prior to two-photon imaging using biolistic gene transfer (180 psi). A total of 5 μg tdTomato and 5 μg GFP-gephyrin-intrabody or 20 μg of the indicated pDisplay MDGA1 vector were coated onto 6–7 mg gold

particles. The age of the culture was expressed as the equivalent postnatal (EP) day; postnatal day at slice culturing + days in vitro.

## Two-photon imaging

Imaging was performed at EP18–24 on transfected hippocampal CA1 pyramidal neurons or mPFC layer II/III pyramidal neurons within 40 µm of the slice surface at 30 °C in recirculating artificial cerebrospinal fluid (ACSF) (127 mM NaCl, 25 mM NaHCO$_3$, 1.25 mM NaH$_2$PO$_4$, 2.5 mM KCl, 25 mM D-glucose) aerated with 95% O$_2$/5% CO$_2$. For each neuron, image stacks (512 × 512 pixels; 0.047 µm/pixel) with 1 µm Z-steps were acquired from one segment of secondary or tertiary apical and/or basal dendrites using a two-photon microscope (Bruker Nano, Inc) with a pulsed Ti::sapphire laser (MaiTai HP; Spectra Physics) tuned to 920 nm (4–5 mW at the sample). All images are maximum projections of three-dimensional image stacks after applying a median filter (2 × 2) to the raw image data.

## Quantification of dendritic spines

All distinct protrusions emanating from the dendritic shaft, regardless of shape, were counted and measured on red fluorescent images using ImageJ (NIH). In all neurons included for analysis, overall spine density did not significantly change over the imaging session. This ensured that the density of synapses under examination was not influenced by cell health (which was always monitored), as widespread spine/filopodia-like structures can form in cases where cell health is compromised. Estimated spine volume was measured from bleed-through–corrected and background-subtracted red fluorescence intensities using the integrated pixel intensity of a boxed region of interest (ROI) surrounding the spine head, as described previously (Kim et al, 2022a).

## Quantification of gephyrin puncta

Gephyrin fluorescence intensity was calculated from bleed-through–corrected and background-subtracted green (GFP-gephyrin intrabody) fluorescence intensities using the integrated pixel intensity of a boxed region surrounding a GFP-gephyrin punctum as described previously (Kim et al, 2022a). Gephyrin enrichment in dendritic shafts was calculated by normalizing GFP-gephyrin fluorescence intensities (as described above) for each punctum to the mean GFP fluorescence intensity determined from four background ROIs on the same dendritic shaft. GFP-gephyrin enrichment was defined as a gephyrin punctum in cases where the ratio of green fluorescence from a punctum to green fluorescence from the dendritic background (G$_p$/G$_d$) was > 1. Among these GFP puncta, those with green fluorescence intensities more than two standard deviations (2 SDs) greater than the local background green fluorescence levels measured from two ROIs near the puncta were classified as true gephyrin puncta and counted in images from the green channel using ImageJ (NIH). A summary of the criteria is as follows: gephyrin puncta, G$_p$/G$_d$ > 1 and expression level > 2 SDs of background levels; no puncta, G$_p$/G$_d$ < 1 or expression level < 2 SDs of the background level.

## Stereotactic surgery and virus injections

P30 mice were anesthetized by intraperitoneal injection of a saline-based 2% Avertin solution (2,2,2-tribromoethyl alcohol dissolved in tert-amyl alcohol [Sigma-Aldrich]), and their heads were fixed in a stereotactic apparatus. Recombinant AAV virus was injected into the hippocampal CA1 region (coordinates: AP −2.1 mm, ML ± 1.3 mm, and DV 1.8 mm) with a Hamilton syringe at a flow rate of 100 nl/min (injected volume, 300 nl) using a Nanoliter 2010 Injector (World Precision Instruments). Each injected mouse was restored to its home cage for 2–4 weeks and used subsequently for immunohistochemical analyses, electrophysiological recordings, or behavioral analyses.

## Electrophysiology

### Cultured hippocampal neurons

Hippocampal cultured neurons from wild-type rat embryos were transfected with MDGA1 and MDGA2 patient mutant variants, followed by analysis at DIV13–16, as previously described (El Chehadeh et al, 2022). Using a Model P-97 pipette puller (Sutter Instruments), pipettes were pulled from borosilicate glass (0.86 mm I.D., 1.5 mm O.D.). The resistance of patch pipettes containing an external solution ranged from 3 to 6 MΩ. The internal solution included 145 mM CsCl, 0.3 mM Na-GTP, 10 mM EGTA, 10 mM HEPES, 5 mM NaCl, and 4 mM Mg-ATP, with the pH adjusted to 7.2–7.4 with CsOH and an osmolarity of 290–295 mOsm/L for mEPSCs and mIPSCs. The external solution included 130 mM NaCl, 1 mM MgCl$_2$, 2 mM CaCl$_2$, 4 mM KCl, 10 mM D-glucose, and 10 mM HEPES with the pH calibrated to 7.2–7.4 with NaOH and an osmolarity of 300–305 mOsm/L. The whole-cell recording was performed at room temperature using M-TSC manipulators (SENSAPEX). Electrophysiological data were obtained using an Axon MultiClamp 700B amplifier and pCLAMP11 software and then digitized using an Axon DigiData 1550B data capture board (Axon Instruments); mEPSCs and mIPSCs were recorded at a holding potential of −70 mV. Synaptic currents were analyzed offline using Clampfit 10.8 software (Molecular Devices). For mEPSC recordings, the external solution contained 1 µM TTX and 50 µM picrotoxin to block Na$^+$ currents and GABA$_A$ receptors, respectively. For mIPSC recordings, 1 µM TTX, 10 µM CNQX, and 50 µM D-AP5 were included in the extracellular solution to block Na$^+$ currents, AMPA receptors, and NMDA receptors, respectively.

### Acute hippocampal CA1 slices

Hippocampal slices (300 µm) were prepared from mice aged P10–12 days, P18–20 days, or 8–10 weeks. Following anesthesia with isoflurane, the mice were euthanized, and their brains were swiftly extracted and placed in a chilled solution with low calcium and high magnesium levels, which was oxygenated (95% O$_2$ and 5% CO$_2$). This solution consisted of 3.3 mM KCl, 1.3 mM NaH$_2$PO$_4$, 26 mM NaHCO$_3$, 11 mM D-glucose, 0.5 mM CaCl$_2$, 10 mM MgCl$_2$, and 211 mM sucrose. Hippocampal slices were prepared using a vibratome (VT1000S; Leica) and transferred to a storage chamber filled with oxygenated artificial cerebrospinal fluid (aCSF) containing 124 mM NaCl, 3.3 mM KCl, 1.3 mM NaH$_2$PO$_4$, 26 mM NaHCO$_3$, 11 mM D-glucose, 2 mM CaCl$_2$, and 1 mM MgCl$_2$. Slices were incubated at 30 °C for at least 60 min prior to experimentation. Subsequently, slices were transferred into the recording chamber and subjected to constant perfusion with standard aCSF that was oxygenated with a mixture of 95% O$_2$ and 5% CO$_2$. All experiments were performed at 32 °C, and slices were used within 4 h. Only cells with an access resistance (R$_a$) smaller than 30 MΩ

were analyzed. All the recordings were conducted using a Multiclamp 700B amplifier and DigiData 1550B Digitizer. To assess mIPSCs, whole cell recordings were performed from hippocampal CA1 pyramidal neurons using glass pipettes (3–5 MΩ) filled with a solution containing 145 mM CsCl, 5 mM NaCl, 10 mM HEPES, 10 mM EGTA, 4 mM Mg-ATP, and 0.3 mM Na-GTP, adjusted to pH 7.2–7.3 with CsOH. The membrane potential was clamped at $-70$ mV. mIPSCs were isolated by the external application of 50 μM D-AP5, 10 μM CNQX, and 1 μM tetrodotoxin to inhibit NMDARs, AMPARs, and $Na^+$ channels, respectively. For the measurement of evoked inhibitory postsynaptic currents (eIPSCs) and delayed asynchronous IPSCs (aIPSCs), patch pipettes were filled with an internal solution consisting of 130 mM Cs-methanesulfonate, 5 mM TEA-Cl, 8 mM NaCl, 0.5 mM EGTA, 10 mM HEPES, 4 mM Mg-ATP, 0.4 mM Na-GTP, 1 mM QX-314, and 10 mM disodium phosphocreatine, adjusted to pH 7.2–7.3 with CsOH. aIPSC recordings were carried out in aCSF where $Ca^{2+}$ was replaced by $Sr^{2+}$ (124 mM NaCl, 3.3 mM KCl, 1.3 mM $NaH_2PO_4$, 26 mM $NaHCO_3$, 11 mM D-glucose, 8 mM $SrCl_2$, and 1 mM $MgCl_2$). Cells were voltage-clamped at 0 mV. Electrical stimulation was applied using a concentric bipolar electrode (FHC), placed on the *stratum radiatum* (SR) or *stratum lacunosum moleculare* (SLM) of the hippocampal CA1 to record somatic inhibition and dendritic inhibition, respectively. eIPSCs and aIPSCs were isolated by inhibiting NMDARs and AMPARs using 50 μM D-AP5, and 10 μM CNQX. To record eIPSCs input-output (I-O) curves, electrical stimulation was applied ranging from 10 to 60 μA with 10 μA increments. Average eIPSC I-O values were measured from three consecutive sweeps. For assessing paired-pulse ratios (PPRs), pairs of stimuli were administered at a 10-Hz frequency. PPRs were calculated as the ratio of the 2nd eIPSC amplitude to the 1st eIPSC amplitude. Average PPRs were analyzed from three consecutive sweeps. To assess delayed aIPSCs, a 40-Hz train stimulation was applied. Synaptic events during the 1-s period after train stimulation were quantified and normalized to the 1-s period before the train stimulation.

## Circular dichroism spectroscopy

Far-UV circular dichroism (CD) from 200 nm to 260 nm was performed on a spectrophotometer (J-1500; JASCO) equipped with a temperature controller and a 1-mm path-length quartz cuvette. Samples of wild-type and mutant MDGA1 (0.2 mg/ml) in PBS were prepared for CD measurements. The bandwidth was 5 nm and the scanning speed was 20 nm/min with a 4-s digital integration time. All spectra were obtained by averaging five scans with buffer subtraction, and the temperature was maintained at 20 °C. Millidegrees were converted to molar residue ellipticity.

## In utero electroporation

E15.5 timed-pregnant ICR mice (Daehan Biolink) were anesthetized with isoflurane, and the uterine horns were exposed by laparotomy. Endotoxin-free plasmids (pCIG2, pCIG2-HA-MDGA1WT, pCIG2-HA-MDGA1 V116M/A688V, or pCIG2-HA-MDGA1 Y635C/E756Q; 2 μg/μl) and 0.5% Fast Green were injected into a lateral ventricle of E15.5 embryos. Electroporation was performed by placing the anode on the side of DNA injection and the cathode on the other side of the head to target cortical

progenitors. Four pulses of 40 V (50 ms duration, 500 ms interval) were applied using an electroporator (ECM 830; BTX). After electroporation, the uterine horns were returned into the maternal abdomen.

## Measurement of estradiol levels by enzyme-linked immunosorbent assay

Estradiol levels were quantified using a Mouse Estradiol (E2) ELISA Kit (AssayGenie; Cat# MOEB2488) following the manufacturer's instructions with minor modifications. Brains from $Mdga1^{Y636C/E751Q}$ KI mice were rapidly dissected, rinsed in ice-cold 1× PBS to remove excess blood, and homogenized in 20 mL of 1× PBS. Tissue homogenates were stored overnight at $\leq -20$ °C and subjected to two freeze–thaw cycles to ensure complete cell lysis. The samples were centrifuged at $5000 \times g$ for 5 min at 4 °C, and the resulting supernatants were collected for ELISA measurements. For the assay, 50 μL of each sample or standard was added to the designated wells, followed by the addition of 50 μL of Detection Reagent A working solution. The plates were sealed and incubated for 1 h at 37 °C, then washed three times with Wash Buffer. After washing, 100 μL of Detection Reagent B working solution was added to each well, and the plates were incubated for 45 min at 37 °C. A second wash step was performed five times. Subsequently, 90 μL of Substrate Solution was added to each well, and the plates were incubated for 15 min at 37 °C protected from light. The reaction was terminated by adding 50 μL of Stop Solution. Absorbance was immediately measured at 450 nm using a preheated microplate reader.

## Mouse behavioral tests

Male and female mice aged 7–9 weeks were used for all behavioral tests. Tests were conducted in a soundproofed room under dim lighting (< 5 lux). All behavioral examinations were conducted in a controlled environment, and mice were acclimated to the experimental room prior to testing. The order of testing was randomized, and experiments were conducted at consistent timeframes to minimize potential circadian influences.

### Open-field test
Mice were placed in a white acrylic open-field box (40 × 40 × 40 cm) and allowed to freely explore the environment for 10 min in dim light (< 5 lux). The distance traveled and time spent in the center zone were recorded using a top-view infrared camera and analyzed using EthoVision XT 15.5 software (Noldus).

### Y-maze test
A Y-shaped white acrylic maze with three 35-cm-long arms at a 120° angle from each other was used. Mice were introduced into the maze and allowed to explore freely for 8 min. An entry was counted when all four limbs of the mouse were within the arm. The movement of mice was recorded with a top-view infrared camera and analyzed using EthoVision XT 15.5 software.

### Novel object-recognition test
An open-field chamber was used. Mice were habituated to the chamber for 10 min. For training sessions, two identical objects were placed in the center of the chamber at regular intervals, and

mice were allowed to explore the objects for 10 min. After the training session, mice were returned to their home cages for 24 h. For test sessions, one of the two objects was replaced with a novel object, which was placed in the same position in the chamber. Mice were returned to the chamber and allowed to explore freely for 10 min. Movement was recorded using an infrared camera, and the number and duration of contacts were analyzed using EthoVision XT 15.5. The discrimination index was calculated as the difference between the time spent exploring novel and familiar objects during the test phase.

### Elevated plus-maze test

A white acrylic maze with two open arms ($30 \times 5 \times 0.5$ cm) and two closed arms ($30 \times 5 \times 30$ cm) was elevated by 75 cm above the floor. Mice were individually placed at the center of the elevated plus-maze and allowed to freely move for 5 min. The time spent in each arm and the number of arm entries were analyzed using EthoVision XT 15.5.

### Light and dark box transition test

The apparatus consisted of a roofless box divided into a closed dark chamber and a brightly illuminated chamber (350 lux). A small entrance allowed free travel between the two chambers. The time spent in each chamber and the number of transitions were analyzed.

### Three-chamber test

The testing apparatus consisted of a white acrylic box divided into three chambers ($20 \times 40 \times 22$ cm each) with small openings on the dividing walls. Wire cups were placed in the corners of both side chambers. Mice were placed in the central chamber for a 10-min habituation period. Following this time, an age-matched social conspecific was placed in the wire cup on the left side-chamber and sociability was assessed by measuring subject exploration times for the enclosed conspecific and empty cup during a 10-min session. In the final 10-min session, a new social conspecific was placed into the previously empty wire cup, and social recognition was assessed.

### Prepulse inhibition test

Mice were individually acclimated to the experimental room for a minimum of 30 min under standard housing conditions with reduced extraneous noise. Each mouse was placed in a sound-attenuating chamber equipped with a startle response measurement apparatus (SR-LAB-Startle Response System; SD Ins.), ensuring minimal movement during testing. On day 1, after a 5-min habituation period in the absence of stimuli, mice were exposed to a series of startle-eliciting stimuli (from 0 to 120 dB) to familiarize them with the startle response. On day 2, for the PPI trials, prepulse stimuli (73, 76, 79, or 82 dB) were presented before the startle-eliciting stimuli (120 dB). The order and inter-trial intervals of prepulse and startle stimuli were randomized, including control trials with no prepulse. Startle responses were recorded for each trial using PPI test software. Data analysis involved calculating the percentage PPI for each prepulse intensity using the formula: $PPI\ (\%) = (R_{pulse} - R_{pre-pulse})/R_{pulse} \times 100$ ($R_{pulse}$ is the average response magnitude in the absence of a prepulse and $R_{pre-pulse}$ the average response magnitude in the presence of a prepulse).

### Ultrasonic vocalization test

Following a 3-day period of single housing, each adult mouse was individually placed in a sound-attenuating chamber equipped with a microphone for USV recording. To elicit vocalizations, an anesthetized conspecific of the opposite sex was introduced. Interaction and vocalizations were recorded for 10 min. For pup experiments, sessions were conducted on P3, P6, P9, and P12. Each pup was temporarily separated from the dam and placed on a heat pad maintained at 370 C. USV recordings were analyzed using DeepSqueak software for the detection and categorization of ultrasonic vocalizations.

### Forced swim test

Mice were individually placed in a glass cylinder ($15 \times 30$ cm) containing water ($24 \pm 1$ °C; depth, 15 cm). Mice were forced to swim for 6 min, and the duration of immobility was measured during the final 4 min. The latency to immobility from the start of the test (delay between the start of the test and appearance of the first bout of immobility, defined as a period of at least 1 s without any active escape behavior) and duration of immobility (defined as the time not spent actively exploring the cylinder or trying to escape from it) were measured. Immobility time was defined as the time the mouse spent floating in the water without struggling, making only minor movements that were strictly necessary to maintain its head above water.

### Tail-suspension test

Mice were individually subjected to the tail suspension test, involving suspension by adhesive tape affixed approximately 1–2 cm from the tail tip. The testing apparatus consisted of a standardized setup, and mice were allowed to hang freely. The total test duration was set at 5 min during which mice were examined for despair-related behaviors. The immobility time was recorded for the final 4 min of the test. Immobility was operationally defined as the absence of limb or body movement, and the recorded duration represented the time spent suspended without exhibiting active escape behaviors or exploratory movements. Additionally, latency to immobility was measured as the period during which the mouse did not engage in any active escape behavior for at least 1 s.

### Marble burying test

Mice were individually acclimated to the experimental room for at least 30 min prior to testing. Standard mouse bedding was placed in home cages to provide a familiar environment. For the marble burying test, a 5-cm layer of fresh bedding was uniformly spread in the Plexiglas cage. Next, 15 glass marbles were evenly distributed on the bedding surface. Each mouse was introduced into the cage and allowed free exploration for a duration of 30 min. During the marble burying phase, the number of marbles buried (defined as being at least two-thirds covered by bedding) was counted and quantified.

## Negative-stain electron microscopy

Human MDGA1 full ectodomain (16 µg/ml) and the Y635C/E756Q mutant (19 µg/ml) were applied to a glow-discharged carbon-coated 300 mesh copper grids (Electron Microscopy Science) and stained with a 0.075% uranyl formate solution. Images were acquired using a Tecnai T12 Bio-TWIN transmission electron microscope equipped with an FEI Eagle 4 K charge-coupled device camera operating at 120 kV.

## Proteomics

### Peptide preparation and phosphopeptide enrichment

Hippocampi from male control, *Mdga1*-cKO, or *Mdga1*[Y636C/E751Q] KI mice (4 replicates) of adult (P60) and E18 mouse embryos were lysed with 1× SDS buffer (5% SDS and 50 mM triethylammonium bicarbonate, pH 8.5). The lysates were digested using an S-trap method with trypsin, according to the provided protocol. The digested peptides were labeled with 18-plex TMT isotopes (Thermo Fisher Scientific), dried by SpeedVac, and desalted with Pierce peptide desalting spin columns (Thermo Fisher Scientific). The combined samples were fractionated into 20 fractions by basic reverse phase liquid chromatography. 5% of the sample was used for total proteome analysis and then the remaining 95% was reserved for phosphoproteomic analysis. The remaining latter portion was combined into 10 fractions for phosphopeptide enrichment. First, Ni-NTA magnetic agarose beads were washed three times with DW and then incubated with 100 mM EDTA (pH 8.0) for 30 min on a rotator. Next, 100 mM $FeCl_3$ solution was added, and the mixture was incubated with rotation for 30 min. The prepared $Fe^{3+}$-NTA beads were washed with DW and incubated overnight with each sample in 80% ACN with 0.1% TFA, at 4 °C on a rotator. The bead-bound phosphopeptides were eluted with elution buffer (50% ACN in 1% ammonium hydroxide), promptly acidified to pH 3.5–4.0 using 10% TFA, and vacuum dried.

### LC-MS/MS and data analysis

LC-MS/MS analysis was performed using an UltiMate 3000 RSLCnano system (Thermo Fisher Scientific) coupled to a Orbitrap Fusion Lumos mass spectrometer (Thermo Fisher Scientific). Mobile phases A and B were composed of 0 and 95.0% acetonitrile containing 0.1% formic acid, respectively. The LC gradient was applied at a flow rate of 250 nL/min during 120 min for peptide separation. The Orbitrap Fusion Lumos was operated in data-dependent mode, and the MS2 scans were performed with HCD fragmentation (37.5% collision energy). MS/MS spectra were identified and quantified using the Integrated Proteomics Pipeline software with the UniProt mouse database and the following search parameters: precursor mass tolerance, 20 ppm; fragment ion mass tolerance, 200 ppm; two or more peptide assignments for protein identification at a false positive rate < 0.01; and TMT reporter ion mass tolerance, 20 ppm. For phosphopeptide identification, phosphorylation of serine, threonine, and tyrosine was set as the differential modifications, with a maximum of three additional modifications permitted. Statistical analysis was conducted using the Perseus software (version 1.6.15), and expression levels were compared using Welch's t-test ($P < 0.05$).

## Statistical analysis

No statistical method was used to pre-determine sample sizes. Rather, the sample sizes were selected based on previous studies published in the field (see Life Science Reporting Summary for references). Animals in the same litter were randomly assigned to the different treatment groups in the various experiments. All statistical analyses were performed using GraphPad Prism 7 software (RRID: SCR_002798). The normality of distributed data was determined using the Shapiro–Wilk normality test. Normally distributed data were compared using Student's *t* test or the

### The paper explained

#### Problem

The prevailing hypothesis is that an excitatory/inhibitory (E/I) imbalance underlies autism spectrum disorder (ASD). However, prior research has focused heavily on glutamatergic synapses, leaving the specific molecular mechanisms of GABAergic synapses in ASD pathology largely underexplored. This limits our comprehensive understanding of how E/I imbalance arises.

#### Results

The current study identifies MDGA1, a GABAergic synapse-specific suppressor, as a novel putative neurodevelopmental disorder biomarker, prompted by clinical observations in two ASD patients with distinct *MDGA1* missense mutations and further supported by our finding of decreased MDGA1 levels in human ASD postmortem brains. We show that these mutations uniquely affect cortical neuron migration and GABAergic synapse suppression, with transmission electron microscopy results highlighting the requirement for an intact MDGA1 extracellular domain. Using mouse models, we demonstrated that ASD-associated *MDGA1* mutations impair specific social behaviors (e.g., altered USV and acoustic startle responses) and developmental stage-specific GABAergic synapse functions in mice, most notably in males. Proteomics also revealed that these mutations are associated with developmental stage-dependent alterations in post-translational modifications of synaptic proteins, particularly presynaptic synapsin II. Crucially, bazedoxifene, an FDA-approved drug, reversed these behavioral, electrophysiological, and biochemical alterations in male *Mdga1*-KI mice.

#### Impact

This study significantly advances the understanding of ASD etiology by highlighting the critical, underexplored role of GABAergic dysfunction and demonstrating how different ASD-linked *MDGA1* substitutions uniquely affect developmental and synaptic processes. Our findings suggest a novel therapeutic strategy for ASD by demonstrating that bazedoxifene, an FDA-approved drug, can rescue communication and startle response impairments in ASD animal models. This work lays essential groundwork for developing targeted therapies for ASD and related neurodevelopmental disorders.

analysis of variance (ANOVA); non-normally distributed data were compared using the Mann–Whitney *U* test, Kruskal–Wallis test followed by Dunn's *post hoc* test, or Tukey's multiple comparison test. Outliers were identified using Grubb's test ($P < 0.05$). Dataset EV9 presents detailed statistics and numerical values. The statistical analyses are described in more detail in the Reporting Summary linked to this article.

## Data availability

The raw MS data files for total proteomics have been deposited to the MassIVE repository with identifier PXD057134 (ftp://massive-ftp.ucsd.edu/v07/MSV000096187/) for data from *Mdga1*-cKO mice and identifier PXD057136 (ftp://massive-ftp.ucsd.edu/v07/MSV000096188/) for data from *Mdga1*[Y636C/E751Q] KI mice.

The source data of this paper are collected in the following database record: biostudies:S-SCDT-10_1038-S44321-026-00402-y.

# Peer review information

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

## Acknowledgements

The authors thank Jinha Kim (DGIST, Korea) for technical assistance. This study was supported by the National Creative Research Initiative Program of the Ministry of Science and ICT (RS-2022-NR070708 to Jaewon K), the National Research Foundation of Korea (NRF) funded by the Korea Government (2022R1C1C200003499 to Seungjoon K; RS-2024-00339642 to HK; NR-2020R1C1C1003426 to J-YA; RS-2023-00264980 to S-KK; RS-2024-00399031 to JYK; and RS-2023-NR076948 to JWU), the Korea Basic Science Institute (C523400 to JYK), Institute for Basic Science (IBS-R030-C1 to HMK), and National Institutes of Health (R01MH124778, R21MH126073 and R21NS133681 to WCO).

## Author contributions

**Seungjoon Kim**: Formal analysis; Funding acquisition; Investigation; Methodology; Writing—review and editing. **Hyeonho Kim**: Formal analysis; Funding acquisition; Investigation; Methodology; Writing—review and editing. **Javier Porta Pelayo**: Formal analysis; Investigation; Methodology. **Sara Alvarez**: Formal analysis; Investigation; Methodology. **Gyubin Jang**: Formal analysis; Investigation; Methodology. **Jinhu Kim**: Formal analysis; Investigation; Methodology. **Byeongchan Kim**: Formal analysis; Investigation; Methodology. **Victoria M Hoelscher**: Formal analysis; Methodology. **Beatriz Calleja-Pérez**: Formal analysis. **Hyunsu Jung**: Formal analysis; Investigation; Methodology. **Yeji Yang**: Data curation; Formal analysis; Investigation; Methodology. **Hea Ji Lee**: Data curation; Formal analysis; Investigation; Methodology. **Jihae Lee**: Data curation; Formal analysis; Methodology. **Seoyeon Kim**: Data curation; Formal analysis; Methodology. **Mar Jiménez de la Peña**: Formal analysis. **Yelin Lee**: Methodology. **Sohye Kim**: Methodology. **Ah-reum Han**: Methodology. **Dong Sun Lee**: Formal analysis. **Sangho Ji**: Formal analysis. **Wookyung Yu**: Formal analysis; Supervision. **Ho Min Kim**: Formal analysis; Supervision; Funding acquisition. **Joon-Yong An**: Formal analysis; Supervision; Funding acquisition. **Won Chan Oh**: Formal analysis; Supervision; Funding acquisition. **Seok-Kyu Kwon**: Formal analysis; Supervision; Funding acquisition. **Jin Young Kim**: Formal analysis; Supervision; Funding acquisition. **Ji Won Um**: Supervision; Funding acquisition; Project administration. **Alberto Fernández-Jaén**: Conceptualization; Formal analysis; Supervision; Writing—review and editing. **Jaewon Ko**: Conceptualization; Formal analysis; Supervision; Funding acquisition; Investigation; Writing—original draft; Project administration; Writing—review and editing.

Source data underlying figure panels in this paper may have individual authorship assigned. Where available, figure panel/source data authorship is listed in the following database record: biostudies:S-SCDT-10_1038-S44321-026-00402-y.

## Disclosure and competing interests statement

The authors declare no competing interests.

# Expanded View Figures

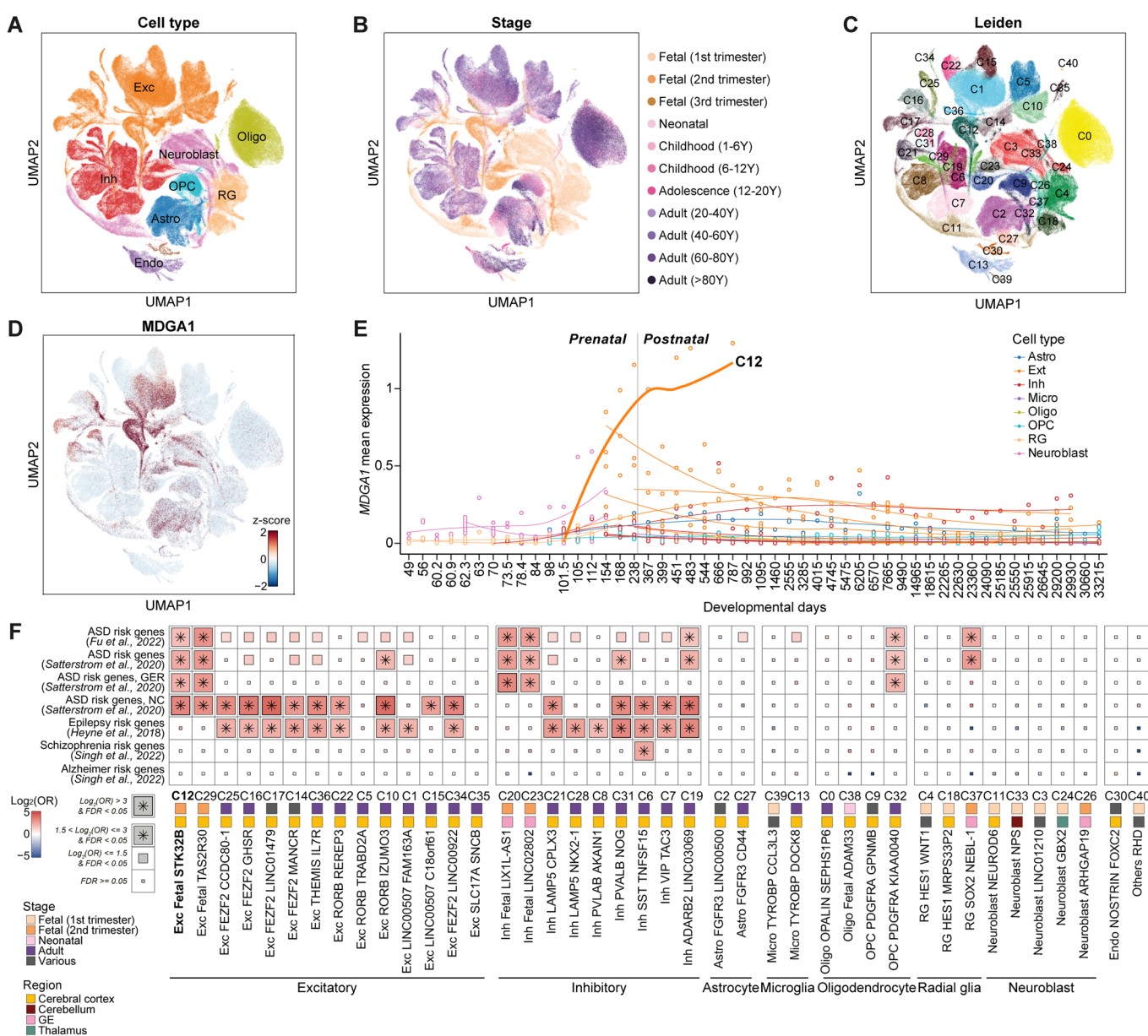

**Figure EV1. MDGA1 expression in the human brain single-cell atlas.**

(A–D) Uniform Manifold Approximation and Projection (UMAP) of the human brain single-cell atlas, colored according to primary cell types (A), developmental stages (B), Leiden clusters (C), and z-score normalized expression of *MDGA1* (D). (E) Temporal expression of MDGA1 across developmental stages. Data represent sample-wise mean log-normalized expression computed using a pseudobulk method. Analysis was restricted to clusters with ≥ 4600 cells (C0–C22). (F) Gene set enrichment analysis between neurological disorder genes and differentially expressed genes in each cluster. Statistical significance was determined using a Fisher's exact test. Significant enrichment was defined as a False Discovery Rate (FDR) < 0.05 and a log2 (odds ratio) > 1.5.

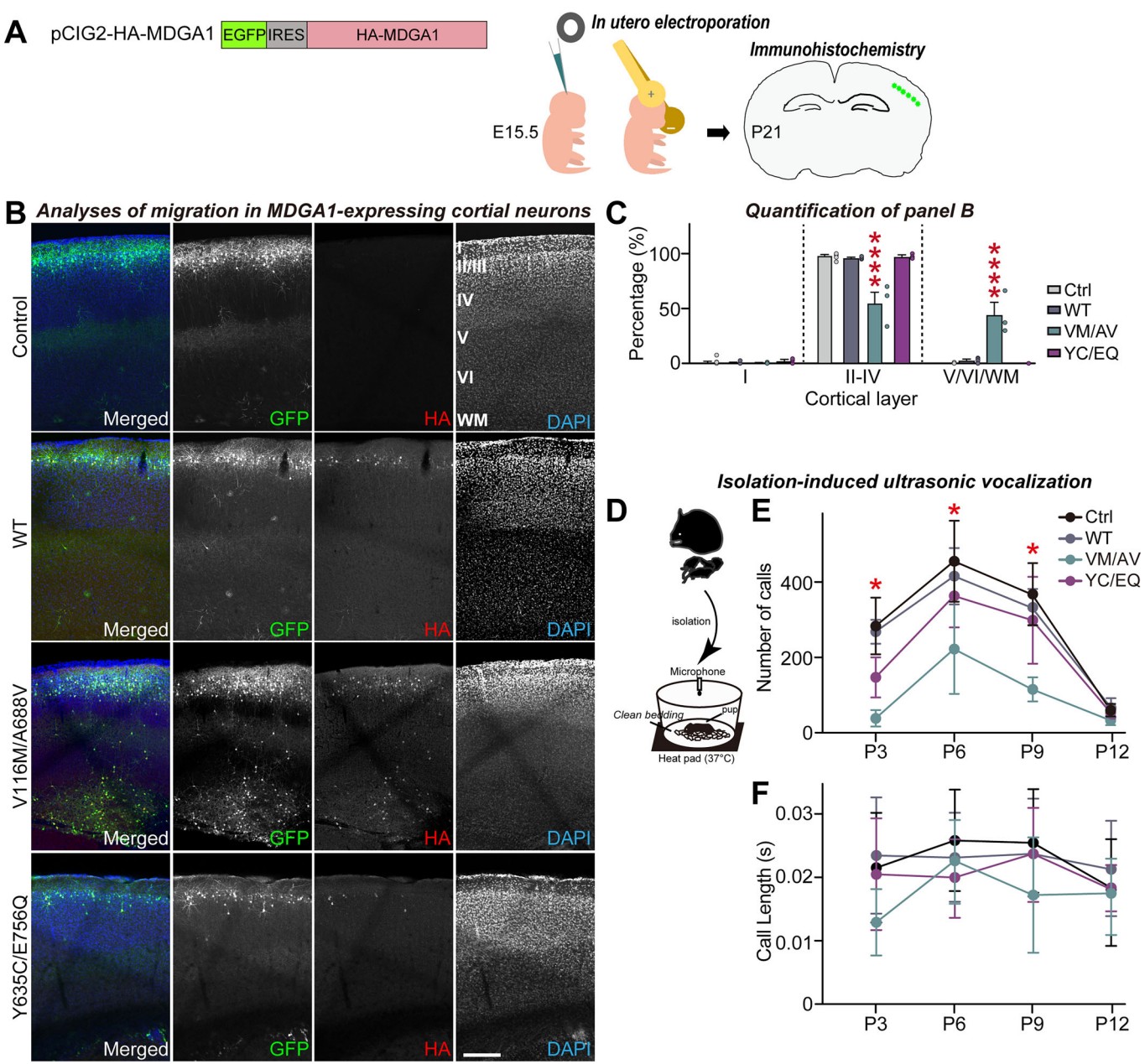

**Figure EV2. Analysis of cortical neuronal migration and ultrasonic vocalization in mice expressing ASD-associated MDGA1 variants.**

(A) Schematic of the in utero electroporation. The indicated plasmids (control or HA-tagged MDGA1 variants [HA-MDGA1-IRES-EGFP]) were electroporated in utero at E15.5 and neuronal migration was analyzed by immunohistochemistry using confocal microscopy at P21. (B, C) Analysis of the migration trajectory of cortical neurons expressing the indicated MDGA1 variants. Expression of the V116M/A688V variant impaired neuronal migration compared to that seen in control neurons or those expressing MDGA1 WT. Data are presented as means ± SEMs ('n' denotes number of images/mice; control, $n = 18/9$; WT, $n = 10/5$; V116M/A688V, $n = 6/3$; and Y635C/E756Q, $n = 5/3$; $^{****}P < 0.0001$ (II-IV), $^{****}P < 0.0001$ (V/VI/WM); Mann–Whitney $U$ test). Scale bar, 500 μm. (D) Schematic of the experimental setup for pup isolation and USV recording. USVs were measured at P3, P6, P9, and P12 from WT pups electroporated in utero with control or MDGA1 variant plasmids. (E) Number of USV calls recorded from pups expressing the control or the indicated MDGA1 variant at P3, P6, P9 and P12. Data are presented as means ± SEMs ($n = 7$–9 pups/group; asterisks (*) denote significant differences between control and the VM/AV group; $^*P = 0.0134$ (P3), $^*P = 0.0338$ (P6), $^*P = 0.0385$ (P9); nonparametric Kruskal–Wallis test followed by Dunn's *post hoc* test). (F) Duration of USV calls recorded at the same time points. No significant differences were observed among groups. Data are presented as means ± SEMs ($n = 7$–9 pups/group).

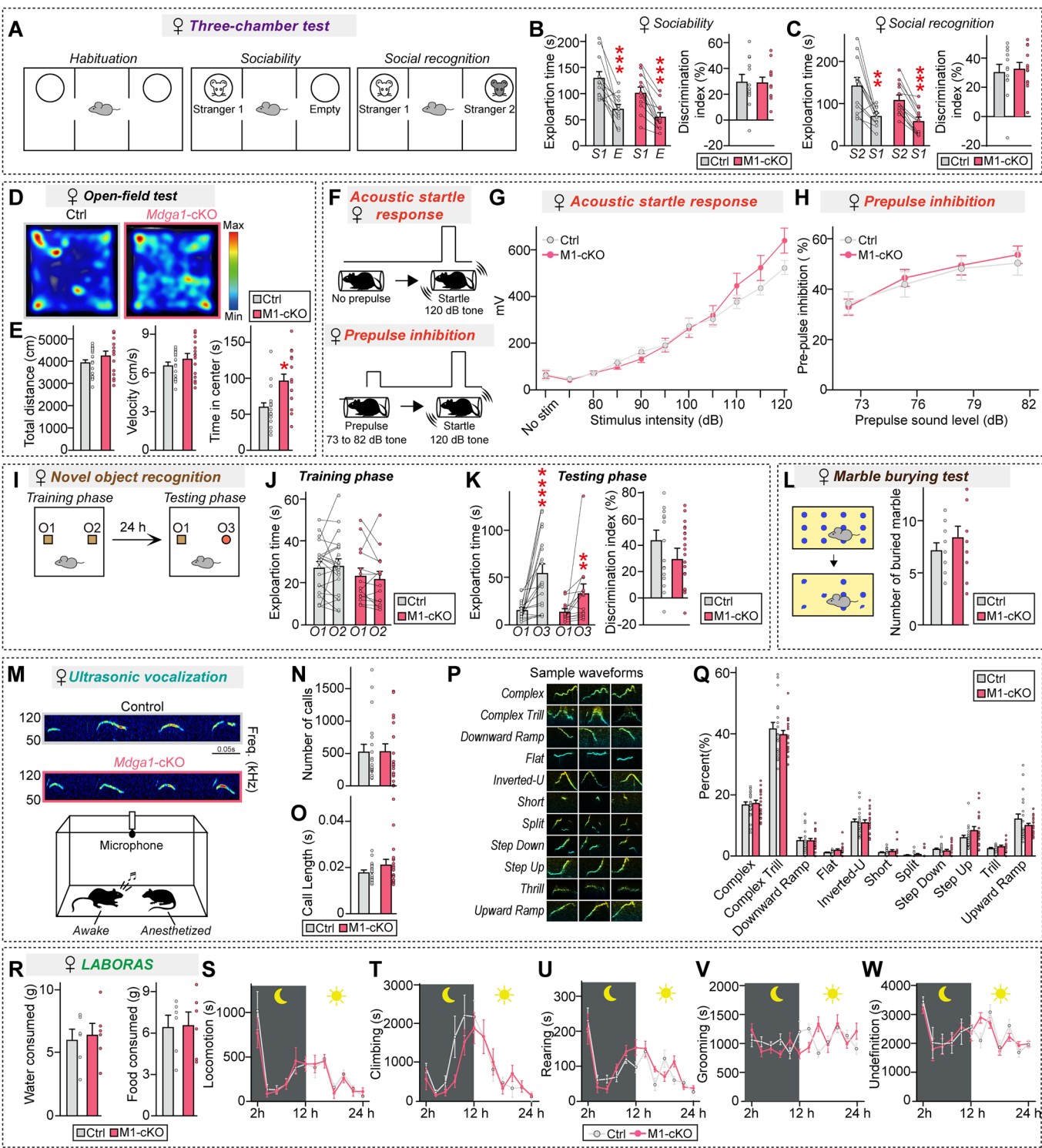

**Figure EV3.  Analysis of behaviors of adult female *Mdga1*-cKO mice.**

(A–C) Three-chamber test results showing a schematic of the test (A) and the exploration time and discrimination index during the sociability (B; stranger 1 vs. empty) and social recognition (C; stranger 1 vs. stranger 2) phases for female control and *Mdga1*-cKO mice. *Mdga1*-cKO females showed normal sociability and intact social recognition, as indicated by comparable exploration times and discrimination indices relative to controls. Data are presented as means ± SEMs ($n = 12$ mice/group; (B) ***$P = 0.0010$ (Ctrl), ***$P = 0.0005$ (M1-cKO); (C) ***$P = 0.0015$ (Ctrl), ***$P = 0.0005$ (M1-cKO); Mann–Whitney $U$ test). (D, E) Open-field test results showing total distance traveled, velocity, and time spent in the center for female control and *Mdga1*-cKO mice. *Mdga1*-cKO mice exhibited increased time spent in the center zone, while total distance traveled and velocity were comparable to those of controls. Data are presented as means ± SEMs ($n = 15$–20 mice/group; (E) *$P = 0.0105$; Mann–Whitney $U$ test). (F) Schematic of the prepulse inhibition (PPI) test. Mice were exposed to a prepulse sound (73–82 dB) followed by a startle pulse (120 dB), and the inhibition of the startle response was measured. (G) Acoustic startle response in female control and *Mdga1*-cKO mice. Data are presented as means ± SEMs ($n = 17$–18 mice/group). (H) PPI of the acoustic startle response in female control and *Mdga1*-cKO mice. Data are presented as means ± SEMs ($n = 14$–17 mice/group). (I–K) Novel object recognition test results showing a schematic of the test (I) and the exploration time and discrimination index during the training (J) and testing phases (K) for female control and *Mdga1*-cKO mice. Quantification of exploration times during training (J) and testing (K) revealed no significant differences in the discrimination index between groups, indicating that recognition memory was intact in *Mdga1*-cKO mice. Data are presented as means ± SEMs ($n = 13$–18 mice/group; (K) ****$P < 0.0001$ (Ctrl), **$P = 0.0034$ (M1-cKO); Wilcoxon matched-pairs signed rank test). (L) Marble burying test results showing the number of marbles buried by female control and *Mdga1*-cKO mice. The number of buried marbles was not significantly different between *Mdga1*-cKO and control mice, suggesting that there was no change in repetitive behavior. Data are presented as means ± SEMs ($n = 11$–14 mice/group). (M–Q) USV results showing a representative sonogram (M) and the number of calls (N) and call length (O) for female control and *Mdga1*-cKO mice. Representative waveforms (P) serve as standardized reference samples for each USV category and are utilized across figures (e.g., Fig. EV4P) as a consistent visual index for classification criteria. Distribution of different call types (Q) is shown. No significant differences in call-type proportions were observed, suggesting that vocal communication was intact. Data are presented as means ± SEMs ($n = 18$–20 mice/group; Mann–Whitney $U$ test). (R–W) Quantification of spontaneous behaviors using the LABORAS system over a 24-h period. No significant differences were observed in food and water intake (R), locomotion (S), climbing (T), rearing (U), grooming (V), or undefined behaviors (W) between *Mdga1*-cKO and control mice. Data are presented as means ± SEMs ($n = 6$ mice/group).

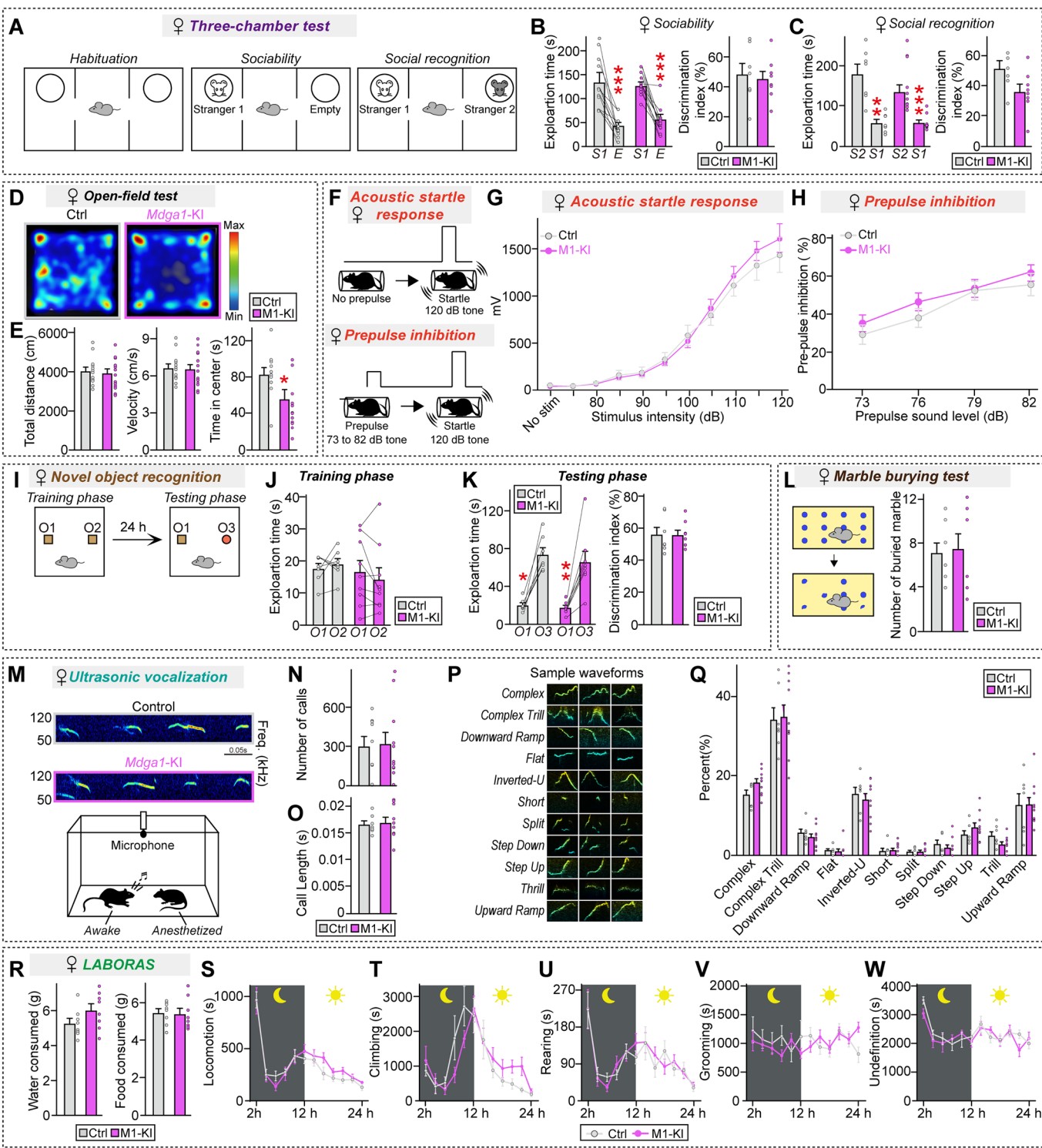

◀ **Figure EV4.   Analysis of behaviors of adult female *Mdga1*^Y636C/E751Q^ KI mice.**

(A–C) Three-chamber test results showing a schematic of the test (**A**) and the exploration time and discrimination index during the sociability (**B**; stranger 1 vs. empty) and social recognition (**C**; stranger 1 vs. stranger 2) phases for female control and *Mdga1*^Y636C/E751Q^ KI mice. *Mdga1*^Y636C/E751Q^ KI females showed normal sociability and intact social recognition, as indicated by comparable exploration times and discrimination indices relative to those of controls. Data are means ± SEMs ($n = 11$–13 mice/group; (**B**) ***$P = 0.0010$ (Ctrl), ***$P = 0.0002$ (M1-KI); (**C**) **$P = 0.0068$ (Ctrl), ***$P = 0.0005$ (M1-KI); Wilcoxon matched-pairs signed rank test). (**D, E**) Open-field test results showing total distance traveled, velocity, and time spent in the center for female control and *Mdga1*^Y636C/E751Q^ KI mice. *Mdga1*^Y636C/E751Q^ KI mice exhibited decreased time spent in the center zone, while the total distance traveled and velocity were comparable to those of controls. Data are presented as means ± SEMs ($n = 11$–13 mice/group; (**E**) *$P = 0.0105$, Mann–Whitney $U$ test). (**F–H**) Schematic (**F**) of the prepulse inhibition (PPI) test. Mice were exposed to a prepulse sound (73–82 dB) followed by a startle pulse (120 dB), and the inhibition of the startle response was measured. Acoustic startle response (**G**) in female control and female *Mdga1*^Y636C/E751Q^ KI mice. Data are presented as means ± SEMs ($n = 11$–14 mice/group). PPI of the acoustic startle response (**H**) in female control and *Mdga1*^Y636C/E751Q^ KI mice. Data are presented as means ± SEMs ($n = 11$–14 mice/group). (**I–K**) Novel object recognition test results showing a schematic of the test (**I**) and the exploration time and discrimination index during the training (**J**) and testing phases (**K**) for female control and *Mdga1*^Y636C/E751Q^ KI mice. Quantification of exploration times during training (**J**) and testing (**K**) revealed no significant difference in the discrimination index between groups, indicating that recognition memory was intact in *Mdga1*^Y636C/E751Q^ KI mice. Data are presented as means ± SEMs ($n = 7$–9 mice/group; (**K**) *$P = 0.0156$ (Ctrl), **$P = 0.0078$ (M1-KI); Wilcoxon matched-pairs signed rank test). (**L**) Marble burying test results showing the number of marbles buried by female control and female *Mdga1*^Y636C/E751Q^ KI mice. The number of buried marbles was not significantly different between *Mdga1*^Y636C/E751Q^ KI and control mice, suggesting that there was no change in repetitive behavior. Data are presented as means ± SEMs ($n = 8$ mice/group). (**M–Q**) USVs recorded from adult female mice under social isolation. Representative sonograms (**M**), quantification of USV call number (**N**) and duration (**O**), and representative waveforms (**P**) are shown. Note that the waveforms in (**P**) serve as standardized reference samples for each USV category and are utilized across figures (e.g., Fig. EV3P) as a consistent visual index for classification criteria. Distribution of call types (**Q**) was assessed; no significant differences in call-type proportions were observed, suggesting that vocal communication was intact. Data are presented as means ± SEMs ($n = 16$–18 mice/group). (**R–W**) Quantification of spontaneous behaviors using the LABORAS system over a 24-h period. No significant differences were observed in food and water intake (**R**), locomotion (**S**), climbing (**T**), rearing (**U**), grooming (**V**), or undefined behaviors (**W**) between *Mdga1*^Y636C/E751Q^ KI and control mice. Data are presented as means ± SEMs ($n = 8$ mice/group).

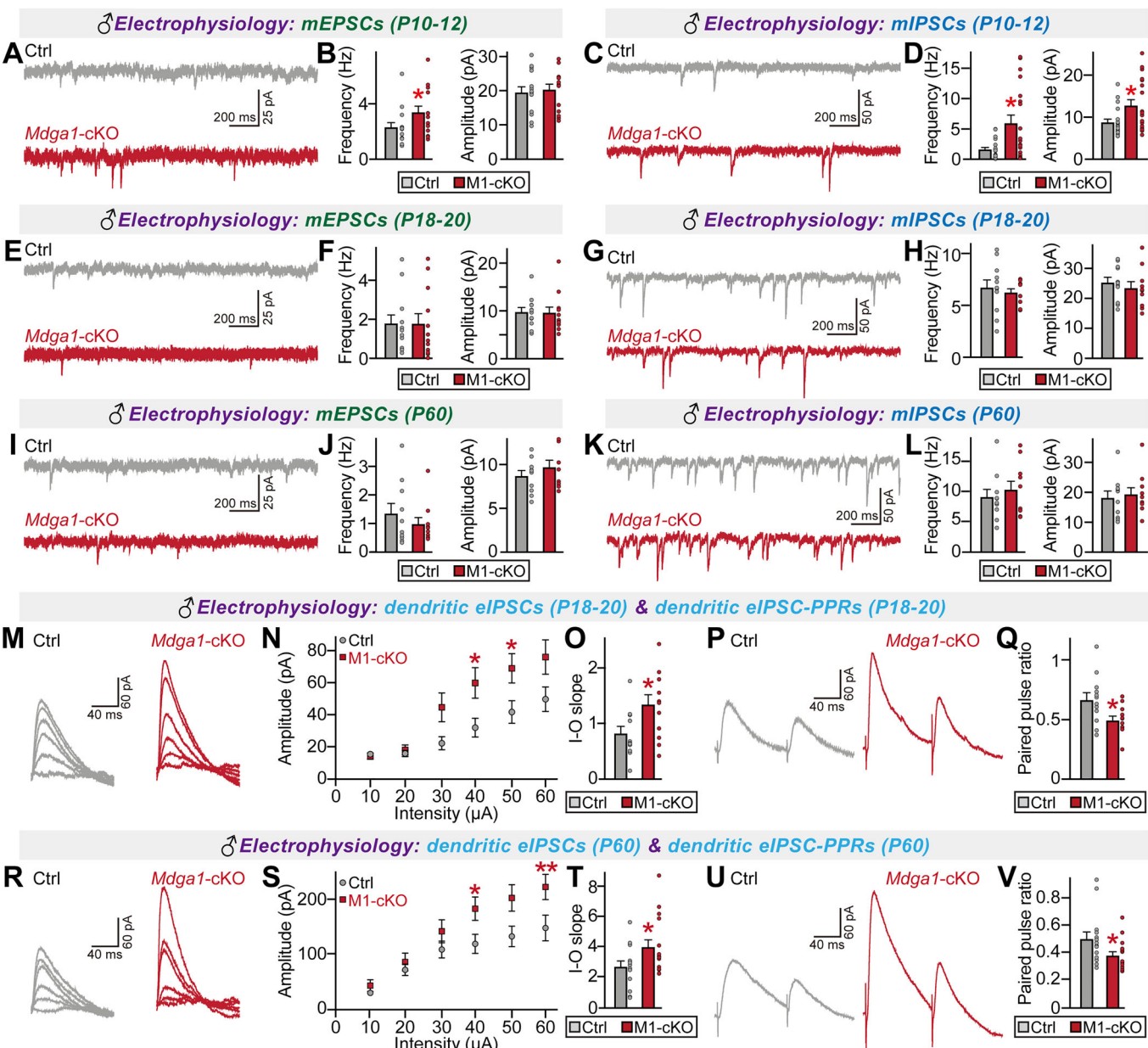

◀ **Figure EV5.  Analysis of electrophysiological properties in the hippocampal CA1 pyramidal neurons of various aged male *Mdga1*-cKO mice.**

(A, B) Representative traces (A) and quantification (B) of mEPSC frequency and amplitude recorded from CA1 pyramidal neurons of P10–12 male mice. *Mdga1*-cKO neurons showed significantly increased mEPSC frequency, while amplitude remained unchanged. Data are means ± SEMs (control, $n = 15/6$; *Mdga1*-cKO, $n = 14/6$; *P = 0.0459, Mann–Whitney *U* test). (C, D) Representative traces (C) and quantification (D) of mIPSCs from P10–12 CA1 pyramidal neurons. *Mdga1*-cKO mice exhibited increased mIPSC frequency and amplitude. Data are means ± SEMs (control, $n = 19/6$; *Mdga1*-cKO, $n = 19/6$; (frequency) *P = 0.0233, (amplitude) *P = 0.0215; Mann–Whitney *U* test). (E, F) Representative traces (E) and quantification (F) of mEPSCs recorded from CA1 pyramidal neurons of P18–20 male mice. No significant differences were observed between genotypes. Data are means ± SEMs (control, $n = 12/4$; *Mdga1*-cKO, $n = 12/4$). (G, H) Representative traces (G) and quantification (H) of mIPSCs recorded from CA1 pyramidal neurons of P18–20 male mice. No significant differences were observed between genotypes. Data are means ± SEMs (control, $n = 11/4$; *Mdga1*-cKO, n = 10/4). (I, J) Representative traces (I) and quantification (J) of mEPSCs recorded from CA1 pyramidal neurons of P60 male mice. No significant differences were observed between genotypes. Data are means ± SEMs ($n = 10/4$ per group). (K, L) Representative traces (K) and quantification (L) of mIPSCs recorded from CA1 pyramidal neurons of P60 male mice. No significant differences were detected between genotypes. Data are means ± SEMs ($n = 10/4$ per group). (M–O) Representative traces (M) and averages of dendritic eIPSCs (N, O) from CA1 pyramidal neurons of male control and *Mdga1*-cKO mice at P18–20. *Mdga1*-cKO mice exhibited a significant increase in eIPSC amplitude. Data are presented as means ± SEMs (control, $n = 12/4$; *Mdga1*-cKO, $n = 12/4$; (N) (40 μA) *P = 0.0402, (50 μA) *p = 0.0449; (O) *p = 0.0242; Mann–Whitney *U* test). (P, Q) Representative traces (P) and averages of paired-pulse ratios (PPRs) of dendritic IPSCs (Q) from CA1 pyramidal neurons of male control and *Mdga1*-cKO mice at P18–20. *Mdga1*-cKO mice showed a significant reduction in PPRs. Data are presented as means ± SEMs (control, $n = 13/4$; *Mdga1*-cKO, $n = 13/4$; *P = 0.0387; Mann–Whitney *U* test). (R–T) Representative traces (R) and averages of dendritic eIPSCs (S, T) from CA1 pyramidal neurons of male control and *Mdga1*-cKO mice at P60. eIPSC amplitude was significantly increased in *Mdga1*-cKO mice. Data are presented as means ± SEMs (control, $n = 13/5$; *Mdga1*-cKO, $n = 16/5$; (S) (40 μA) *P = 0.0435, (60 μA) **P = 0.0066; (T) *P = 0.0499; Mann–Whitney *U* test). (U, V) Representative traces (U) and averages of dendritic eIPSC-PPRs (V) from CA1 pyramidal neurons of male control and *Mdga1*-cKO mice at P60. *Mdga1*-cKO mice showed significantly decreased dendritic eIPSC-PPRs. Data are presented as means ± SEMs (control, $n = 14/5$; *Mdga1*-cKO, $n = 16/5$; *P = 0.0267; Mann–Whitney *U* test).

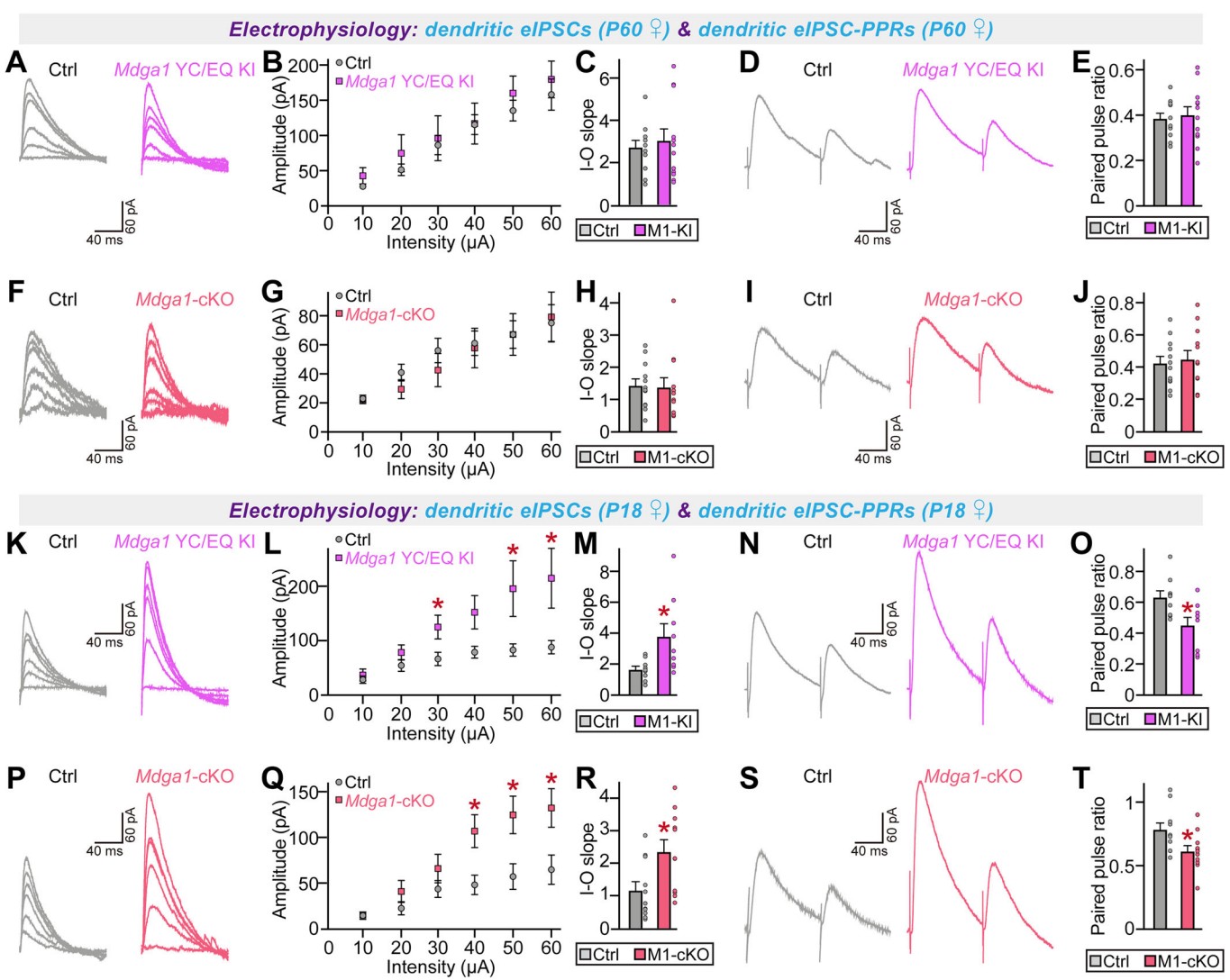

**Figure EV6. Analysis of synchronous GABAergic evoked synaptic transmission in the hippocampal CA1 pyramidal neurons of adult female *Mdga1*-cKO and *Mdga1*[Y636C/E751Q] KI mice.**

(A–E) Representative traces (A, D), input-output (I-O) curves (B, C), and paired-pulse ratios (PPRs) (E) of dendritic eIPSCs recorded from CA1 pyramidal neurons in adult female control and *Mdga1*[Y636C/E751Q] KI mice. No significant differences were observed. Data are presented as means ± SEMs ($n = 12$ cells/group). (F–J) Representative traces (F, I), I-O curves (G, H), and PPRs (J) of dendritic eIPSCs recorded from CA1 pyramidal neurons in adult female control and *Mdga1*-cKO mice. No significant differences were observed. Data are presented as means ± SEMs ($n = 12$ cells/group). (K–O) Representative traces (K, N), I-O curves (L, M), and PPRs (O) of dendritic eIPSCs recorded from CA1 pyramidal neurons in P18 female control and *Mdga1*[Y636C/E751Q] KI mice. *Mdga1*[Y636C/E751Q] KI mice showed significantly increased eIPSC amplitude (M) and decreased PPRs (O). Data are presented as means ± SEMs ($n = 9$ cells/group; (L) (30 μA) *$P = 0.0333$, (50 μA) *$P = 0.0333$, (60 μA) *$P = 0.0151$; (M) *$P = 0.0142$; (O) *$P = 0.0188$; Mann–Whitney $U$ test). (P–T) Representative traces (P, S), I-O curves (Q, R), and PPRs (T) of dendritic eIPSCs recorded from CA1 pyramidal neurons in P18 female control and *Mdga1*-cKO mice. *Mdga1*-cKO mice exhibited significantly increased eIPSC amplitude (Q) and decreased PPRs (T). Data are presented as means ± SEMs ($n = 11$ cells/group; (Q) (40 μA) *$P = 0.0199$, (50 μA) *$P = 0.0158$, (60 μA) *$P = 0.0135$; (R) *$P = 0.0192$; (T) *$P = 0.0225$; Mann–Whitney $U$ test).

