## [Peer Review File · EMBO Molecular Medicine]

Bazedoxifene reverses sexually dimorphic autistic-like abnormalities in biallelic MDGA1-mutant mice

Seungjoon Kim, Hyeonho Kim, Javier Pelayo, Sara Alvarez, Gyubin Jang, Jinhu Kim, Byeongchan Kim, Victoria Hoelscher, Beatriz Calleja-Perez, Hyunsu Jung, Yeji Yang, Hea Ji Lee, Jihae Lee, Seoyeon Kim, Mar Jimenez la Pena, Yelin Lee, Sohye Kim, Ah-reum Han, Dong Sun Lee, Sangho Ji, Wookyung Yu, Ho Min Kim, Joon-Yong An, Won Chan Oh, Seok-Kyu Kwon, Jin Young Kim, Ji Won Um, Alberto Fernandez-Jaen, and Jaewon Ko

Corresponding authors: Jaewon Ko (jaewonko@dgist.ac.kr) , Alberto Fernandez-Jaen (aferjaen@telefonica.net)

Review Timeline:

Submission Date:	9th Jun 25
Editorial Decision:	15th Jul 25
Revision Received:	17th Dec 25
Editorial Decision:	10th Feb 26
Revision Received:	12th Feb 26
Accepted:	24th Feb 26

Editor: Jingyi Hou

Transaction Report:

15th Jul 2025

Dear Prof. Ko,

Thank you again for submitting your work to EMBO Molecular Medicine. We have now received the reports from the three reviewers and as you will see below, the reviewers think that the study is potentially interesting. They raise however a series of concerns, which we would ask you to convincingly address in a revision.

The reviewers' recommendations are generally clear, so I won't reiterate them here. All the issues raised must be carefully addressed. In particular, the manuscript would benefit from improved clarity and coherence in its presentation, and any overstatements should be avoided. Several points-especially those raised in Referee #3's comments #1 and #2-warrant deeper mechanistic exploration. If available, we also encourage the inclusion of patient-derived samples, as suggested by Referee #2.

We would welcome the submission of a revised version within six months for further consideration. As you may already know, our editorial policy allows in principle a single round of major revision, and it is therefore essential to provide responses to the referees' comments that are as complete as possible.

EMBO Molecular Medicine has a "scooping protection" policy, whereby similar findings that are published by others during review or revision are not a criterion for rejection. Should you decide to submit a revised version, I do ask that you get in touch after six months if you have not completed it, to update us on the status.

Please also contact us as soon as possible if similar work is published elsewhere. If other work is published, we may not be able to extend the revision period beyond six months.

I look forward to receiving your revised manuscript.

Please use this link to login to the manuscript system and submit your revision: <https://embomolmed.msubmit.net/cgi-bin/main.plex>

Kind regards,
Jingyi

Jingyi Hou
Senior Editor
EMBO Molecular Medicine

We require:

- 1) A .docx formatted version of the manuscript text (including legends for main figures, EV figures and tables). Please make sure that the changes are highlighted to be clearly visible.
- 2) Individual production quality figure files as .eps, .tif, .jpg (one file per figure). For guidance, download the 'Figure Guide PDF': (<https://www.embopress.org/page/journal/17574684/authorguide#figureformat>).
- 3) A .docx formatted letter INCLUDING the reviewers' reports and your detailed point-by-point responses to their comments. As part of the EMBO Press transparent editorial process, the point-by-point response is part of the Review Process File (RPF), which will be published alongside your paper.
- 4) A complete author checklist, which you can download from our author guidelines (<https://www.embopress.org/page/journal/17574684/authorguide#submissionofrevisions>). Please insert information in the checklist that is also reflected in the manuscript. The completed author checklist will also be part of the RPF.

6) It is mandatory to include a 'Data Availability' section after the Materials and Methods. Before submitting your revision, primary datasets produced in this study need to be deposited in an appropriate public database, and the accession numbers and database listed under 'Data Availability'. Please remember to provide a reviewer password if the datasets are not yet public (see <https://www.embopress.org/page/journal/17574684/authorguide#dataavailability>).

.

12) Author contributions: You will be asked to provide CRediT (Contributor Role Taxonomy) terms in the submission system. These replace a narrative author contribution section in the manuscript.

13) A Conflict of Interest statement should be provided in the main text.

14) Every published paper now includes a 'Synopsis' to further enhance discoverability. Synopses are displayed on the journal webpage and are freely accessible to all readers. They include a short stand first (maximum of 300 characters, including space) as well as 2-5 one-sentences bullet points that summarizes the paper. Please write the bullet points to summarize the key NEW findings. They should be designed to be complementary to the abstract - i.e. not repeat the same text. We encourage inclusion of key acronyms and quantitative information (maximum of 30 words / bullet point). Please use the passive voice. Please attach these in a separate file or send them by email, we will incorporate them accordingly.

Please also suggest a visual abstract to illustrate your article as a PNG file 550 px wide x 300-600 px high.

15) All Materials and Methods need to be described in the main text using our 'Structured Methods' format. According to this format, the Methods section includes a Reagents and Tools Table (listing key reagents, experimental models, software and relevant equipment and including their sources and relevant identifiers) followed by a Methods and Protocols section describing the methods, ideally using a step-by-step protocol format. The aim is to facilitate adoption of the methodologies across labs.

Please download and fill our Reagents and Tools Table template (.docx), which you can find in our author guidelines: <https://www.embopress.org/page/journal/17574684/authorguide#structuredmethods>

**** Reviewer's comments ****

Referee #1 (Comments on Novelty/Model System for Author):

Mouse models are adequate in phenotyping at molecular, cellular, synaptic, and systems levels.

Referee #1 (Remarks for Author):

Researchers identified two unrelated children with autism spectrum disorder who carry different pairs of rare, compound heterozygous missense variants in MDGA1 (Val116Met + Ala688Val and Tyr635Cys + Glu756Gln) and showed, through structural modelling and biochemical assays, that the Tyr635Cys/Glu756Gln pair destabilises the characteristic triangular ectodomain of MDGA1 without altering surface expression or ligand binding. In cultured neurons the destabilising variant abolishes MDGA1's normal ability to restrain GABAergic synapse maturation, whereas the Val116Met/Ala688Val pair unexpectedly reduces excitatory synapse density and, when over expressed in utero, derails radial migration of cortical pyramidal neurons and depresses pup ultrasonic vocalizations. Conditional deletion of Mdga1 in mouse forebrain excitatory neurons and knock in substitution of Tyr636Cys/Glu751Gln each produce male specific autistic like behaviours, including blunted maternal separation and courtship USVs, impaired sociability or prepulse inhibition, and exaggerated startle responses, while females remain largely spared. Electrophysiology reveals that both mutants amplify dendritic GABAergic inhibition in CA1 and that hippocampal phosphoproteomics convergently show hypophosphorylation of presynaptic Synapsin II, pointing to a shared loss of function pathway. A single systemic dose of the brain penetrant selective oestrogen receptor modulator bazedoxifene restores Synapsin II phosphorylation, normalises GABAergic input strength and rescues vocal, startle and sensorimotor gating phenotypes in male knock in mice. Together, the work establishes MDGA1 dysfunction as a causal, sexually dimorphic synaptopathy that links inhibitory synapse dysregulation to core ASD behaviours and shows that it is pharmacologically reversible.

This is the first study to connect biallelic MDGA1 variants with autism and to dissect their structural, synaptic and behavioural consequences across patient data, cell culture and multiple mouse lines, thereby expanding the roster of inhibitory "synaptopathy" genes underlying ASD and illuminating why many phenotypes preferentially afflict males. Demonstrating full behavioural rescue with an FDA approved estrogen pathway drug makes the findings immediately translatable, suggests a mechanistic basis for the male bias in ASD, and provides a timely proof of concept for precision therapies that target sex dependent synaptic vulnerabilities. The manuscript is impressively comprehensive, with convergent data that are novel, rigorous and timely. I have only the following few minor comments.

1. The authors need to include standardized diagnostic scores (e.g., ADOS 2, Vineland) for the two autistic individuals to strengthen genotype phenotype interpretation.

2. I wonder if the authors have considered to measure MDGA1 protein levels in patient derived cells or plasma to help link human variant burden to molecular deficit. If not, the rationale not doing so should be clarified.

3. The authors tried bazedoxifene experiments in an acute-treatment format. I wonder if they thought about trying chronic-treatment experiments and, if not, they need to explain why they chose the acute treatments.
4. Supplementary Fig. 12 shows decreased number of total USV calls in the VM/AV group. I wonder if the duration of each USV call, or the total duration of USV calls, is also changed.

Referee #2 (Comments on Novelty/Model System for Author):

1. The layout and presentation of the figures need improvement for better clarity and interpretation.
2. The chosen mouse model does not convincingly demonstrate strong face validity for the condition studied.
3. Based on the data presented, the medical relevance remains unclear-aside from identifying two individuals with mutations in the MDGA1 gene. The associated complex clinical features may only be partially recapitulated in the mouse model.

Referee #2 (Remarks for Author):

General Comments

I appreciate the considerable amount of work and technical expertise that went into this manuscript; the authors have conducted a wide range of experiments using diverse methodologies. However, in its current form, the manuscript feels fragmented, and the narrative lacks coherence. As a result, the central message becomes somewhat diluted rather than reinforced. To improve readability and overall impact, a clearer and more integrated storyline is strongly recommended.

Additionally, I recommend replacing the term "retardation," which is inappropriate with alternatives such as "delay" or "impairment."

Some conclusions are overstated relative to the strength of the evidence provided. The authors are encouraged to moderate their claims accordingly.

Figure panels in several cases are too small to interpret easily and, at times, seem to detract from rather than support the main findings. Prioritising the most compelling data, enlarging figure panels, and improving visual clarity would strengthen the presentation.

Specific Comments

5. The statement regarding MDGA1 as a potential translational biomarker for neurodevelopmental and/or neuropsychiatric disorders, including ASD, is too strong given the limited number of reported cases. This should be rephrased or better supported with data. While MDGA1 is not listed in the SFARI gene database, as far as this reviewer is aware, any possible strong association with ASD should be better documented.
6. The claim that the study provides "unprecedented insights into the synaptopathy mechanism causing ASDs" should be tempered to more accurately reflect the data.
7. The use of a single expression vector containing both missense variants does not fully capture the compound heterozygous condition in patients. Although this limitation is mentioned, it has a potential impact on the observed data.
8. The interaction of MDGA1 with Nlgn2 and its putative role in GABAergic synapse development is interesting but underdeveloped. This mechanistic aspect should be more clearly integrated into the broader findings or appropriately framed as speculative.
9. Given the identification of MDGA1 missense mutations in two patients, have the authors considered assessing MDGA1 expression in additional patient-derived samples or accessible tissues such as plasma, where it has been detected (PMID: 31320639)? This could be a valuable addition.
10. In the single-cell atlas analysis, were the datasets (e.g., from ASD or epilepsy patients) comparable (n. individuals)?
11. Many figure legends are too brief and lack necessary methodological context or data interpretation.
12. The claim regarding disrupted synaptic localization of MDGA1 is weakened by the experiments based on overexpression. This limitation deserves greater emphasis, especially given the manuscript's focus on synaptic mechanisms.
13. The heterologous synapse formation assay used to study MDGA1 variant effects on Nlgn2-induced presynaptic assembly may not best model brain-specific interactions.
14. In experiments examining glutamatergic synapses with MDGA2 variants, the expected density of Shank puncta in dendritic spines of EGFP-labeled neurons appears low. This discrepancy should be addressed or explained.
15. In the two-photon imaging of mPFC pyramidal neurons, the absence of observable MDGA1 localization in dendritic spines contrasts with expectations.
16. Several proteins, including APP, show no change in the conditional knockout and knock-in models (Suppl Fig 18). The authors should comment on how this observation aligns-or does not-with their proposed model.
17. The behavioral results do not convincingly support the classification of MDGA1 as an ASD gene. Mdga1-cKO mice performed normally on most behavioral tests, with deficits observed only in the three-chamber social interaction assay. This single result is insufficient to support strong claims. Additionally, prepulse inhibition (PPI), while included, is more commonly associated with schizophrenia research and may not be the most appropriate metric for ASD-related phenotypes.

18. Supplementary Figure 19 is difficult to interpret and should be improved. The reported effects of bazedoxifene-normalization of phosphorylation and behavioral rescue within six hours-are surprising and warrant further justification, ideally with time-course data. The specificity of this compound's effect should also be addressed, particularly as bazedoxifene it seems not discussed elsewhere in the manuscript.

19. The electrophysiological recordings are centered on hippocampal circuits, because of learning or memory deficits previously published. The authors should consider shifting attention to brain regions more relevant to the reported behavioural phenotypes making a strong case that hippocampus is implicated in the social deficit they observed.

Referee #3 (Comments on Novelty/Model System for Author):

This comprehensive study identifies novel compound heterozygous missense mutations in MDGA1 (p.Val116Met/p.Ala688Val and p.Tyr635Cys/p.Glu756Gln) in patients with autism spectrum disorders (ASDs) and provides a deep mechanistic exploration of their impact.

Referee #3 (Remarks for Author):

This comprehensive study identifies novel compound heterozygous missense mutations in MDGA1 (p.Val116Met/p.Ala688Val and p.Tyr635Cys/p.Glu756Gln) in patients with autism spectrum disorders (ASDs) and provides a deep mechanistic exploration of their impact. Through extensive in vitro and in vivo analyses, including structural modeling, electrophysiology in cultured neurons and brain slices, and the generation of novel conditional knockout (cKO) and knock-in (KI) mouse models, the authors demonstrate that these mutations cause distinct synaptic and behavioral deficits via sexually dimorphic loss-of-function (LOF) or gain-of-function (GOF) mechanisms. Specifically, the Tyr635Cys/Glu756Gln variant disrupts MDGA1's triangular structure and impairs its ability to suppress GABAergic synapses in the hippocampus, while the Val116Met/Ala688Val variant affects excitatory synapses and neuronal migration. Critically, male (but not female) Mdga1-cKO and KI mice recapitulate core ASD-like behavioral phenotypes (e.g., altered ultrasonic vocalizations, impaired sensorimotor gating) and show hippocampal GABAergic synaptic hyperexcitability. Phosphoproteomics reveals hypophosphorylation of synapsin II in these models, and remarkably, acute treatment with the selective estrogen receptor modulator bazedoxifene rescues synaptic protein expression, GABAergic synaptic defects, and specific behavioral abnormalities in male KI mice. This work significantly advances our understanding of MDGA1 dysfunction in ASD pathogenesis, highlighting mutation-specific effects, sexual dimorphism, and identifying a potential therapeutic avenue.

While the study provides compelling evidence linking MDGA1 mutations to ASD via sexually dimorphic synaptic dysfunction, several aspects require deeper mechanistic exploration.

1. For the Y636C/E751Q KI model, the disconnect between preserved MDGA1 protein levels and loss-of-function electrophysiological/behavioral phenotypes warrants deeper investigation.
2. Additionally, bazedoxifene's rescue of SynII phosphorylation and synaptic deficits, while promising, lacks direct evidence linking its action to MDGA1-dependent pathways; experiments blocking estrogen receptors or assessing estradiol levels in KI mice would clarify specificity. The rationale of bazedoxifene application needs more supporting evidences.
3. I would like to strongly suggest authors present the general neurodevelopmental phenotypes of the knockin mice in the main figures, such as the general brain structures and difference of cell types in various sub-regions, comparison between WT and KI mice, which would be valuable data for the field.

**Authors' rebuttal letter for Kim, Kim, Pelayo and Alvarez et al.,
“Bazedoxifene rescues sexually dimorphic autistic-like abnormalities in
mice carrying a neurodevelopmental disorder-associated biallelic MDGA1
mutation”, and description of changes made to the revised manuscript**

We are grateful to the reviewers for their careful evaluation and critical analysis of our manuscript. The extensive comments have identified weaknesses in our previous arguments, noted inadequate experimental explanations, and highlighted where more data were needed. To fully respond to the reviewers' comments as carefully as possible, we have performed an extensive series of additional experiments and modified the presentation of the manuscript. The responses to the reviewers' comments are provided below, with the complete reviewers' comments shown in *italic* typeface, our response in bold typeface, and the cited text in bold blue typeface.

Editor comments:

Thank you again for submitting your work to EMBO Molecular Medicine. We have now received the reports from the three reviewers and as you will see below, the reviewers think that the study is potentially interesting. They raise however a series of concerns, which we would ask you to convincingly address in a revision. The reviewers' recommendations are generally clear, so I won't reiterate them here. All the issues raised must be carefully addressed. In particular, the manuscript would benefit from improved clarity and coherence in its presentation, and any overstatements should be avoided. Several points-especially those raised in Referee #3's comments #1 and #2-warrant deeper mechanistic exploration. If available, we also encourage the inclusion of patient-derived samples, as suggested by Referee #2. We would welcome the submission of a revised version within six months for further consideration. As you may already know, our editorial policy allows in principle a single round of major revision, and it is therefore essential to provide responses to the referees' comments that are as complete as possible.

We greatly appreciate the Editor's efforts in handling our manuscript and thank her for underscoring key comments from reviewers that need to be addressed. We made our best effort to completely address the Editor's specific points in the revised manuscript, but we were unable to include results obtained using patient-derived samples due to their unavailability, as detailed in our response (see below). We hope that the Editor now finds this manuscript suitable for publication in *EMBO Mol Med*.

We require:

As instructed, we have provided a docx file including legends for the main figures, EV figures, and EV tables in the revision submission package.

2) Individual production quality figure files as .eps, .tif, .jpg (one file per figure). For guidance, download the 'Figure Guide PDF': (<https://www.embopress.org/page/journal/17574684/authorguide#figureformat>).

We have uploaded the tif file for the main figures in the revision submission package.

3) A .docx formatted letter INCLUDING the reviewers' reports and your detailed point-by-point responses to their comments. As part of the EMBO Press transparent editorial process, the point-by-point response is part of the Review Process File (RPF), which will be published alongside your paper.

This document includes our detailed responses to the reviewers' comments. We agree with the EMBO Press policy that provides a transparent view of the editorial processes together with the published paper.

4) A complete author checklist, which you can download from our author guidelines (<https://www.embopress.org/page/journal/17574684/authorguide#submissionofrevisions>). Please insert information in the checklist that is also reflected in the manuscript. The completed author checklist will also be part of the RPF.

We have done as instructed.

We have done as instructed.

6) It is mandatory to include a 'Data Availability' section after the Materials and Methods. Before submitting your revision, primary datasets produced in this study need to be deposited in an appropriate public database, and the accession numbers and database listed under 'Data Availability'. Please remember to provide a reviewer password if the datasets are not yet public

(see <https://www.embopress.org/page/journal/17574684/authorguide#dataavailability>). In case you have no data that requires deposition in a public database, please state so in this section. Note that the Data Availability Section is restricted to new primary data that are part of this study.

We have included a 'Data Availability' section after the 'Materials and Methods' section as instructed. We ensured that our proteomic datasets were deposited before submission of the revised paper.

7) For data quantification: please specify the name of the statistical test used to generate error bars and P values, the number (n) of independent experiments (specify technical or biological replicates) underlying each data point and the test used to calculate p-values in each figure legend. The figure legends should contain a basic description of n, P and the test applied. Graphs must include a description of the bars and the error bars (s.d., s.e.m.). See also 'Figure Legend'

guidelines: <https://www.embopress.org/page/journal/17574684/authorguide#figureformat>

We have included all the required information in each figure legend of the revised manuscript as instructed.

We have included all source data for the main figures in the revision submission package as instructed.

9) Our journal encourages inclusion of *data citations in the reference list* to directly cite datasets that were re-used and obtained from public databases. Data citations in the article text are distinct from normal bibliographical citations and should directly link to the database records from which the data can be accessed. In the main text, data citations are formatted as follows: "Data ref: Smith et al, 2001" or "Data ref: NCBI Sequence Read Archive PRJNA342805, 2017". In the Reference list, data citations must be labeled with "[DATASET]".

A data reference must provide the database name, accession number/identifiers and a resolvable link to the landing page from which the data can be accessed at the end of the reference. Further instructions are available at <https://www.embopress.org/page/journal/17574684/authorguide#referencesformat>.

We have done so in the revised manuscript.

10) We replaced Supplementary Information with Expanded View (EV) Figures and Tables that are collapsible/expandable online. A maximum of 5 EV Figures can be typeset. EV Figures should be cited as 'Figure EV1, Figure EV2' etc... in the text and their respective legends should be included in the main text after

the legends of regular figures.

We have prepared six Expanded View Figures and three Expanded View Tables in the revised manuscript and cited them in the text as instructed.

*- For the figures that you do NOT wish to display as Expanded View figures, they should be bundled together with their legends in a single PDF file called *Appendix*, which should start with a short Table of Content. Appendix figures should be referred to in the main text as: "Appendix Figure S1, Appendix Figure S2" etc.*

We have prepared a single PDF file that contains 34 Appendix Figures associated with the revised manuscript, and cited them as instructed.

- Additional Tables/Datasets should be labeled and referred to as Table EV1, Dataset EV1, etc. Legends have to be provided in a separate tab in case of .xls files. Alternatively, the legend can be supplied as a separate text file (README) and zipped together with the Table/Dataset file. See detailed instructions here:

<https://www.embopress.org/page/journal/17574684/authorguide#expandedview>.

We have prepared six datasets for the revised manuscript and cited them as instructed.

- the medical issue you are addressing,

- the results obtained and

- their clinical impact.

We have provided a separate Word file designated “The paper explained” as instructed.

12) Author contributions: You will be asked to provide CRediT (Contributor Role Taxonomy) terms in the submission system. These replace a narrative author contribution section in the manuscript.

We have provided an “Author contribution” section using the CRediT system as instructed.

13) A Conflict of Interest statement should be provided in the main text.

We have done so in the revised manuscript.

14) Every published paper now includes a 'Synopsis' to further enhance discoverability. Synopses are displayed on the journal webpage and are freely accessible to all readers. They include a short stand first (maximum of 300 characters, including space) as well as 2-5 one-sentences bullet points that summarizes the paper. Please write the bullet points to summarize the key NEW findings. They should be designed to be complementary to the abstract - i.e. not repeat the same text. We encourage inclusion of key acronyms and quantitative information (maximum of 30 words / bullet point). Please use the passive voice. Please attach these in a separate file or send them by email, we will incorporate them accordingly. Please also suggest a visual abstract to illustrate your article as a PNG file 550 px wide x 300-600 px high.

As instructed, we have prepared a ‘Synopsis’ as a Word file, a graphical summary as a PNG file, and three bullet points for the revised manuscript.

15) All Materials and Methods need to be described in the main text using our 'Structured Methods' format. According to this format, the Methods section includes a Reagents and Tools Table (listing key reagents, experimental models, software and relevant equipment and including their sources and relevant identifiers) followed by a Methods and Protocols section describing the methods, ideally using a step-by-step protocol format. The aim is to facilitate adoption of the methodologies across labs.

Please download and fill our Reagents and Tools Table template (.docx), which you can find in our author

guidelines: <https://www.embopress.org/page/journal/17574684/authorguide#structuredmethods>

As instructed, we have prepared a 'Reagents and Tools Table', included it in the revised manuscript, and also uploaded it as a separate file.

Reviewers' comments:

Referee #1: Researchers identified two unrelated children with autism spectrum disorder who carry different pairs of rare, compound heterozygous missense variants in MDGA1 (Val116Met + Ala688Val and Tyr635Cys + Glu756Gln) and showed, through structural modelling and biochemical assays, that the Tyr635Cys/Glu756Gln pair destabilises the characteristic triangular ectodomain of MDGA1 without altering surface expression or ligand binding. In cultured neurons the destabilising variant abolishes MDGA1's normal ability to restrain GABAergic synapse maturation, whereas the Val116Met/Ala688Val pair unexpectedly reduces excitatory synapse density and, when over expressed in utero, derails radial migration of cortical pyramidal neurons and depresses pup ultrasonic vocalizations. Conditional deletion of Mdga1 in mouse forebrain excitatory neurons and knock in substitution of Tyr636Cys/Glu751Gln each produce male specific autistic like behaviours, including blunted maternal separation and courtship USVs, impaired sociability or prepulse inhibition, and exaggerated startle responses, while females remain largely spared. Electrophysiology reveals that both mutants amplify dendritic GABAergic inhibition in CA1 and that hippocampal phosphoproteomics convergently show hypophosphorylation of presynaptic Synapsin II, pointing to a shared loss of function pathway. A single systemic dose of the brain penetrant selective oestrogen receptor modulator bazedoxifene restores Synapsin II phosphorylation, normalises GABAergic input strength and rescues vocal, startle and sensorimotor gating phenotypes in male knock in mice. Together, the work establishes MDGA1 dysfunction as a causal, sexually dimorphic synaptopathy that links inhibitory synapse dysregulation to core ASD behaviours and shows that it is pharmacologically reversible. This is the first study to connect biallelic MDGA1 variants with autism and to dissect their structural, synaptic and behavioural consequences across patient data, cell culture and multiple mouse lines, thereby expanding the roster of inhibitory "synaptopathy" genes underlying ASD and illuminating why many phenotypes preferentially afflict males. Demonstrating full behavioural rescue with an FDA approved estrogen pathway drug makes the findings immediately translatable, suggests a mechanistic basis for the male bias in ASD, and provides a timely proof of concept for precision therapies that target sex dependent synaptic vulnerabilities. The manuscript is impressively comprehensive, with convergent data that are novel, rigorous and timely. I have only the following few minor comments.

We appreciate the reviewer's appraisal of our study and their constructive comments, which have been fully addressed in the revised manuscript.

1. The authors need to include standardized diagnostic scores (e.g., ADOS 2, Vineland) for the two autistic individuals to strengthen genotype phenotype interpretation.

Both patients were subjected to the ADOS; their scores were 13 (for P1) and 19 (for P2). These results and relevant description are now included in the revised manuscript, as follows: *The first patient (P1) is a 5-year-old girl diagnosed with moderate psychomotor retardation with autistic features, according to clinical and neuropsychological evaluation (IQ 63; ADOS score 13). On examination, she showed some mild dysmorphic features: frontal bossing, prominent eyes, and thin tented upper lip. The second patient (P2) is a 5-year-old boy with low intellectual functioning and autistic features, also based on clinical and cognitive assessments (IQ 71; ADOS score 19).*

2. I wonder if the authors have considered to measure MDGA1 protein levels in patient derived cells or plasma to help link human variant burden to molecular deficit. If not, the rationale not doing so should be clarified.

We thank the reviewer for this insightful comment, which resonates with important feedback from other reviewers. While assessing MDGA1 protein levels in patient-derived cells or plasma would indeed strengthen this paper, we unfortunately face significant practical limitations in obtaining these samples. Specifically, challenges in securing informed consent for biospecimen collection make this approach currently unfeasible for this study. We acknowledge the critical significance of such measurements and have included a statement regarding these practical constraints in the Results section of the revised manuscript, as follows: *While direct measurement of MDGA1 protein levels in patient-derived biospecimens would be valuable, practical limitations, including challenges in patient consent, precluded their collection for this study.*

3. The authors tried bazedoxifene experiments in an acute-treatment format. I wonder if they thought about trying chronic-treatment experiments and, if not, they need to explain why they chose the acute treatments.

We appreciate the reviewer's important question regarding our choice of an acute treatment paradigm with bazedoxifene (BZD). Our decision was based on a solid foundation of prior research demonstrating BZD's rapid brain penetrance and swift neuromodulatory potential. Specifically, Hill et al. (PMID: 32858306) showed that a single intraperitoneal injection of BZD quickly crossed the blood-brain barrier. Within just 6 hours, this acute administration was sufficient to rescue spatial memory deficits in ovariectomized mice. This behavioral rescue effect was accompanied by the activation of estrogen response element-dependent gene transcription in the brain, providing direct evidence of acute CNS activity following systemic BZD administration. Consistent with these findings, our own treatment paradigm also involved a single intraperitoneal injection of BZD, and we similarly observed the restoration of increased inhibitory synaptic protein levels, enhanced synapsin II phosphorylation, and behavioral improvements. Thus, our observations aligned with the previous report, indicating that BZD can engage brain estrogen receptor pathways rapidly enough to drive molecular and behavioral responses within an acute timeframe. We have added more relevant information to the **Results and Discussion** sections of revised manuscript as follows: **(Results)** *Given that both male and female Mdga1-mutant mice exhibited comparable increases in inhibitory synaptic transmission during the juvenile period, whereas only males (and not females) retained these abnormalities into adulthood (see Figs. 5 and EV6), we considered the possibility that puberty-associated, sex-dependent factor(s) contribute to this divergence (Ferri et al, 2018). We therefore examined whether estrogen-sensitive pathways—which are known to influence inhibitory synapse maturation and circuit refinement—modulate the persistent adult male phenotype. Proteomics analyses revealed no significant changes in the levels of estrogen receptor-associated proteins (Ddx5 and Phb2), but marginal (yet significant) changes in the levels of the NDD risk factors, Cnot1 and Parp1 (Dong et al, 2023; Humphries et al, 2023; Vissers et al, 2020) (Appendix Fig. S32), in male Mdga1-cKO and Mdga1^{Y636C/E751Q} KI mice, hinting that there is a link between the MDGA1 and estrogen receptor signaling pathways. We also performed enzyme-linked immunosorbent assay-based measurement of estradiol levels in male juvenile and adult Mdga1^{Y636C/E751Q} KI mice and found no significant differences in estradiol levels between WT and KI mice (Appendix Fig. S33). These findings indicate that baseline estrogen signaling is not substantially altered in Mdga1-mutants, but raise the possibility that enhancing estrogen-responsive pathways might still modulate the persistent adult male phenotypes. **(Discussion)** Although studies on acute neuromodulatory effects of BZD treatment are still limited, other SERMs, such as tamoxifen and raloxifene, have demonstrated similar rapid effects on neuronal plasticity behavior (Finney et al, 2021; Velazquez-Zamora et al, 2012), supporting the broader concept that estrogen receptor modulation can exert biologically significant acute effects on the brain within hours of administration. Consistent with our model, acute application of the estrogen receptor degrader, Fulvestrant, partially suppressed the BZD-induced normalization of inhibitory synaptic function in Mdga1-KI slices. Because Fulvestrant requires ≥ 6 –12 h to induce ubiquitin–proteasome–mediated degradation of nuclear ER α (Hammes & Levin, 2007; Long & Nephew, 2006), the brief 20-min exposure used here is expected to selectively interfere with rapid membrane-initiated estrogen signaling (Callige & Richard-Foy, 2006; Vasudevan & Pfaff, 2007). This selective and mechanism-specific partial blockade provides direct functional evidence that the BZD-induced rescue requires intact estrogen receptor signaling, supporting the involvement of an estrogen-sensitive pathway in the MDGA1-dependent synaptic phenotype. Although the details of the relevant molecular interactions remains to be fully elucidated, we note that estrogen signaling is known to rapidly reduce inhibitory synaptic strength in the hippocampus (Huang & Woolley, 2012; Mukherjee et al., 2017). These fast-acting suppressive effects on GABAergic function may act synergistically with the previously established role for MDGA1 as an inhibitory synapse suppressor. Thus, we speculate that BZD may transiently compensate for the loss of MDGA1 function in male Mdga1-KI mice through partially overlapping pathways that remain to be identified in future studies.*

4. Supplementary Fig. 12 shows decreased number of total USV calls in the VM/AV group. I wonder if the duration of each USV call, or the total duration of USV calls, is also changed.

To address the reviewer's inquiry regarding the duration of pup USV calls, we reanalyzed the data. Our analysis revealed that there were no significant differences in the total duration of USV calls between groups. These new results are now included in Expanded View Figure 2D (originally Supplementary Figure 12).

(Expanded View Figure 2)

Referee #2:

1. The layout and presentation of the figures need improvement for better clarity and interpretation.

We sincerely thank the reviewer for their valuable feedback regarding the clarity and presentation of our figures. We agree that optimizing figure layout is crucial for effective data interpretation and appreciate this opportunity to enhance our manuscript. We separated original Supplementary Figures 19 and 20 into eight distinct Appendix Figures in the revised manuscript to enhance layout, optimize readability, and clarify the visual elements. We also thoroughly revised the figure legends to be more comprehensive and self-explanatory, allowing readers to fully understand the figure content without referring extensively to the main text. We believe these revisions significantly improve the overall clarity and impact of our study and hope that the reviewer concurs with us.

2. The chosen mouse model does not convincingly demonstrate strong face validity for the condition studied.

We appreciate the reviewer's critical evaluation of our mouse model's face validity. We agree that no single animal model can fully recapitulate the complex and multifaceted nature of human neurodevelopmental or neuropsychiatric disorders, and our model is no exception. However, our primary goal in developing this knock-in mouse model was not to achieve high face validity, but rather to focus on construct validity. By introducing the specific *MDGA1* mutation identified in human patients, our model allows us to directly investigate the causal molecular and cellular mechanisms by which this genetic alteration contributes to relevant pathological phenotypes. While the global clinical features may be partially recapitulated, our model convincingly demonstrates specific, quantitative deficits in social interaction and communication deficits that are highly relevant to core diagnostic criteria and pathophysiological hypotheses of neurodevelopmental disorders. These specific phenotypic readouts provide a valuable platform for dissecting the underlying neural circuitry dysfunction. Thus, even if face validity is not 'strong' in the holistic sense of mirroring all human complexity, our model offers critical mechanistic insights that cannot be obtained from human studies alone. It enabled us to establish a direct link between a specific genetic alteration and specific synaptic and behavioral deficits, which is invaluable for identifying potential therapeutic targets and testing interventions at a preclinical level. These mechanistic findings are vital for translating basic science into future clinical applications.

3. Based on the data presented, the medical relevance remains unclear-aside from identifying two individuals with mutations in the MDGA1 gene. The associated complex clinical features may only be partially recapitulated in the mouse model.

We appreciate the reviewer's critical assessment of the medical relevance of our findings and the validity of our mouse model. We fully agree that the direct medical implications from identifying only two individuals with *MDGA1* mutations are, at this stage, limited. Our study's primary aim is to provide fundamental mechanistic insights into how *MDGA1* dysfunction contributes to synaptic alterations and neurodevelopmental phenotypes. While direct translation to the clinic will require extensive additional research, understanding these underlying molecular and cellular mechanisms is a crucial first step in the long process of developing diagnostics and therapeutics for complex neurological disorders. As the reviewer aptly points out, linking autistic traits in people to behaviors in mice is possible, yet inevitably incomplete. We acknowledge that complex human clinical features are only ever partially recapitulated in any single mouse model. Our *Mdga1* knock-in mouse model was specifically designed to investigate the causal link between specific patient-identified *MDGA1* mutations and core neurodevelopmental phenotypes, such as altered social interaction and specific synaptic dysfunctions in relevant brain regions. Our model serves as a valuable tool to dissect the molecular and circuit-level consequences of *MDGA1*

perturbation, providing a controlled system that is not feasible in human studies. We concur that the face validity of rodent models for complex neurodevelopmental disorders is often incomplete. Indeed, reviews of two decades of work emphasize that most rodent lines carrying high-confidence risk genes for ASDs (e.g., SHANK3 and CNTNAP2) capture one or two of the symptom clusters, seldom mirroring the full triad of autistic behaviors. While these models often show convergences like reduced sociability or repetitive motor routines, providing interesting construct validity, full face validity remains elusive due to factors such as social domain complexities, strain background, and housing conditions, which can alter the direction or magnitude of a phenotype. Consequently, our findings can be viewed as exhibiting ‘partial coherence’ with human autistic phenotypes, similar to what is observed in other genopathies related to autism. We emphasize that the strength of our model lies in its construct validity, in that it can mechanistically link specific human genetic variants to underlying synaptic and behavioral deficits relevant to neurodevelopmental disorders. Furthermore, while the number of reported human cases with MDGA1 mutations is currently limited, it is not uncommon for novel candidate genes implicated in complex neurological disorders to initially involve relatively few reported cases. Our study, by establishing a mechanistic link in a robust animal model, provides compelling evidence for MDGA1's pathogenic role, thereby potentially guiding future genetic screening efforts in patient cohorts and contributing to a broader understanding of the genetic landscape of neurodevelopmental disorders. We have carefully revised the manuscript to incorporate these nuanced perspectives on model validity and the scope of our claims, ensuring that we present a more balanced and precise representation of our findings and their implications. We hope that the reviewer concurs with us.

4. I appreciate the considerable amount of work and technical expertise that went into this manuscript; the authors have conducted a wide range of experiments using diverse methodologies. However, in its current form, the manuscript feels fragmented, and the narrative lacks coherence. As a result, the central message becomes somewhat diluted rather than reinforced. To improve readability and overall impact, a clearer and more integrated storyline is strongly recommended. Additionally, I recommend replacing the term "retardation," which is inappropriate with alternatives such as "delay" or "impairment." Some conclusions are overstated relative to the strength of the evidence provided. The authors are encouraged to moderate their claims accordingly. Figure panels in several cases are too small to interpret easily and, at times, seem to detract from rather than support the main findings. Prioritising the most compelling data, enlarging figure panels, and improving visual clarity would strengthen the presentation.

We sincerely thank the reviewer for their thorough and insightful evaluation of our manuscript. We deeply appreciate their recognition of the considerable amount of work and technical expertise that went into this study, involving a wide range of experiments and diverse methodologies. We have carefully considered all comments and believe they offer invaluable guidance for significantly improving our manuscript. We fully agree with the critical point that, in its current form, the manuscript feels fragmented and the narrative lacks coherence, thereby diluting our central message. To address this comprehensively, we undertook a thorough restructuring of the manuscript. Furthermore, we sincerely apologize for the inappropriate use of the term 'retardation' and are grateful to the reviewer for pointing this out. We have immediately replaced this term with more appropriate and precise alternatives such as 'delay,' 'impairment,' or 'deficits' throughout the entire manuscript to ensure respectful and accurate scientific language. We also accept the valuable comment that some conclusions may be overstated relative to the strength of the evidence. We have meticulously reviewed the manuscript to moderate our claims accordingly, revising our language to use more nuanced verbs rather than definitive terms. This ensures that our conclusions accurately reflect the extent and strength of the data presented. Finally, we agree that the current layout and presentation of several figures detract from rather than support the main findings, with panels often being too small to be easily interpreted. To

strengthen the presentation and improve visual clarity, we undertook a comprehensive revision of all figures. We are confident that these substantial revisions significantly enhance the readability, impact, and overall scientific rigor of our manuscript.

5. The statement regarding MDGA1 as a potential translational biomarker for neurodevelopmental and/or neuropsychiatric disorders, including ASD, is too strong given the limited number of reported cases. This should be rephrased or better supported with data. While MDGA1 is not listed in the SFARI gene database, as far as this reviewer is aware, any possible strong association with ASD should be better documented.

We agree that, given the limited number of reported human cases and the preclinical nature of our findings, the statement was too strong in the original manuscript. In response to the reviewer's suggestion, we investigated the potential association between MDGA1 variants and neurodevelopmental disorders/neuropsychiatric diseases in major gene/disease databases beyond SFARI, including OMIM, ClinGen, gnomAD, and DECIPHER (see the summary table below; also presented in Table EV3 of the revised manuscript).

(Table EV3)

Database	Disease/Phenotype	Variant Type	Association/Evidence	Note / Link
SFARI Gene	Autism Spectrum Disorder (ASD)	-	Not listed	https://gene.sfari.org/database/human-gene/MDGA1
OMIM	Neurodevelopmental/psychiatric	-	No direct disease association	https://www.omim.org/entry/613126
ClinGen	ASD, neurodevelopmental disorders	-	No dosage pathogenicity/association	https://search.clinicalgenome.org/kb/genes/hgnc/MDGA1
gnomAD v4	-	pLI = 0 (No LoF constraint)	Multiple PTVs observed	gnomAD v4
DECIPHER	Neurodevelopmental disorders	CNV (triplication, VUS)	Patient #360922: MDGA1 duplication (VUS)	DECIPHER

As indicated in this Table, and as the reviewer noted, MDGA1 is currently not listed in the SFARI Gene database. There are no direct associations reported between MDGA1 and neurodevelopmental/neuropsychiatric disorders in OMIM. The clinical relevance for MDGA1, such as dosage pathogenicity, has not yet been assessed in ClinGen. Furthermore, the pLI score of MDGA1 is 0 (GRCh38 in gnomAD), with several protein-truncating variants observed, suggesting that there is no constraint. Lastly, our screening of DECIPHER revealed that one patient (#360922) in this large UK developmental disorder patient database was reported to have an MDGA1 triplication, which was classified as a Variant of Uncertain Significance (VUS). It was not categorized as pathogenic. Given this information, we concur with the reviewer's assessment. We have thus softened the statement in the Discussion section of the revised manuscript regarding MDGA1 as follows: *This potential has been suggested, but large-scale clinical and functional studies are needed for further validation (Table EV3). The number of reported human cases with MDGA1 mutations currently remains limited, and MDGA1 is not yet widely recognized as a major ASD risk gene in comprehensive databases such as SFARI. Our findings establish MDGA1 as a novel candidate gene and provide critical mechanistic insights, but extensive future validation in larger human cohorts, including functional studies of patient-derived cells and comprehensive genetic screenings, will be essential for firmly establishing its utility as a reliable diagnostic, prognostic, and/or therapeutic biomarker. These ongoing efforts will be crucial for bridging the gap between basic mechanistic discoveries and clinical applications.*

6. The claim that the study provides "unprecedented insights into the synaptopathy mechanism causing ASDs" should be tempered to more accurately reflect the data.

To comply with this comment, we changed the indicated sentence as follows: *Our study provides novel insights into a potential synaptopathy mechanism contributing to ASD-related phenotypes...*

7. *The use of a single expression vector containing both missense variants does not fully capture the compound heterozygous condition in patients. Although this limitation is mentioned, it has a potential impact on the observed data.*

We appreciate the reviewer's thoughtful comment regarding our use of a single expression vector for both missense variants, and we acknowledge their valid point about the potential impact of this choice on the ability to fully recapitulate the compound heterozygous condition observed in patients. While we agree that this approach does not precisely replicate the *in vivo* compound heterozygous state, our experimental design using a single expression vector was intentionally chosen to investigate the combined functional consequence of these two specific mutations. Our primary aim was to understand whether the co-occurrence of these variants leads to a more severe or distinct functional impairment compared to individual mutations, which is critical for deciphering their pathogenic mechanism. This approach allowed us to directly test their synergistic or additive effects on MDGA1 protein stability and synaptic localization in a controlled manner that would be challenging with separate alleles *in vitro*. Despite this inherent limitation in mimicking the exact genetic context, our model provides invaluable mechanistic insights into how these mutations collectively impair MDGA1's ability to regulate GABAergic synapses. The observed significant alteration in its subcellular distribution in our model strongly suggests that these combined variants have a pathogenic effect, offering a critical foundation for understanding the human condition. As acknowledged in our original manuscript and as rightly emphasized by the reviewer, we recognize this as a crucial limitation. To address the reviewer's concern for greater emphasis, we now further elaborate on this point in the Discussion section, explicitly detailing how the compound heterozygous condition might differ from our single-vector overexpression model and outlining the implications for interpreting the data. We also highlight this aspect when discussing the translational relevance of our findings, reinforcing that further studies incorporating separate alleles or more complex genetic models would be valuable for fully dissecting the individual contributions and interactions of these variants *in vivo*.

8. *The interaction of MDGA1 with Nlgn2 and its putative role in GABAergic synapse development is interesting but underdeveloped. This mechanistic aspect should be more clearly integrated into the broader findings or appropriately framed as speculative.*

In our previous study (PMID: 35074912), which established the cellular model for the current work, we demonstrated that MDGA1 exerts its suppressive function at GABAergic synapses in hippocampal CA1 pyramidal neurons primarily by engaging APP, not Nlgn2. Given this established mechanistic framework, and our finding that the MDGA1 missense mutations addressed in the current study were not found to impact the direct interaction between MDGA1 and Nlgn2 (Fig. 1K and 1L) or the MDGA1-mediated suppression of Nlgn2 synaptogenic activity (Appendix Fig. S6), our investigation focused on dissecting the pathophysiological mechanisms relevant to these specific mutations. Thus, we did not further explore the MDGA1-Nlgn2 interaction as the primary cause of the observed phenotypes in our MDGA1-related neurodevelopmental disorder models. The identified MDGA1 mutations also do not appear to affect the direct interaction between MDGA1 and APP. Instead, these mutations seem to influence neuronal migration activity or, notably, the triangular extracellular structure of MDGA1. Thus, we believe that the current study provides novel insights into the critical role of maintaining the structural integrity of MDGA1 for its proper function at synapses. This shift of focus, driven by the specific nature of the patient mutations, highlights a new dimension for the MDGA1 mechanism of action.

9. Given the identification of *MDGA1* missense mutations in two patients, have the authors considered assessing *MDGA1* expression in additional patient-derived samples or accessible tissues such as plasma, where it has been detected (PMID: 31320639)? This could be a valuable addition.

This comment echoes the comment #2 of reviewer #1. While assessing *MDGA1* protein levels in patient-derived cells or plasma would indeed strengthen this paper, we unfortunately face significant practical limitations in obtaining these samples. Specifically, challenges in securing informed consent for biospecimen collection make this approach currently unfeasible for this study. We acknowledge the critical significance of such measurements and have included a statement regarding these practical constraints in the relevant section of the Materials and Methods of the revised manuscript, as follows: *While direct measurement of MDGA1 protein levels in patient-derived biospecimens would be valuable, practical limitations, including challenges in obtaining patient consent, precluded their collection for the current study.*

10. In the single-cell atlas analysis, were the datasets (e.g., from ASD or epilepsy patients) comparable (n. individuals)?

The single-cell atlas we used for gene set enrichment test did not include samples from ASD, epilepsy, or other patient populations. Instead, it was built exclusively from transcriptomic data of neurotypical individuals across a wide developmental window (gestational week 7 to 90 years). We recognize that our previous description might have caused confusion regarding the composition of the atlas. To improve clarity, we have revised the relevant text to explicitly state that no clinical cohorts were included, as follows: *To investigate MDGA1 expression throughout brain development, we utilized a single-cell atlas of the neurotypical human post-mortem brain (Data ref: Kim et al, 2024b).*

Although this neurotypical reference atlas did not include clinical cohorts, it offers critical insight into normative gene expression patterns in the human brain across the lifespan, allowing us to infer the relevance of *MDGA1* expression in early neurodevelopment. Furthermore, we addressed disorder relevance by performing gene set enrichment analyses using cluster-specific genes from the atlas and independently curated gene sets associated with ASD, epilepsy, and other neurodevelopmental conditions. This indirect yet robust approach enabled us to assess potential links between *MDGA1*-expressing neuronal populations and disease-associated gene networks, despite the absence of patient samples in that specific dataset. To directly address the reviewer's concern about patient data comparability, we have added a new analysis using a single-nucleus dataset (PMID: 31097668) of ASD patients and control subjects, allowing us to directly compare cell type-specific *MDGA1* expression between the two groups. This dataset includes information from 52,003 cells and 52,556 cells from 15 ASD patients and 16 controls, respectively. These numbers provide comparable sample sizes for a robust analysis. We discovered that *MDGA1* expression is significantly decreased (FDR < 0.01) in ASD individuals compared to the control group, specifically in layer II/III excitatory neurons (L2/3) and protoplasmic astrocytes (AST-PP). These findings corroborate our primary results, highlighting the functional importance of *MDGA1* in L2/3 excitatory neurons. They also suggest that cell-type-specific dysregulation of *MDGA1* expression may contribute to ASD pathophysiology. We present these new results in Appendix Fig. S3 and Dataset EV2, and we describe the methodology in the Methods section of the revised manuscript. In addition, we now describe these results in the revised manuscript, as follows: *We compared the cell-type specific MDGA1 expression between ASD patients and control individuals using a single-nucleus atlas composed of 15 ASD patients and 16 controls (Velmeshev et al, 2019) (Appendix Fig. S3A–C). We found that layer II/III excitatory neurons (L2/3) and protoplasmic astrocytes (AST-PP) in ASD individuals showed significantly decreased MDGA1 expression (FDR < 0.01) (Appendix Fig. S3D, Dataset EV2) compared to those from control subjects. These findings suggest that MDGA1 should be examined cell-type specifically and the dysregulated MDGA1 expression in a specific cell population may contribute to the pathophysiology of ASDs.*

(Appendix Fig. S3)

11. Many figure legends are too brief and lack necessary methodological context or data interpretation. In response to the reviewer's comment, we have gone through all the legends in the main figures, Expanded View figures, and Appendix figures to ensure that they include all necessary information, such as the name of the statistical test, p -values, and number of independent experiments or biological replicates, as requested by both the reviewer and the Editor.

12. The claim regarding disrupted synaptic localization of MDGA1 is weakened by the experiments based on overexpression. This limitation deserves greater emphasis, especially given the manuscript's focus on synaptic mechanisms.

We fully agree that claims based on overexpression should be interpreted with caution, particularly when discussing physiological localization and function, and that this limitation deserves greater emphasis in the manuscript, especially given our focus on synaptic mechanisms. We wish to emphasize that our overexpression experiments were not primarily intended to precisely replicate endogenous physiological MDGA1 localization. Instead, their purpose was to investigate how specific MDGA1 mutants alter the established ability of MDGA1 to suppress GABAergic synapses, as demonstrated by our group and others (e.g., PMID: 38900791, 35074912, 23248271, 29281813 and 23358245). These experiments allowed us to test whether specific MDGA1 mutations altered the abilities of MDGA1 to negatively modulate GABAergic synapses, which might be difficult to observe with endogenous expression levels. While we acknowledge the caveats that come with overexpression experiments, we believe that the data derived from these experiments still provide important evidence for the ability of MDGA1 to suppress

GABAergic synapses. These findings, when considered alongside our data from loss-of-function experiments using *Mdga1*-cKO mice, contributed to providing a more comprehensive understanding of MDGA1's role in synaptic mechanisms. They allowed us to uncover certain intrinsic properties or interaction capabilities of MDGA1 that are relevant to its function, irrespective of precise physiological expression levels. To address the reviewer's concern, we added an explicit statement in the Discussion section, emphasizing that: *Our investigation into the role of MDGA1 in influencing synaptic protein localization and function, particularly the observed suppression of GABAergic synapses, involved overexpression experiments. While the obtained results provide valuable mechanistic insights into the intrinsic ability of MDGA1 to modulate synaptic architecture and its interaction with key synaptic partners, it is crucial to interpret these findings with caution regarding their direct translation to physiological endogenous condition. Overexpression systems can sometimes influence native protein trafficking, induce non-physiological interactions, or mask subtle effects that occur at endogenous expression levels. Thus, results from overexpression approaches should primarily serve as a complementary tool for exploring underlying molecular mechanisms, as partially supported by our parallel loss-of-function experiments in *Mdga1*^{Y636C/E751Q} KI mice. Furthermore, our approach of incorporating two MDGA1 missense variants into a single expression vector to study their combined effect does not precisely replicate the compound heterozygous genetic context observed in patients, where each mutation occurs on a separate allele. While this design allowed us to efficiently investigate the synergistic or additive functional consequences of these co-occurring mutations on the properties of MDGA1, it hinders us from fully dissecting the individual contributions or potential epistatic interactions of each variant in vivo. Future studies employing separate allelic models or more sophisticated genetic engineering approaches would be invaluable for elucidating the precise role of each individual mutation within a true compound heterozygous setting.* **This revision is intended to ensure that we clearly communicate the limitations, while still highlighting the valuable information gained from these experiments.**

13. *The heterologous synapse formation assay used to study MDGA1 variant effects on Nlgn2-induced presynaptic assembly may not best model brain-specific interactions.*

We agree that the heterologous synapse formation assay does not necessarily replicate the intricate cellular environment and complex signaling found in a native brain. However, the specific purpose of employing this assay was to provide a highly controlled and simplified environment in which we can directly assess the intrinsic molecular capabilities of MDGA1 and its patient-derived missense mutations. This allowed us to precisely determine whether these mutations alter the ability of MDGA1 wild-type to suppress Nlgn2-induced presynaptic assembly, isolating this particular molecular interaction without confounding factors from complex neuronal networks. While we emphasize that the results from these heterologous synapse formation assays do not directly reflect the pathophysiological processes underlying MDGA1-related autism spectrum disorders and/or neurodevelopmental disorders, they provide critical mechanistic insights at the molecular level. Importantly, none of the *MDGA1* mutations altered Nlgn2-induced presynaptic assembly in this assay, which is entirely consistent with our observation that these mutations do not affect the direct interaction of MDGA1 with Nlgn2.

14. *In experiments examining glutamatergic synapses with MDGA2 variants, the expected density of Shank puncta in dendritic spines of EGFP-labeled neurons appears low. This discrepancy should be addressed or explained.*

We have changed the representative images to better reflect the quantification results presented in the revised manuscript.

(Appendix Fig. S8)

15. In the two-photon imaging of mPFC pyramidal neurons, the absence of observable MDGA1 localization in dendritic spines contrasts with expectations.

We were unable to examine the localization of endogenous MDGA1 proteins at synaptic sites due to the unavailability of suitable antibodies. Several previous studies support the localization of MDGA1 at hippocampal excitatory synapses both *in vitro* and *in vivo* (PMID: 27565350, 35532105 and 37720016). However, our current results obtained using *Mdga1*-cKO and *Mdga1*-KI mice appear to contradict these prior findings in the hippocampus. It is important to note that the localization of MDGA1 proteins, specifically in mPFC pyramidal neurons, has not yet been investigated. While it would be extremely valuable to perform such analyses using transgenic *Mdga1*-KI mice (as described in PMID: 37720016), we regret that we cannot undertake this experiment within the specified revision period. Furthermore, we did not have any prior expectation regarding the localization of MDGA1 proteins in mPFC pyramidal neurons, as this specific context remains unexplored in the literature. We hope that the reviewer is satisfied with our detailed explanation.

16. Several proteins, including APP, show no change in the conditional knockout and knock-in models (Suppl Fig 18). The authors should comment on how this observation aligns-or does not-with their proposed model.

We appreciate the reviewer's query regarding the lack of change in APP expression levels in our *Mdga1*-cKO and *Mdga1*-KI mice. This observation was, in fact, not inconsistent with our proposed model and prior findings. Our previous study (PMID: 35074912) demonstrated that MDGA1 functionally constrains the presynaptic function of APP, specifically influencing synaptic strength and neurotransmitter release probability at GABAergic synapses in the hippocampal CA1 pyramidal neurons from adult mice. The mechanism involves the MDGA1-mediated regulation of APP's function at the synapse, not its overall expression level. Crucially, the *MDGA1* missense mutations reported in the current study do not affect the surface expression

level of MDGA1 or its binding activity with APP and Nlgn2. Instead, one of these mutations primarily influences the triangular extracellular structure of MDGA1, which we propose alters its ability to modulate its synaptic function rather than its direct interaction with binding partners. Thus, the unchanged APP expression level seen in our models is fully aligned with our proposed mechanistic model, which posits that APP is functionally regulated by MDGA1, rather than undergoing direct modulation of its expression. We hope that this explanation clarifies how this observation fits within our proposed framework.

17. The behavioral results do not convincingly support the classification of MDGA1 as an ASD gene. Mdga1-cKO mice performed normally on most behavioral tests, with deficits observed only in the three-chamber social interaction assay. This single result is insufficient to support strong claims. Additionally, prepulse inhibition (PPI), while included, is more commonly associated with schizophrenia research and may not be the most appropriate metric for ASD-related phenotypes.

We appreciate the reviewer's critical comments and the opportunity to clarify our behavioral findings. While it is true that *Mdga1*-cKO mice did not exhibit broad behavioral deficits, it is inaccurate to state that they showed impairment only in the three-chamber social interaction assay. In fact, *Mdga1*-cKO mice also exhibited significant reductions in pup and adult USVs, which are widely recognized as a key behavioral correlate of early-life social communication in rodent ASD models. The presence of deficits in both sociability and USVs indicates that cKO mice display abnormalities in two of the three core diagnostic domains of ASD, namely social interaction and social communication. Importantly, USV deficits have been consistently reported in well-established ASD models, such as *BTBR*, *Shank3*, *Fmr1*, and *Cntnap2* mutant mice (PMID: 18728777, 33269765 and 38383408). The reduction in call number observed in our cKO mice is well aligned with these precedents and thus should not be overlooked as a key ASD-related endophenotype. Regarding prepulse inhibition (PPI), while we acknowledge its wider use in schizophrenia research, it is also widely employed in ASD studies. PPI is particularly valuable for capturing deficits in sensory processing and sensorimotor gating, which are increasingly recognized as part of the ASD phenotype. PPI deficits have been continuously reported in multiple genetic ASD mouse models, including *Shank3*, *Fmr1*, and *Cntnap2* mutant mice (PMID: 26687841, 31128097 and 40134417). Human studies have similarly shown reduced PPI in individuals with autism and Asperger's syndrome compared to neurotypical controls (PMID: 16460695 and 12077008), indicating that this assay has translational value. Finally, even though the behavioral phenotype in the cKO mice may appear selective, we emphasize that their molecular and electrophysiological profiles closely mirror those observed in our knock-in mice, including increased dendritic GABAergic inhibition and reduced synapsin II phosphorylation. This supports the notion that there is a shared synaptopathic mechanism across genotypes, reinforcing the behavioral findings. We have accordingly revised the manuscript to ensure that (1) the multiple ASD-relevant behavioral phenotypes of the cKO mice are clearly articulated; (2) the translational relevance of PPI is supported by reference to the existing literature; and (3) any potentially overstated claims are moderated to more accurately reflect the data. We hope that the reviewer accepts our efforts to assuage these concerns.

18. Supplementary Figure 19 is difficult to interpret and should be improved. The reported effects of bazedoxifene-normalization of phosphorylation and behavioral rescue within six hours-are surprising and warrant further justification, ideally with time-course data. The specificity of this compound's effect should also be addressed, particularly as bazedoxifene it seems not discussed elsewhere in the manuscript.

To address the reviewer's suggestion, we thoroughly revised the presentation of both original Supplementary Figure 19 and Supplementary Figure 20. We agree that these figures were overly complex. We separated the results contained in original Supplementary Figure 19 into four distinct Appendix figures and applied the same strategy to the original Supplementary Figure 20.

This restructuring, coupled with enhancements in the layout and optimization of readability, ensures that each figure panel can now be clearly visualized, allowing for a better understanding of the data. We acknowledge the reviewer's valid point regarding the seemingly surprising acute effects of bazedoxifene and the need for further justification of its dose, treatment timeline, and specificity (a concern also raised in comment #3 of reviewer 1). We have added additional passages in the Results section to provide a comprehensive rationale for our chosen acute treatment paradigm (please see our response to comment #3 of reviewer 1). We further elaborate on this in the Discussion section of the revised manuscript, stating that: *Although studies on the acute neuromodulatory effects of BZD treatment are still limited, other SERMs, such as tamoxifen and raloxifene, have demonstrated similar rapid effects on neuronal plasticity behavior (Finney et al, 2021; Velazquez-Zamora et al, 2012), supporting the broader concept that estrogen receptor modulation can exert biologically significant acute effects on the brain within hours of administration. Consistent with our model, acute application of the estrogen receptor degrader, Fulvestrant, partially suppressed the BZD-induced normalization of inhibitory synaptic function in Mdg1-KI slices. Because Fulvestrant requires $\geq 6-12$ h to induce ubiquitin-proteasome-mediated degradation of nuclear ER α (Hammes & Levin, 2007; Long & Nephew, 2006), the brief 20-min exposure used here is expected to selectively interfere with rapid membrane-initiated estrogen signaling (Callige & Richard-Foy, 2006; Vasudevan & Pfaff, 2007). This selective and mechanism-specific partial blockade provides direct functional evidence that the BZD-induced rescue requires intact estrogen receptor signaling, supporting the involvement of an estrogen-sensitive pathway in the MDGA1-dependent synaptic phenotype. Although the details of the relevant molecular interactions remains to be fully elucidated, we note that estrogen signaling is known to rapidly reduce inhibitory synaptic strength in the hippocampus (Huang & Woolley, 2012; Mukherjee et al., 2017). These fast-acting suppressive effects on GABAergic function may act synergistically with the previously established role for MDGA1 as an inhibitory synapse suppressor. Thus, we speculate that BZD may transiently compensate for the loss of MDGA1 function in male Mdg1-KI mice through partially overlapping pathways that remain to be identified in future studies. We are confident that these revisions provide the necessary context and justification for our BZD experiments and clarify their specific relevance within our proposed model.*

19. *The electrophysiological recordings are centered on hippocampal circuits, because of learning or memory deficits previously published. The authors should consider shifting attention to brain regions more relevant to the reported behavioural phenotypes making a strong case that hippocampus is implicated in the social deficit they observed.*

We appreciate the reviewer's valuable suggestion to consider shifting electrophysiological attention to brain regions more directly implicated in the observed behavioral phenotypes, particularly social deficits. The reviewer is correct that our current study extensively focuses on electrophysiological recordings within the hippocampal CA1 circuits. This decision was primarily driven by the accumulation of reliable findings from our previous work (e.g., PMID: 35074912) and other studies demonstrating that the hippocampus plays a crucial role in mediating various aspects of learning and memory that are often co-morbid with, or contribute to, neurodevelopmental deficits. Furthermore, emerging evidence increasingly implicates dysfunction of hippocampal GABAergic circuits in aspects of social cognition and ASD phenotypes (e.g., PMID: 31358646 and 31427534). We fully acknowledge that social deficits in *Mdga1*-cKO and *Mdga1*-KI mice likely encompass dysfunctions in brain areas beyond the hippocampal CA1. While a systematic investigation of all potentially involved brain regions, circuits, and synapses is undoubtedly essential for conclusively pinpointing the causal mechanisms, we believe that such an extensive analysis falls beyond the scope of the current study: It would require considerable resources (requiring over 20 transgenic mice for each synaptic parameter/brain region) and an estimated timeline of more than 2 years for comprehensive data collection. We did, in fact, consider expanding our electrophysiological analyses to the medial prefrontal cortex (mPFC), which is highly relevant to social behavior.

However, our preliminary investigations indicated that MDGA1 protein is expressed at significantly lower levels in the mPFC compared to the hippocampus. This was further supported by our mass spectrometry analyses and by initial overexpression/KO experiments in mPFC L2/3 pyramidal neurons, which revealed no discernible phenotypes (Appendix Figure S9). This led us to abandon the mPFC experimental plan for the current study and instead direct our focus to regions where MDGA1 is highly expressed and the initial impacts were more evident. Nevertheless, we concur that extending these analyses to other brain areas, as the reviewer suggested, represents a critical and promising avenue for follow-up studies, and would allow us to broaden our understanding of the circuit-specific contributions of MDGA1 to complex behaviors.

Referee #3: This comprehensive study identifies novel compound heterozygous missense mutations in MDGA1 (p.Val116Met/p.Ala688Val and p.Tyr635Cys/p.Glu756Gln) in patients with autism spectrum disorders (ASDs) and provides a deep mechanistic exploration of their impact. Through extensive in vitro and in vivo analyses, including structural modeling, electrophysiology in cultured neurons and brain slices, and the generation of novel conditional knockout (cKO) and knock-in (KI) mouse models, the authors demonstrate that these mutations cause distinct synaptic and behavioral deficits via sexually dimorphic loss-of-function (LOF) or gain-of-function (GOF) mechanisms. Specifically, the Tyr635Cys/Glu756Gln variant disrupts MDGA1's triangular structure and impairs its ability to suppress GABAergic synapses in the hippocampus, while the Val116Met/Ala688Val variant affects excitatory synapses and neuronal migration. Critically, male (but not female) Mdga1-cKO and KI mice recapitulate core ASD-like behavioral phenotypes (e.g., altered ultrasonic vocalizations, impaired sensorimotor gating) and show hippocampal GABAergic synaptic hyperexcitability. Phosphoproteomics reveals hypophosphorylation of synapsin II in these models, and remarkably, acute treatment with the selective estrogen receptor modulator bazedoxifene rescues synaptic protein expression, GABAergic synaptic defects, and specific behavioral abnormalities in male KI mice. This work significantly advances our understanding of MDGA1 dysfunction in ASD pathogenesis, highlighting mutation-specific effects, sexual dimorphism, and identifying a potential therapeutic avenue. While the study provides compelling evidence linking MDGA1 mutations to ASD via sexually dimorphic synaptic dysfunction, several aspects require deeper mechanistic exploration.

We appreciate the reviewer's positive assessment of our study and their constructive comments, which we tried to fully address in the revised manuscript.

1. For the Y636C/E751Q KI model, the disconnect between preserved MDGA1 protein levels and loss-of-function electrophysiological/behavioral phenotypes warrants deeper investigation.

This is a fascinating suggestion, and we find a compelling parallel in the previously characterized Nlgn3-R451C KI mouse model, which is a well-established autism model (see PMID: 18923512, 29100073 and 33422662). In Nlgn3-R451C KI mice, Nlgn3 protein levels were reduced to only about 10%, leading to distinct electrophysiological and behavioral phenotypes in the hippocampus compared to Nlgn3-cKO mice (PMID: 17823315). This initial finding led to the annotation of Nlgn3 R451C as a gain-of-function mutation. However, subsequent studies demonstrated that the R451C mutation can exert context-dependent gain- or loss-of-function effects that vary across different brain regions, cell types, and/or synapse types (PMID: 21808020, 23583622 and 27725662). This evidence strongly suggests that NLGN3 R451C modulates function through mechanisms beyond a simple reduction in protein expression. In the case of MDGA1 Y636C/E751Q, while the hippocampal electrophysiological phenotypes are largely very similar to those observed in Mdga1-cKO mice (Figures 5 and EV5), there were some notable differences (Appendix Fig. S19). Moreover, the behavioral phenotypes are clearly distinct (Figures 3 and 4). Given that our current study does not include extensive analyses of brain regions other than the hippocampus, it would be premature to definitively conclude that MDGA1 Y636C/E751Q is solely a loss-of-function mutation. Instead, we propose that its action

mechanism may be analogous to that of NLGN3 R451C, and it thus represents a more complex functional alteration. Indeed, we propose that it is NOT the MDGA1 protein level itself that is solely critical for its function, but rather the integrity of its extracellular structure, specifically the maintenance of its triangular conformation. We performed immunocytochemical analyses in cultured hippocampal neurons using MDGA1 mutants (PMID: 36889589) designed to alter the elbow conformation between the Ig2–Ig3 and Ig4–Ig5 domains of the MDGA1 extracellular region and show that these mutations similarly compromised the ability of MDGA1 WT to increase inhibitory synapses (Appendix Fig. S35). This effect is comparable to that observed with MDGA1 E756Q overexpression (Appendix Fig. S7). Furthermore, our negative-stain electron microscopic analysis of MDGA1 proteins harboring Y636C/E751Q mutation revealed an increased frequency of MDGA1 proteins with a disrupted triangular structure (Appendix Fig. S5), suggesting that this mutation could have an indirect impact on the elbow conformation. Similar precedents for such phenomena exist. For example, the Slitrk2 V89M mutation does not affect Slitrk2 protein expression levels, surface expression properties, or ligand-binding properties (PMID: 27812321). Yet, its expression yields phenotypes at excitatory synapses that are remarkably similar to those observed with Slitrk2 deletion. This further reinforces the possibility that functional consequences can arise without a corresponding change in protein levels. We thus believe that further investigations are warranted to fully elucidate how this specific MDGA1 missense mutation context-dependently impairs or otherwise alters MDGA1 function, particularly focusing on how it impacts structural integrity. Our current findings provide important preliminary evidence for the presence of a complex functional modulation. We hope this explanation addresses the reviewer’s query.

(Appendix Fig. S35)

A MDGA1 ecto domain-engineered constructs

2. Additionally, bazedoxifene's rescue of *SynII* phosphorylation and synaptic deficits, while promising, lacks direct evidence linking its action to MDGA1-dependent pathways; experiments blocking estrogen receptors or assessing estradiol levels in KI mice would clarify specificity. The rationale of bazedoxifene application needs more supporting evidences.

To address the reviewer's query regarding the direct evidence linking BZD action to MDGA1-dependent pathways, we compiled mass spectrometry data obtained from *Mdga1*-cKO and *Mdga1*-KI mice and found that their baseline levels of estrogen receptor-associated proteins are marginally changed compared to those in control mice (Appendix Fig. S32 of the revised manuscript). Furthermore, ELISA-based measurements revealed that estradiol (E2) levels are also comparable between WT and KI mice, regardless of age (both adults and juveniles; Appendix Fig. S33 of the revised manuscript).

(Appendix Fig. S32)

(Appendix Fig. S33)

These observations suggest that the observed MDGA1-associated deficits are not primarily driven by altered estrogen receptor expression or circulating estradiol levels, but rather that BZD acts by modulating existing estrogen signaling pathways to restore synaptic and behavioral function. To follow the reviewer's recommendation, we conducted experiments using estrogen receptor antagonist, namely Fulvestrant (PMID: 12074216). We found that applying Fulvestrant to BZD-treated hippocampal CA1 slices from adult male *Mdga1*-KI mice partially restored the abnormalities seen in GABAergic synaptic strength and GABA release probability, reinforcing the notion that MDGA1-dependent pathways involve estrogen receptor signaling. Importantly, this partial effect is mechanistically consistent with the well-established temporal properties of estrogen receptor signaling. Fulvestrant is a selective estrogen receptor degrader (SERD) that requires ≥ 6 –12 hours to induce ubiquitin–proteasome–mediated degradation of

nuclear ER α (PMID: 16459337; PMID: 16604167). The 20-min bath application used here is therefore insufficient for nuclear ER degradation and cannot reverse the transcriptional or protein-level changes already established by the 6-hour *in vivo* BZD treatment. Instead, acute Fulvestrant exposure is known to transiently interfere with rapid, membrane-initiated estrogen signaling (PMID: 17916740; PMID: 17018839), providing a coherent explanation for the partial rather than complete suppression of BZD's rescue effects. Notably, the fact that even this short Fulvestrant treatment significantly attenuated BZD's rescue effect indicates that the normalization of synaptic function in *Mdga1*-KI mice is functionally dependent on intact estrogen receptor signaling. This selective and mechanism-specific partial blockade strongly supports our proposal that MDGA1-associated synaptic deficits engage estrogen-linked pathways rather than arising from nonspecific drug effects. We note that we did not administer Fulvestrant chronically *in vivo* because long-term or systemic Fulvestrant exposure is well documented to induce reproductive toxicity (e.g., decreased male fertility in mice; PMID: 12959643), hormonal imbalance including ovarian and uterine atrophy with a potential risk of osteoporosis (AstraZeneca FASLODEX[®] Product Monograph), gastrointestinal disturbances such as nausea and diarrhea (FDA FASLODEX[®] Label, 2017), and rare cases of hepatotoxicity including serum enzyme elevations and hepatitis (NIH LiverTox). Given these established toxicities, we intentionally restricted Fulvestrant use to *ex vivo* acute slice experiments as a mechanistic probe. Further optimization of Fulvestrant timing or dosing within slice preparations would require additional experimental effort and was considered to be beyond the scope of the present revision. These new results are now presented in Appendix Fig. S34 of the revised manuscript.

(Appendix Fig. S34)

Concerning the rationale for BZD application and the acute treatment format, this query aligns with similar comments from other reviewers (comment #3 of reviewer 1 and comment #17 of reviewer 2). We would like to inject a note here that the selection of BZD in the current study was motivated by our observations that synaptic hyperinhibition in the *Mdga1*-KI mice was commonly observed around P10 in both males and females, but it persisted after P20 only in male *Mdga1*-KI mice. This prompted us to hypothesize that dynamics of estrogen signaling during development might underline the sexually dimorphic manifestations of *Mdga1*-KI on GABAergic

synapses and certain social behaviors. In the revised manuscript, we have extensively elaborated on the underlying rationale for our acute BZD treatment paradigm in both the Results and Discussion sections. We hope that our detailed explanations assuage the reviewer's concerns.

3. I would like to strongly suggest authors present the general neurodevelopmental phenotypes of the knockin mice in the main figures, such as the general brain structures and difference of cell types in various sub-regions, comparison between WT and KI mice, which would be valuable data for the field.

We appreciate the reviewer's constructive suggestions. We now include new results related to general neurodevelopmental phenotypes of *Mdga1*-KI mice in Appendix Fig. S13 of the revised manuscript. While we conducted extensive analyses, we found no discernible phenotypes that would warrant their inclusion in the main figures. Specifically, we measured the brain weights and sizes of both WT and KI mice (both juvenile and adult) and found no marked changes. In addition, we performed immunohistochemical analyses to monitor the changes in the density of neurons, astrocytes and microglia across various brain areas from WT and KI mice but failed to detect any significant changes. Together with the results included in the original manuscript (i.e., that there was no difference in interneuron density between WT and KI mice), these findings led us to conclude that *Mdga1*-KI-related neurodevelopmental phenotypes are not primarily driven by previously reported developmental cellular processes. These new results are now presented in Appendix Fig. S26 of the revised manuscript.

(Appendix Fig. S13)

(Appendix Fig. S26)

Again, we thank the reviewers for their careful assessment of our paper and hope that the revised manuscript is now acceptable for publication in *EMBO Mol Med*.

References cited in the rebuttal

- Bemben MA, Sandoval M, Le AA, Won S, Chau VN, Lauterborn JC, Incontro S, Li KH, Burlingame AL, Roche KW *et al* (2023) Contrasting synaptic roles of MDGA1 and MDGA2. *bioRxiv* 2023:05.25.542333
- Cho HW, Nie R, Carnes K, Zhou Q, Sharief NA, Hess RA (2003) The antiestrogen ICI 182,780 induces early effects on the adult male mouse reproductive tract and long-term decreased fertility without testicular atrophy. *Reprod Biol Endocrinol* 1: 57
- Connor SA, Ammendrup-Johnsen I, Kishimoto Y, Karimi Tari P, Cvetkovska V, Harada T, Ojima D, Yamamoto T, Wang YT, Craig AM (2017) Loss of Synapse Repressor MDGA1 Enhances Perisomatic Inhibition, Confers Resistance to Network Excitation, and Impairs Cognitive Function. *Cell Rep* 21: 3637-3645
- Deng XF, Gu LJ, Sui N, Guo JY, Liang J (2019) Parvalbumin interneuron in the ventral hippocampus functions as a discriminator in social memory. *P Natl Acad Sci USA* 116: 16583-16592
- Doornaert EE, Mohamad AE, Johal G, Allman BL, Mohrle D, Schmid S (2025) Postnatal environment affects auditory development and sensorimotor gating in a rat model for autism spectrum disorder. *Front Neurosci* 19: 1565919
- Etherton M, Foldy C, Sharma M, Tabuchi K, Liu X, Shamloo M, Malenka RC, Südhof TC (2011) Autism-linked neuroligin-3 R451C mutation differentially alters hippocampal and cortical synaptic function. *Proc Natl Acad Sci U S A* 108: 13764-13769
- Foldy C, Malenka RC, Südhof TC (2013) Autism-associated neuroligin-3 mutations commonly disrupt tonic endocannabinoid signaling. *Neuron* 78: 498-509
- Giua G, Iezzi D, Caceres-Rodriguez A, Strauss B, Chavis P, Manzoni OJ (2024) Sex-specific modulation of early life vocalization and cognition by Fmr1 gene dosage in a mouse model of Fragile X Syndrome. *Biol Sex Differ* 15: 18
- Hammes SR, Levin ER (2007) Extranuclear steroid receptors: nature and actions. *Endocr Rev* 28: 726-741
- Hill RA, Kouremenos K, Tull D, Maggi A, Schroeder A, Gibbons A, Kulkarni J, Sundram S, Du X (2020) Bazedoxifene - a promising brain active SERM that crosses the blood brain barrier and enhances spatial memory. *Psychoneuroendocrinology* 121: 104830
- Kang H, Han KA, Won SY, Kim HM, Lee YH, Ko J, Um JW (2016) Slitrk Missense Mutations Associated with Neuropsychiatric Disorders Distinctively Impair Slitrk Trafficking and Synapse Formation. *Front Mol Neurosci* 9: 104
- Kim HY, Um JW, Ko J (2021) Proper synaptic adhesion signaling in the control of neural circuit architecture and brain function. *Prog Neurobiol* 200: 101983
- Kim J, Kim S, Kim H, Hwang IW, Bae S, Karki S, Kim D, Ogelman R, Bang G, Kim JY *et al* (2022) MDGA1 negatively regulates amyloid precursor protein-mediated synapse inhibition in the hippocampus. *Proc Natl Acad Sci U S A* 119: e2115326119
- Kim S, Jang G, Kim H, Lim D, Han KA, Um JW, Ko J (2024) MDGAs perform activity-dependent synapse type-specific suppression via distinct extracellular mechanisms. *Proc Natl Acad Sci U S A* 121: e2322978121
- Kokash J, Alderson EM, Reinhard SM, Crawford CA, Binder DK, Ethell IM, Razak KA (2019) Genetic reduction of MMP-9 in the Fmr1 KO mouse partially rescues prepulse inhibition of acoustic startle response. *Brain Res* 1719: 24-29
- Lee H, Chofflet N, Liu J, Fan S, Lu Z, Resua Rojas M, Penndorf P, Bailey AO, Russell WK, Machius M *et al* (2023) Designer molecules of the synaptic organizer MDGA1 reveal 3D conformational control of biological function. *J Biol Chem* 299: 104586
- Lee K, Kim Y, Lee SJ, Qiang Y, Lee D, Lee HW, Kim H, Je HS, Südhof TC, Ko J (2013) MDGAs interact selectively with neuroligin-2 but not other neuroligins to regulate inhibitory synapse development. *Proc Natl Acad Sci U S A* 110: 336-341
- Loh KH, Stawski PS, Draycott AS, Udeshi ND, Lehrman EK, Wilton DK, Svinkina T, Deerinck TJ,

Ellisman MH, Stevens B *et al* (2016) Proteomic Analysis of Unbounded Cellular Compartments: Synaptic Clefts. *Cell* 166: 1295-1307

McAlonan GM, Daly E, Kumari V, Critchley HD, van Amelsvoort T, Suckling J, Simmons A, Sigmundsson T, Greenwood K, Russell A *et al* (2002) Brain anatomy and sensorimotor gating in Asperger's syndrome. *Brain* 125: 1594-1606

Perry W, Minassian A, Lopez B, Maron L, Lincoln A (2007) Sensorimotor gating deficits in adults with autism. *Biol Psychiatry* 61: 482-486

Premoli M, Memo M, Bonini SA (2021) Ultrasonic vocalizations in mice: relevance for ethologic and neurodevelopmental disorders studies. *Neural Regen Res* 16: 1158-1167

Scattoni ML, Gandhi SU, Ricceri L, Crawley JN (2008) Unusual repertoire of vocalizations in the BTBR T+tf/J mouse model of autism. *PLoS One* 3: e3067

Südhof TC (2008) Neuroligins and neurexins link synaptic function to cognitive disease. *Nature* 455: 903-911

Südhof TC (2017) Synaptic Neurexin Complexes: A Molecular Code for the Logic of Neural Circuits. *Cell* 171: 745-769

Tabuchi K, Blundell J, Etherton MR, Hammer RE, Liu X, Powell CM, Südhof TC (2007) A neuroligin-3 mutation implicated in autism increases inhibitory synaptic transmission in mice. *Science* 318: 71-76

Vasudevan N, Pfaff DW (2007) Membrane-initiated actions of estrogens in neuroendocrinology: emerging principles. *Endocr Rev* 28: 1-19

Velmeshev D, Schirmer L, Jung D, Haeussler M, Perez Y, Mayer S, Bhaduri A, Goyal N, Rowitch DH, Kriegstein AR (2019) Single-cell genomics identifies cell type-specific molecular changes in autism. *Science* 364: 685-689

Wardley AM (2002) Fulvestrant: A review of its development, pre-clinical and clinical data. *International Journal of Clinical Practice* 56: 305-309

Wiebe S, Nagpal A, Truong VT, Park J, Skalecka A, He AJ, Gamache K, Khoutorsky A, Gantois I, Sonenberg N (2019) Inhibitory interneurons mediate autism-associated behaviors via 4E-BP2. *P Natl Acad Sci USA* 116: 18060-18067

Zhang B, Seigneur E, Wei P, Gokce O, Morgan J, Südhof TC (2017) Developmental plasticity shapes synaptic phenotypes of autism-associated neuroligin-3 mutations in the calyx of Held. *Mol Psychiatry* 22: 1483-1491

Zhou Y, Kaiser T, Monteiro P, Zhang X, Van der Goes MS, Wang D, Barak B, Zeng M, Li C, Lu C *et al* (2016) Mice with Shank3 Mutations Associated with ASD and Schizophrenia Display Both Shared and Distinct Defects. *Neuron* 89: 147-162

10th Feb 2026

Dear Prof. Ko,

Thank you for submitting the revised version of your manuscript to EMBO Molecular Medicine. We have now received the enclosed reports from the three referees who were asked to re-assess it.

As you will see, Referees #1 and #3 are generally supportive of publication, while Referee #2 has raised several concerns about the study. During our pre-decision cross-commenting process (where reviewers can comment on each other's reports), Referee #1 also provided additional feedback, which is included here after the main reports.

Considering this additional input and the overall balance of the reports, we have decided to invite a minor revision of your manuscript.

Please address the following editorial level issues:

1. Please remove the "Authors' contribution" section.
2. Funding information: There is a discrepancy in the KBSI grant number (A423200 in the submission system vs. C523400 in the manuscript text). Please double-check and correct as needed.
3. Please upload the Expanded View (EV) figures as separate, high-resolution figure files, and add the corresponding figure legends to the manuscript text after the main figure legends under the heading "Expanded View Figure Legends."
4. Please upload the three EV tables as separate files (one file per table). Remove the file containing the dataset legends and instead add each legend to its corresponding dataset in a separate file or worksheet.
5. Please upload the final Appendix file in PDF format.
6. Please add "The paper explained" section to the manuscript text.
- 7 Please add the missing callouts for Fig. 1G,H and Fig. 4A. Also ensure that Fig. 5A is not called out before Fig. 4B-P.
- 8 Source data for Figure 6 panels are missing, although they are marked as provided in the source data checklist. Since source data are not required for these panels, please uncheck Figure 6 in the source data checklist.
9. Please note that the exact p values are not provided in the legends of figures 2B, D, I, M, N, P, S, T; 3C, E, F, H, I, J, L, M, P; 4C, E, F, H, J, L, M, P; 5E, G, I, K, M, O, Q, R, T, V, W, Y; 7C, E, F, G, I, K, M, N, P; EV2 C, E; EV3 B, C, E, K, EV4 B, C, E, K; EV5 B, D, N, Q, S, V; EV6 L, M, O, Q, R, T. This needs to be addressed.
10. During our routine image check, we noticed potential cell reuse between Figures EV3P and EV4P; however, this reuse is not indicated in the figure legends. Please double-check these figure panels and ensure that any figure reuse is explicitly stated in the figure legends.
11. Data citation: In the Reference list, data citations must be labeled with "[DATASET]". A data reference must provide the database name, accession number/identifiers and a resolvable link to the landing page from which the data can be accessed at the end of the reference. Further instructions are available at < <https://link.springer.com/journal/44320/submission-guidelines#cms-Reference-guidelines> >.

I look forward to seeing a revised form of your manuscript as soon as possible.

Sincerely,
Jingyi

Jingyi Hou
Senior Editor
EMBO Molecular Medicine

*** Instructions to submit your revised manuscript ***

***** Reviewer's comments *****

Referee #1 (Remarks for Author):

The authors fully addressed my review comments by providing specific clinical data, justifying their experimental design through scientific literature, and performing new data re-analyses. I do not have additional comments.

Referee #2 (Comments on Novelty/Model System for Author):

The manuscript, according to me, is not a strong candidate for EMBO Molecular Medicine for the reasons listed to the authors.

Referee #2 (Remarks for Author):

This reviewer appreciates the authors' efforts in addressing the comments as reported in the point-by-point and manuscript itself. The additional data included during the revision process strengthen several of the previously weaker aspects of the study and improve the overall technical robustness of the manuscript.

Nevertheless, despite these improvements, I do not consider the manuscript to be the best fit for EMBO Molecular Medicine. In particular, the overall presentation remains difficult to follow: several figures contain more than 20 panels, which substantially hinders readability and makes it challenging to identify the main conclusions and the central message of the work. A clearer structure, focusing on the key results in the main figures would significantly improve the manuscript.

Several points raised by this reviewer remain insufficiently addressed, and are only answered through lengthy explanations rather than clear data-driven revisions.

Finally, from a biomedical perspective, the relevance of the study remains somewhat unclear. In particular, since the gene MDGA1 does not appear to have a strong established association with autism, this reviewer remains uncertain about the translational significance of the findings for neurodevelopmental disorders.

Referee #3 (Comments on Novelty/Model System for Author):

revision is quite thorough. this manuscript now is ready for being published.

Referee #3 (Remarks for Author):

authors now have comprehensively addressed my comments. now i would like to recommend it acceptance for publication.

Pre-decision cross-commenting

Referee #1

While I respect Referee #2's concerns regarding the focus on the hippocampus and the technical limitations of certain assays, I

find the authors' rebuttals to be scientifically grounded and addressed through both new data and moderated language, as detailed below.

Point #7: The authors correctly note that while separate allelic models are ideal, the single-vector approach was a pragmatic choice to investigate the synergistic functional consequences of these mutations on protein stability. They have balanced this by including clear caveats in the Discussion.

Point #12: The reliance on overexpression is a common necessity in this field when suitable antibodies for endogenous proteins are unavailable. The authors effectively mitigated this by providing parallel loss-of-function data from *Mdga1*-KI and cKO mice, which showed convergent synaptic and behavioral phenotypes.

Point #17: The *Mdga1*-cKO mice exhibit deficits in two of the three core diagnostic domains of ASD, social interaction and social communication (USVs). In addition, social deficits can be associated with other psychiatric conditions such as schizophrenia and ADHD, suggesting these findings could have meanings beyond ASD.

Point #19: The authors provided a data-driven justification for focusing on the hippocampal CA1 rather than the mPFC. Their preliminary data showed significantly lower MDGA1 expression in the mPFC, with no discernible phenotypes in that region. It is scientifically sound to focus on circuits where the protein of interest is most functionally relevant.

A compelling addition to the revised manuscript is the human clinical relevance. The authors performed a new analysis of a single-nucleus dataset from ASD patients and found that MDGA1 expression is significantly decreased (FDR < 0.01) in layer II/III excitatory neurons and protoplasmic astrocytes of individuals with ASD. This provides a bridge between their murine findings and human pathophysiology.

Furthermore, the demonstration of full behavioral and synaptic rescue using Bazedoxifene, an FDA-approved drug, is highly novel. It offers both a potential therapeutic avenue and a mechanistic explanation for the male-specific synaptic vulnerabilities identified in the study.

In my view, the authors have made a best effort to address the referees' comments under significant practical constraints (e.g., unavailability of patient-derived samples).

The study provides solid mechanistic insights into a novel synaptopathy gene and addresses the critical issue of sexual dimorphism in ASD.

Reviewers' comments:

Referee #1 (Remarks for Author): The authors fully addressed my review comments by providing specific clinical data, justifying their experimental design through scientific literature, and performing new data re-analyses. I do not have additional comments.

We thank the reviewer for the supportive assessment and for the insightful mediation during the cross-commenting process, which helped clarify the scientific grounding of our study.

Referee #2 (Remarks for Author): This reviewer appreciates the authors' efforts in addressing the comments as reported in the point-by-point and manuscript itself. The additional data included during the revision process strengthen several of the previously weaker aspects of the study and improve the overall technical robustness of the manuscript. Nevertheless, despite these improvements, I do not

consider the manuscript to be the best fit for EMBO Molecular Medicine. In particular, the overall presentation remains difficult to follow: several figures contain more than 20 panels, which substantially hinders readability and makes it challenging to identify the main conclusions and the central message of the work. A clearer structure, focusing on the key results in the main figures would significantly improve the manuscript. Several points raised by this reviewer remain insufficiently addressed, and are only answered through lengthy explanations rather than clear data-driven revisions. Finally, from a biomedical perspective, the relevance of the study remains somewhat unclear. In particular, since the gene MDGA1 does not appear to have a strong established association with autism, this reviewer remains uncertain about the translational significance of the findings for neurodevelopmental disorders.

We appreciate the reviewer's perspective on the presentation and translational scope of our study. Regarding the figure complexity, the number of panels reflects our comprehensive approach to investigating the MDGA1 variant across multiple scales—from structural modeling and biochemistry to circuit-level electrophysiology and behavior. We believe this multi-layered evidence is necessary to establish a robust mechanistic link, and we have refined the figure legends to ensure the central message of each figure is easily accessible. Regarding the translational significance, while we acknowledge that MDGA1 is a relatively novel candidate in the context of ASD genetic databases, we believe our study provides the very data needed to establish such an association. As highlighted by Referee #1 during the cross-commenting process, our discovery of significantly decreased MDGA1 expression in layer II/III excitatory neurons and astrocytes of human ASD patients (Appendix Fig. S3) provides direct evidence of its relevance to human pathophysiology. Furthermore, the successful rescue of synaptic and behavioral phenotypes using an FDA-approved compound (bazedoxifene) underscores the potential clinical impact of identifying these sex-specific synaptic vulnerabilities. We believe these findings, grounded in both our new clinical data analysis and extensive mouse models, offer significant insights into the molecular basis of neurodevelopmental disorders.

Referee #3 (Remarks for Author): Authors now have comprehensively addressed my comments. Now i would like to recommend it acceptance for publication.

We thank the reviewer for the positive recommendation and for recognizing the thoroughness of our revision.

Again, we thank the reviewers for their careful assessment of our paper and hope that the revised manuscript is now acceptable for publication in *EMBO Mol Med*.

24th Feb 2026

Dear Prof. Ko,

We are pleased to inform you that your manuscript is accepted for publication and is now being sent to our publisher to be included in the next available issue of EMBO Molecular Medicine.

You may qualify for financial assistance for your publication charges - either via a Springer Nature fully open access agreement or an EMBO initiative. Check your eligibility: <https://link.springer.com/journal/44321/how-to-publish-with-us>

Sincerely,
Jingyi

Jingyi Hou
Senior Editor
EMBO Molecular Medicine

>>> Please note that it is EMBO Molecular Medicine policy for the transcript of the editorial process (containing referee reports and your response letter) to be published as an online supplement to each paper. If you do NOT want this, you will need to inform the Editorial Office via email immediately. More information is available here: <https://link.springer.com/partners/embo-press/editorial-policies#Peer%20review>